# HYPERGRAPH CONVOLUTIONAL NETWORKS VIA EQUIVALENCY BETWEEN HYPERGRAPHS AND UNDIRECTED GRAPHS

## ABSTRACT

As a powerful tool for modeling the complex relationships, hypergraphs are gaining popularity from the graph learning community. However, commonly used algorithms in deep hypergraph learning were not specifically designed for hypergraphs with *edge-dependent vertex weights* (EDVWs). To fill this gap, we build the equivalency condition between EDVW-hypergraphs and undirected simple graphs, which enables utilizing existing undirected graph neural networks as subroutines to learn high-order interactions induced by EDVWs of hypergraphs. Specifically, we define a generalized hypergraph with vertex weights by proposing a unified random walk framework, under which we present the equivalency condition between generalized hypergraphs and undigraphs. Guided by the equivalency results, we propose a Generalized Hypergraph Convolutional Network (GHCN) architecture for deep hypergraph learning. Furthermore, to improve the long-range interactions and alleviate the over-smoothing issue, we further propose the Simple Hypergraph Spectral Convolution (SHSC) model by constructing the Discounted Markov Diffusion Kernel from our random walk framework. Extensive experiments from various domains including social network analysis, visual objective classification, and protein fold classification demonstrate that the proposed approaches outperform state-of-the-art spectral methods with a large margin.

## 1 INTRODUCTION

Hypergraphs, whose edges link to the arbitrary number of vertices to model high-order relationships, attract much attention recently from researchers in the graph learning community (Feng et al., 2019; Jiang et al., 2019; Zhang et al., 2021). The topology of hypergraph can be considered to be embedded in vertex weights, which represents the connection strength between hyperedges and vertices (Chitra & Raphael, 2019). The vertex weights can be divided into two categories, edge-independent and edge-dependent (Chitra & Raphael, 2019; Hayashi et al., 2020), depending on whether the vertex weights are related to the incident hyperedges or not (EDVW-hypergraph implies that each vertex $v$ is assigned a weight $q_e(v) \in \mathbb{R}$ for each incident hyperedge $e$). Many hypergraph neural networks have been proposed (Feng et al., 2019; Jiang et al., 2019; Yadati, 2020; Huang & Yang, 2021) to handle hypergraphs with edge-independent vertex weights (EIVW-hypergraph). However, rigorous study for hypergraphs with edge-dependent vertex weights (EDVW-hypergraph) is still lacked, though they enjoy stronger expressive power (Ding & Yilmaz, 2010; Huang et al., 2010; Li et al., 2018a; Zhang et al., 2018).

In this work, we are devoted to spectral-based methods, which are considered to be robust and to allow for simple model property analysis (Wu et al., 2020). Unfortunately, the spectral theory of EDVW-hypergraph has not been well established. A recent work (Chitra & Raphael, 2019) provides the spectral theory for EDVW-hypergraphs, and indicates that the defined EDVW-hypergraphs is equivalent to digraphs. Though digraph convolutional networks can be directly used for EDVW-hypergraphs, most existing convolutional algorithms for digraphs are more or less related to undirected graphs, e.g., (Monti et al., 2018; Li et al., 2020; Ma et al., 2019; Tong et al., 2020). Essentially, they transform digraphs to undigraphs with various techniques due to the challenges of directed graph Laplacian. In addition, key theoretical analysis of digraph convolutional networks properties,

Figure 1: An example of hypergraph and its equivalent weighted undirected graph ($\mathbf{W}^C = (\mathbf{W}^C)^\top$), where $q^i(\cdot) = Q_i(\cdot, e), i \in \{1, 2\}$. Here, $\mathbf{Q_1} = \mathbf{Q_2}$ are both edge-dependent. The characteristics of the hypergraph are encoded in the weighted incidence matrices (i.e. vertex weights) $\mathbf{Q_1}, \mathbf{Q_2}$. $\mathbf{W}^C := \mathbf{Q_2}\mathbf{D}_e^{-1}\mathbf{Q_2}^\top$ denotes the edge-weight matrix of the clique graph and can be viewed as the embedding of high-order relationships.

such as expressive power (Xu et al., 2019) or the performance of models in deep layers (Li et al., 2018b), yet remains open.

An arguably more succinct way for EDVW-hypergraph learning is to use undigraph convolutional networks. Compared with digraphs, the undigraph convolutional networks have been extensively studied and there exist various effective strategies for analyzing their properties, to name a few (Kipf & Welling, 2017; Chen et al., 2020; Zhu & Koniusz, 2021). Furthermore, one can take established techniques on undigraphs as a subroutine for hypergraph learning, which would largely ease hypergraph learning in real-world scenarios. It remains unclear, however, whether EDVW-hypergraphs can be made equivalent to undigraphs in a principled way, thereby hindering this promising prospect.

In this paper, we focus on constructing the equivalency condition between hypergraphs and undigraphs. This makes it possible to utilize undigraphs as low-order encoders of hypergraphs, and thus enabling hypergraph learning directly from the equivalent undirected graphs by means of undigraph convolutional networks. Along this route, we have made the following contributions:

1) We define a generalized hypergraph capturing EDVWs via designing a two-step unified random walk framework. In addition, we construct the equivalency conditions to fill the gap between the generalized hypergraph and corresponding undigraph.

2) We present a unified Laplacian from the equivalency conditions and propose two spectral convolution models (GHCN, SHSC) for EDVW-hypergraphs learning. We analyze the over-smoothing issues (Li et al., 2018b) based on their spectral properties deduced from equivalency conditions.

3) Extensive experiments across various domains (social network analysis, visual objective classification and protein modeling) demonstrate the effectiveness of the proposed methods (GHCN and SHSC) for both EIVW-hypergraph and EDVW-hypergraph learning tasks. Notably, we are first to adopt EDVW-hypergraphs for protein structures and obtain significant performance boost.

Due to space limit, more discussions on related works are deferred to Appendix A.

## 2 THE THEORY OF EQUIVALENCY

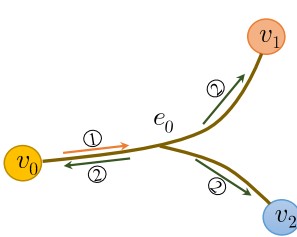

Figure 2: Hypergraph Random Walk. ① denotes the first step and ② represents the second.

**Notations.** $\mathbf{I} \in \mathbb{R}^{n \times n}$ denotes the identity matrix, and $\mathbf{1} \in \mathbb{R}^n$ represents the vector with ones in each component. We use boldface letter $\mathbf{x} \in \mathbb{R}^n$ to indicate an $n$-dimensional vector, where $x(i)$ is the $i^{\text{th}}$ entry of $\mathbf{x}$. We use a boldface capital letter $\mathbf{A} \in \mathbb{R}^{m \times n}$ to denote an $m$ by $n$ matrix and use $A(i, j)$ to denote its $ij^{\text{th}}$ entry.

Let $\mathcal{G}(\mathcal{V}, \mathcal{E}, \omega)$ be a graph with vertex set $\mathcal{V}$, edge set $\mathcal{E}$, and edge weights $\omega$. Let $\mathcal{H}(\mathcal{V}, \mathcal{E}, \mathbf{W}, \mathbf{Q})$ be the hypergraph with vertex set $\mathcal{V}$, edge set $\mathcal{E}$. Let $w(e)$ denote the weight of hyperedge $e$. $\mathbf{W} \in \mathbb{R}^{|\mathcal{E}| \times |\mathcal{E}|}$ is the edge-weights diagonal matrix with entries $W(e, e) = w(e)$. $\mathbf{Q}$ denotes edge-vertex-weights matrix with entries $Q(u, e) = q_e(u) \in \mathbb{R}$ if $u \in e$ and 0 if $u \notin e$. $\mathbf{Q}$ is said to be *edge-independent* if $q_e(u) = q(u) \in \mathbb{R}$ for all $e \ni u$, and called *edge-dependent* otherwise (Chitra & Raphael, 2019). If $q_e(u)$ equals to 1 for all linked $u$ and $e$ ,$\mathbf{Q}$ would reduce to the binary incident matrix $\mathbf{H}$, which is widely used to represent the structure of hypergraph (Zhou et al., 2006; Carletti et al., 2021). The matrix $\mathbf{Q}$ can also be viewed as a weighted incident matrix. Note that a graph is a special case of a hypergraph with hyperedge degree $|e| = 2$. We say a hypergraph is connected when its unweighted clique graph (Chitra & Raphael, 2019) is

connected and we assume hypergraphs are connected. Throughout this paper, equivalency refers to equivalency between hypergraphs and undigraphs if not otherwise specified.

## 2.1 Unified Random Walk and Generalized Hypergraphs

Traditionally, hypergraph random walk is defined by a *two-step* manner (Zhou et al., 2006; Ducournau & Bretto, 2014) (Fig. 2). Then, Chitra & Raphael (2019) raise a new random walk involving the edge-dependent vertex weights into the second step and build the equivalency between EDVW-hypergraphs and digraphs. However, due to the limitation of their hypergraph definition, they fail to answer whether the EDVW-hypergraphs can be equal to undigraphs. So in this part, we integrate the existing two-step random walk methods (Chitra & Raphael, 2019; Carletti et al., 2020; 2021) to obtain a comprehensive *unified random walk framework* with the vertex weights added into the first step, based on which we further define a generalized hypergraph.

**Definition 1 (Unified Random Walk on Hypergraphs)** *The unified random walk on a hypergraph $\mathcal{H}$ is defined in a two-step manner: Given the current vertex $u$,*

***Step I****: choose an arbitrary hyperedge $e$ incident to $u$, with the probability*

$$p_1 = \frac{w(e) \sum_{v \in \mathcal{V}} Q_2(v, e) \rho(\sum_{v \in \mathcal{V}} Q_2(v, e)) Q_1(u, e)}{\sum_{e \in \mathcal{E}} w(e) \sum_{v \in \mathcal{V}} Q_2(v, e) \rho(\sum_{v \in \mathcal{V}} Q_2(v, e)) Q_1(u, e)}; \tag{1}$$

***Step II****: choose an arbitrary vertex $v$ from $e$, with the probability*

$$p_2 = \frac{Q_2(v, e)}{\sum_{v \in \mathcal{V}} Q_2(v, e)}, \tag{2}$$

*where $w(e)$ denotes the hyperedge weight; $Q_1(u, e)$ denotes the contribution of vertex $u$ to hyperedge $e$ in the first step while $Q_2(v, e)$ represents the contribution of vertex $v$ to hyperedge $e$ in the second step; and $\rho(\cdot)$ is a real-valued function, for example, a power function $(\cdot)^\sigma$.*

Suppose $\delta(e) := \sum_{v \in \mathcal{V}} Q_2(v, e)$ is the degree of hyperedge, and $d(v) := \sum_{e \in \mathcal{E}} w(e) \delta(e) \rho(\delta(e)) Q_1(v, e)$ is the degree of vertex $v$. The transition probability of our unified random walk on a hypergraph from vertex $u$ to vertex $v$ is:

$$P(u, v) = \sum_{e \in \mathcal{E}} \frac{w(e) Q_1(u, e) Q_2(v, e) \rho(\delta(e))}{d(u)}. \tag{3}$$

Here, $P(u, v)$ can be written in a $|\mathcal{V}| \times |\mathcal{V}|$ matrix form: $\mathbf{P} = \mathbf{D}_v^{-1} \mathbf{Q}_1 \mathbf{W} \rho(\mathbf{D}_e) \mathbf{Q}_2^\top$, where $\mathbf{D}_v$, $\mathbf{D}_e$ are diagonal matrices with entries $D_v(v, v) = d(v)$ and $D_e(e, e) = \delta(e)$, respectively. $\rho(\mathbf{D}_e)$ represents the function $\rho$ acting on each element of $\mathbf{D}_e$. Notably, the $\rho$ does not always work properly. We present the failure conditions in the Appendix G.4. More detailed intuition of our purpose to design $P(u, v)$ can be found in Appendix B.2.

Under our framework, the existing hypergraph random walk (Zhou et al., 2006; Chitra & Raphael, 2019; Carletti et al., 2020; 2021) can be seen as special cases with specific $p_1$, $p_2$ (see Appendix B.3). The definition of our framework is lazy since it allows self-loops ($P(v, v) > 0$). The non-lazy version is provided in Appendix C. Based on the framework, we define the generalized hypergraph for constructing equivalency between EDVW-hypergraphs and undigraphs as follows.

**Definition 2 (Generalized hypergraph)** *A generalized hypergraph is a hypergraph associated with the unified random walk in Definition 1, denoted as $\mathcal{H}(\mathcal{V}, \mathcal{E}, \mathbf{W}, \mathbf{Q}_1, \mathbf{Q}_2)$. Here, $\mathbf{Q}_1$, $\mathbf{Q}_2$ are vertex weights matrices, each of which can be edge-independent or edge-dependent.*

## 2.2 The Equivalency Between Hypergraphs and undigraphs

In this part, we provide the condition under which the generalized hypergraph is equivalent to weighted undirected clique graph. By establishing the equivalency, we can view graphs as low-order encoders of EDVW-hypergraphs, so any undigraph technique can be used as a subroutine for hypergraph learning.

The clique graph of $\mathcal{H}(\mathcal{V}, \mathcal{E}, \mathbf{W}, \mathbf{Q}_1, \mathbf{Q}_2)$ is denoted as $\mathcal{G}^C$, which is a unweighted graph with vertices $\mathcal{V}$ and edge set $\{(u, v) : u, v \in e, e \in \mathcal{E}\}$. Actually, $\mathcal{G}^C$ turns all hyperedges into cliques. We first present the definition of the equivalency in connection with Markov chains.

**Definition 3** *(Chitra & Raphael (2019))* *Let* $M_1, M_2$ *be the Markov chains with the same finite state space, and let* $\mathbf{P}^{M_1}$ *and* $\mathbf{P}^{M_2}$ *be their probability transition matrices, respectively.* $M_1, M_2$ *are equivalent if* $\mathbf{P}_{u,v}^{M_1} = \mathbf{P}_{u,v}^{M_2}$ *for all states* $u$ *and* $v$.

The equivalency between digraph and hypergraph is straightforward and we defer to Appendix E. On the other hand, it is easy to understand that any simple undigraph can be constructed as a special generalized hypergraph, while the converse is not necessarily true. We provide a counterexample in Appendix G.4. This result implies that undigraph convolutional networks can not trivially be extended to generalized hypergraphs. So, we give one of the implicit equations for formulating the equivalency problem to explore the explicit equivalency conditions.

**Lemma 1** *Let* $\mathcal{H}(\mathcal{V}, \mathcal{E}, \mathbf{W}, \mathbf{Q}_1, \mathbf{Q}_2)$ *denote the generalized hypergraph in Definition 2. Let* $\mathcal{F}_{(Q_1, Q_2)}(u, v) := \sum_{e \in \mathcal{E}} w(e)\rho(\delta(e))Q_1(u, e)Q_2(v, e)$ *and* $\mathcal{T}_{(Q)}(u) := \sum_{e \in \mathcal{E}} w(e)\delta(e)\rho(\delta(e))Q(u, e)$. *When* $\mathbf{Q}_1, \mathbf{Q}_2$ *satisfies the following equation*

$$\mathcal{T}_{(Q_2)}(u)\mathcal{T}_{(Q_1)}(v)\mathcal{F}_{(Q_1, Q_2)}(u, v) = \mathcal{T}_{(Q_2)}(v)\mathcal{T}_{(Q_1)}(u)\mathcal{F}_{(Q_1, Q_2)}(v, u), \forall u, v \in \mathcal{V} \qquad (4)$$

*there exists a weighted undirected clique graph* $\mathcal{G}^C$ *such that a random walk on* $\mathcal{H}$ *is equivalent to a random walk on* $\mathcal{G}^C$ *with edge weights* $\omega(u, v) = \mathcal{T}_{(Q_2)}(u)\mathcal{F}_{(Q_1, Q_2)}(u, v)/\mathcal{T}_{(Q_1)}(u)$ *if* $\mathcal{T}_{(Q_1)}(u) \neq 0$ *and 0 otherwise.*

Figure 3: The division of Equivalency problem (undigraph). The blue box indicates condition (1) in Thm. 1 and red box indicates condition (2). The dark blue part is still open yet.

The proof can be found in Appendix G.2. Next, we derive the explicit condition under which random walks on hypergraphs are equivalent to undigraphs.

**Theorem 1 (Equivalency between generalized hypergraph and weighted undigraph)** *Let* $\mathcal{H}(\mathcal{V}, \mathcal{E}, \mathbf{W}, \mathbf{Q}_1, \mathbf{Q}_2)$ *denote the generalized hypergraph in Definition 2. When* $\mathcal{H}$ *satisfies any of the condition bellow:*

   *Condition (1)* $\mathbf{Q}_1$ *and* $\mathbf{Q}_2$ *are both edge-independent;*   *Condition (2)* $\mathbf{Q}_1 = k\mathbf{Q}_2$ $(k \in \mathbb{R})$,

*there exists a weighted undirected clique graph* $\mathcal{G}^C$ *such that a random walk on* $\mathcal{H}$ *is equivalent to a random walk on* $\mathcal{G}^C$.

It is easy to verify that conditions (1) and (2) satisfy equation 4 (proof in Appendix G.3). This theorem brings insights from three aspects: i) Hypergraph foundation. The equivalency itself is a fundamental problem and remains open before this work (Agarwal et al., 2006; Chitra & Raphael, 2019). Our conclusion provides more adaptive and explores more essential conditions thanks to the unity of our random walk and the generalization of hypergraph defined (Figure 3). Notably, Chitra & Raphael (2019) shows that the equivalency condition is: $\mathbf{Q}_1 = \mathbf{H}$ and $\mathbf{Q}_2$ is edge-independent, which can be viewed as a special case of the condition (1), while Theorem 1 implies the equivalency is not only related to the dependent relationships between vertex weights and edges, but also to whether the vertex weights used in the first and second step of the unified random walk are proportional. Furthermore, thanks to the introduced $\mathbf{Q}_1$ in the generalized hypergraph, we can obtain the equivalency between EDVW-hypergraph and undigraph (i.e. condition (2) ), filling an important gap in the equivalency theory of EDVW-hypergraph. ii) Providing a theoretical basis for hypergraph applications. The condition (2), containing both the edge-independent vertex weights and edge-dependent vertex weights cases which match different hypergraph applications (Ding & Yilmaz, 2010; Zeng et al., 2016; Zhang et al., 2018), provides those with theoretical explanations. The condition (2) also reveals that the existing random walks (Carletti et al., 2020; 2021) on hypergraphs are equivalent to undigraph. iii) Hypergraph learning. The equivalent undigraphs can be viewed as the lower-order encoders of hypergraphs. Thus, one can obtain hypergraph representations directly by exploiting the undigraph learning methods. Especially, the condition (2), which implies that an EDVW-hypergraph can be equivalent to an undigraph, gives the theoretical guarantee for learning hypergraphs without losing edge-dependent vertex weights via undigraph neural networks. Next, we provide the spectral theory for the generalized hypergraphs.

### 2.3 Generalized Hypergraph Laplacian and Its Spectral Properties

To build a bridge between spectral convolution and equivalency, we derive the equivalent undirected weighted graph Laplacian based on Lemma 1 and Thm 1. Then the graph Laplacian can be considered as the Laplacian of generalized hypergraph. Starting from the deduced Laplacian, it is direct and convenient to construct the spectral convolution for hypergraphs. Formally, we have

**Corollary 1** *Let $\mathcal{H}(\mathcal{V}, \mathcal{E}, \mathbf{W}, \mathbf{Q}_1, \mathbf{Q}_2)$ be the generalized hypergraph in Definition 2. Let $\hat{\mathbf{D}}_v$ be a $|\mathcal{V}| \times |\mathcal{V}|$ diagonal matrix with entries $\hat{D}_v(v, v) := \hat{d}(v) := \sum_{e \in \mathcal{E}} w(e)\delta(e)\rho(\delta(e))Q_2(v, e)$. No matter $\mathcal{H}$ satisfies condition (1) or condition (2) in Thm. 1, it obtains the unified explicit form of stationary distribution $\pi$ and Laplacian matrix $\mathbf{L}$ as:*

$$\pi = \frac{\mathbf{1}^\top \hat{\mathbf{D}}_v}{\mathbf{1}^\top \hat{\mathbf{D}}_v \mathbf{1}} \ and \ \mathbf{L} = \mathbf{I} - \hat{\mathbf{D}}_v^{-1/2} \mathbf{Q}_2 \mathbf{W} \rho(\mathbf{D}_e) \mathbf{Q}_2^\top \hat{\mathbf{D}}_v^{-1/2}. \tag{5}$$

There are two observations from Corollary 1: (i) This stationary distribution is different from the classical $\pi(v) = d(v)/\sum_v d(v)$ (Zhou et al., 2006), which depends on the vertex degree $d(v)$. Our $\pi(v)$ is related to $\hat{d}(v)$, which means that we cannot trivially extend from classical theory. The detailed proof is deferred to Appendix G.6; (ii) Both $\pi$ and $\mathbf{L}$ are only related to $\mathbf{Q}_2$, independent of $\mathbf{Q}_1$, implying possible information loss of $\mathbf{Q}_1$ under the setting required an asymmetrical two-steps random walk(i.e. $\mathbf{Q}_1 \neq \mathbf{Q}_2$) to construct a hypergraph. Meanwhile, $\mathbf{L}$ can be viewed as a normalization Laplacian led by the equivalent undigraph $\mathcal{G}^C$ with adjacency matrix $\mathbf{Q}_2 \mathbf{W} \rho(\mathbf{D}_e) \mathbf{Q}_2^\top$, based on which we can design the EDVW-hypergraph spectral convolutions.

Corollary 1 implies that no matter $\mathbf{Q}_1, \mathbf{Q}_2$ are both edge-independent or $\mathbf{Q}_1 = \mathbf{Q}_2$, we can directly use the Laplacian matrix $\mathbf{L}$ to analyze the spectral properties. The spectral properties of Laplacian (Chung & Graham, 1997) are vital in the research of graph neural networks to design a stable and effective convolution operator. Therefore, we deduce two important conclusions concerning eigenvalues of $\mathbf{L}$ as the basic theory of Laplacian application under the equivalency conditions in Thm. 1. One claims the eigenvalues range of $\mathbf{L}$ is $[0, 2]$ (details in Appendix G.7), and the other describes the rate of convergence of our unified random walk listed bellow (see Appendix G.8)

**Corollary 2** *Let $\mathcal{H}(\mathcal{V}, \mathcal{E}, \mathbf{W}, \mathbf{Q}_1, \mathbf{Q}_2)$ be the generalized hypergraph in Definition 1. When $\mathcal{H}$ satisfies any of two conditions in Thm. 1, let $\mathbf{L}$ and $\pi$ be the hypergraph Laplacian matrix and stationary distribution from Corollary 1. Let $\lambda_H$ denote the smallest nonzero eigenvalue of $\mathbf{L}$. Assume an initial distribution $\mathbf{f}$ with $f(i) = 1$ ($f(j) = 0, \forall j \neq i$) which means the corresponding walk starts from vertex $v_i$. Let $\mathbf{p}^{(k)} = \mathbf{f}\mathbf{P}^k$ be the probability distribution after $k$ steps unified random walk where $\mathbf{P}$ denotes the transition matrix, then $p^{(k)}(j)$ denotes the probability of finding the walker in vertex $v_j$ after $k$ steps. We have:*

$$\left| p^{(k)}(j) - \pi(j) \right| \leq \sqrt{\frac{\hat{d}(j)}{\hat{d}(i)}} (1 - \lambda_H)^k. \tag{6}$$

For hypergraph learning, Corollary 2 implies that the spectral convolution deduced from the Laplacian in Corollary 1 might suffer from the over-smoothing issue (Zhu & Koniusz, 2021). Appendix F.4 gives a theoretical perspective for analyzing the over-smoothing issue based on this Corollary, which further inspires the design of convolutions in Section 3 to alleviate the issue.

## 3 Two Spectral Convolutions for Hypergraphs

Intuitively, there are two possible routes for designing EDVW-hypergraph neural networks: 1. message passing; 2. spectral-based methods. Message passing has proved to be a powerful tool for extracting information from hypergraphs or graphs. However, on the one hand, researchers usually adopt heuristic ideas to design message-passing models, leading to a lack of theoretical guarantees (Huang & Yang, 2021). This makes it difficult to analyze properties of corresponding neural networks directly, such as over-smoothing issue (Li et al., 2018b). On the other hand, spectral-based methods not only have a solid foundation in graph signal processing (Kipf & Welling, 2017; Feng et al., 2019), but also effectively inspires the design of message passing techniques (Xu et al., 2019).

Here, we design two hypergraph spectral convolutions for hypergraph learning based on equivalency conditions in Thm. 1. As the Laplacian $\mathbf{L}$ enjoys the same formula under any of the two equivalency conditions, algorithms deduce by $\mathbf{L}$ would be adaptive to any generalized hypergraph satisfied Thm. 1, including EIVW-hypergraph and EDVW-hypergraph.

## 3.1 GENERALIZED SPECTRAL HYPERGRAPH CONVOLUTION

We denote $\mathbf{K}$ as $\mathbf{Q}_2 \mathbf{W} \rho(\mathbf{D}_e) \mathbf{Q}_2^\top$ for expression simplicity. Then the unified hypergraph Laplacian matrix $\mathbf{L}$ in Corollary 1 can be expressed as $\mathbf{L} = \mathbf{I} - \hat{\mathbf{D}}_v^{-1/2} \mathbf{K} \hat{\mathbf{D}}_v^{-1/2}$. Similar to Defferrard et al. (2016), we approximate the convolutional kernel with Chebyshev polynomial to avoid eigenvalue decomposition of $\mathbf{L}$. Finally, the following *Generalized Hypergraph Convolutional Network* (**GHCN**) is obtained by introducing the *re-normalization* technique (Kipf & Welling, 2017):

$$\mathbf{X}^{(l+1)} = \psi(\tilde{\mathbf{T}} \mathbf{X}^{(l)} \mathbf{\Theta}), \tag{7}$$

where $\tilde{\mathbf{T}} := \tilde{\mathbf{D}}_v^{-1/2} \tilde{\mathbf{K}} \tilde{\mathbf{D}}_v^{-1/2}$, and $\tilde{\mathbf{K}} = \mathbf{K} + \mathbf{I}$ can be regarded as the weighted adjacency matrix of the equivalent undigraph with self-loops, and $\tilde{D}(v, v) = \sum_{u \in \mathcal{V}} \tilde{K}(v, u)$. $\mathbf{\Theta}$ is a learnable parameter matrix and $\psi(\cdot)$ denotes an activation function. More details of derivation can be found in Appendix F.1. Most of existing hypergraph Laplacians (Zhou et al., 2006; Carletti et al., 2020; 2021) can be viewed as special forms of the Laplacian matrix $\mathbf{L}$ and can be derived the special convolutions of GHCN. The biggest advantage of GHCN is that it can handle both EIVW-hypergraphs and EDVW-hypergrpahs. Besides, the unity of random walk makes it possible to aggregate more fine-grained information. Note that HGNN (Feng et al., 2019) can be viewed as a specific case of GHCN without the re-normalization trick ($\rho(\cdot) = (\cdot)^{-1}$, $\mathbf{Q} = \mathbf{H}$).

However, Corollary 2 indicates that initial signals or features intend to converge to a certain vector, causing vertices hard to distinguish between each other. This implies that GHCN also suffers from over-smoothing issue (Li et al., 2018b), which limits the depth of the convolution model and hinders the aggregation of long-range information. In fact, the analysis also suggests that HGNN (Feng et al., 2019) suffers from the over-smoothing issue. Next, we would propose a new spectral convolution to alleviate this issue.

## 3.2 A SIMPLE SPECTRAL CONVOLUTION BASED ON MARKOV DIFFUSION KERNEL

In order to aggregate long-range information, we propose *the discounted Markov Diffusion process* to aggregate the information of vertices and propose a new convolutional architecture for hypergraphs.

**Discounted Markov Diffusion Kernel for Hypergraphs.** A diffusion process on a generalized hypergraph can be understood as a process of information extracting among vertices. To generate the underlying feature map ($\mathbf{Z}(t)$), we first utilize the transition probabilities of the Markov chain in Definition 1 to design a *discounted average visiting rate*: $\bar{v}_{ik}(t) = \frac{1}{t} \sum_{\tau=1}^{t} \alpha^\tau \Pr(s(\tau) = k | s(0) = i)$, where $t \in \mathbb{Z}^{++}$. Notably, we introduce a discount factor $\alpha \in (0, 1]$ to exponentially weaken the long-step transition probabilities from vertex $i$ to vertex $k$. Specially, we set $\bar{v}_{ik}(0) = 1$ if and only if $i = k$, otherwise $\bar{v}_{ik}(0) = 0$. Assume that $\bar{\mathbf{v}}_i(t) = (\bar{v}_{i1}(t), \cdots, \bar{v}_{i|\mathcal{V}|}(t))^\top$. Then, we adopt the diffusion distance $d_{ij}(t) = \|\bar{\mathbf{v}}_i(t) - \bar{\mathbf{v}}_j(t)\|_2^2 = \|\mathbf{Z}^\top(t)(\bar{\mathbf{v}}_i(0) - \bar{\mathbf{v}}_j(0))\|_2^2$ proposed in Pons & Latapy (2005) to generate our *discounted Markov diffusion kernel* $\mathbf{K}_{MD}(t) = \mathbf{Z}(t)\mathbf{Z}^\top(t)$, where $\mathbf{Z}(t) = \sum_{\tau=1}^{t} \frac{\alpha^\tau}{t} \mathbf{P}^\tau$ can be viewed as the weighted sum of the transition matrix (derivation in Appendix F.2). The intuition behind this diffusion distance is that if two vertices diffuse similarly based on hypergraph random walk, we are supposed to take them similar. Then, the influence of the two vertices to other vertices in the hypergraph is considered in a similar manner.

**Simple Hypergraph Spectral Convolution (SHSC).** Inspired by the diffusion kernel-based GCNNs from Zhu & Koniusz (2021), based on our discounted Markov diffusion kernel, we propose a *Simple Hypergraph Spectral Convolution* (**SHSC**) as follows:

$$\mathbf{Y} = \psi\left(\left(\beta \sum_{k=1}^{K} \frac{\alpha^k}{K} \tilde{\mathbf{T}}^k + (1-\beta)\mathbf{I}\right) \mathbf{X} \mathbf{\Theta}\right). \tag{8}$$

Recall $\tilde{\mathbf{T}}$ is the transition matrix we used in equation 7, $\psi(\cdot)$ is the activation function to map the node representation of output, and $\Theta$ is the parameter of filter to be learned during training. SHSC is able to balance the global information aggregation and the vertex's own information by introducing a hyperparameter $\beta \in [0,1]$ to control the self-loops $(\tilde{\mathbf{T}})^0 = \mathbf{I}$. Through the discounted Markov Diffusion process, our SHSC could gain stronger local and weaker global information, thereby improving the expressive power in the deep layers. On the other hand, SHSC uses only one linear layer, which greatly improves the computational efficiency and effectively alleviates the overfitting problem. Actually, SHSC is a spatial-based model while we can analyze it from a spectral-based perspective, as SHSC can be viewed as a polynomial filter with specific coefficients (see Appendix F.3). Finally, we study why SHSC can relieve over-smoothing. Specifically, we utilize Corollary 2 to generate a lower-bounded quantity of the defined *over-smoothing energy*, based on which we show that SHSC can relieve the over-smoothing issue compared to multi-layer GHCNs (details in Appendix F.4).

Table 1: Summary of classificaiton accuracy(%) results. We report the average test accuracy and its standard deviation over 10 train-test splits. The number in parentheses corresponds to the number of layers of the model. (OOM: our of memory)

| Dataset | Architecture | Cora (co-authorship) | DBLP (co-authorship) | Cora (co-citation) | Pubmed (co-citation) | Citeseer (co-citation) |
|---|---|---|---|---|---|---|
| MLP | - | 52.02±1.7 | 78.72±0.6 | 52.02±1.7 | 69.86±1.6 | 55.03±1.3 |
| HyperGCN | spectral-based | 60.66±10.8 | 84.82±9.7 | 62.35±9.3 | 68.12±9.7 | 56.94±6.3 |
| HGNN | spectral-based | 69.23±1.6 | 88.55±0.18 | 55.60±1.8 | 46.41±0.7 | 38.98±1.1 |
| HNHN | message-passing | 63.95±2.4 | 84.43±0.3 | 41.59±3.1 | 41.94±4.7 | 33.60±2.1 |
| HGAT | message-passing | 65.42±1.5 | OOM | 52.21±3.5 | 46.28±0.53 | 38.32±1.5 |
| UniGNN | message-passing | 75.30±1.2 | 88.80±0.2 | 70.10±1.4 | 74.40±1.0 | 63.60±1.3 |
| SSGC | spectral-based | 72.04±1.2 | 88.61±0.16 | 68.79±2.1 | 74.49±1.3 | 60.52±1.7 |
| GHCN (ours) | spectral-based | 74.79±0.91 | 89.04±0.19 | 69.45±2.0 | **75.37±1.2** | 62.67±1.2 |
| SHSC (ours) | spectral-based | **76.05±0.75(6)** | **89.17±0.21(16)** | **70.64±1.8 (32)** | 75.08±1.1(4) | **65.14±0.97(32)** |

## 4 EMPIRICAL STUDIES

In this section, we evaluate our proposed methods on four tasks: citation network classification, visual object classification, protein quality assessment (regression), and fold classification. For simplicity, we set $\rho(\cdot)$ to be a power function $(\cdot)^\sigma$ ($\sigma$ is a hyper-parameter) and additional experiments to compare the performance with different $\rho$ is deferred to Appendix I.10. The weight matrix of edges, we set, to be an identity matrix by default in our GHCN and SHSC models. Notably, although GHCN and SHSC are designed for EDVW-hypergraph learning, they work for EIVW-hypergraph as well thanks to the unified Laplacian in Corollary 1. In our experiments, citation network classification belongs to EIVW-hypergraph learning tasks, and visual object classification and protein learning are EDVW-hypergraph learning tasks. For baselines designed for EIVW-hypergraph, we replace $\mathbf{Q}_1, \mathbf{Q}_2$ with $\mathbf{H}$.

Table 2: Classification accuracy (%) on ModelNet40. The *embedding* means the output representation of MVCNN+GVCNN Extractor.

| Methods | input | Accuracy |
|---|---|---|
| MVCNN (Feng et al., 2018) | image | 90.1 |
| PointNet (Qi et al., 2017a) | point | 89.2 |
| PointNet++ (Qi et al., 2017b) | point | 90.1 |
| DGCNN (Wang et al., 2019) | point | 92.2 |
| InterpCNN (Mao et al., 2019) | point | 93.0 |
| SimpleView (Uy et al., 2019) | image | 93.6 |
| pAConv (Xu et al., 2021) | point | 93.9 |
| HGAT (Ding et al., 2020) | embedding | 96.4 |
| UniGNN (Huang & Yang, 2021) | embedding | 96.7 |
| HGNN | embedding | 97.2 |
| GHCN(ours) | embedding | 97.3 |
| SHSC(ours) | embedding | **97.7** |

Due to space limitations, we defer additional experimental results on Over-smoothing Analysis, Hyper-parameter Sensitivity Analysis, and Ablation Analysis to Appendix I.6, I.7, I.9, respectively. The insights are: 1) SHSC can efficiently alleviate over-smoothing; 2) The hyper-parameters $\sigma, \beta, \alpha$ are sensitive at a relatively large range but insensitive in a small range of the performance optimum; 3) The re-normalization trick and edge-dependent vertex weights are both effective. In addition, we provide the computational complexity analysis for our models in Appendix I.11.

### 4.1 CITATION NETWORK CLASSIFICATION

This is a semi-supervised node classification task. The datasets include co-authorship and co-citation datasets: PubMed, Citeseer, Cora (Sen et al., 2008), and DBLP (Rossi & Ahmed, 2015). We adopt the same public datasets (https://github.com/malllabiisc/HyperGCN) and train-test

splits of Yadati et al. (2019). Note these datasets satisfy the condition that $\mathbf{Q}_1 = \mathbf{Q}_2 = \mathbf{H}$. So this experiment can be regarded as a special case of the applications of our models. For baselines, we include Multi-layer perceptron(MLP), HNHN (Dong et al., 2020), HGAT (Ding et al., 2020), UniGNN (Huang & Yang, 2021), and two recent spectral-based hypergraph convolutional neural networks (HGCNNs): HGNN (Feng et al., 2019) and HyperGCN (Yadati et al., 2019).

**Comparison with SOTAs.** As shown in Table 19, the results successfully verify the effectiveness of our models and achieve a new SOTA performance across all five datasets. Notably, both GHCN and SHSC have an

Table 3: Test accuracy on visual object classification. Each model we ran 10 random seeds and report the mean ± standard deviation. BOTH means GVCNN+MVCNN, which represents combining the features or structures to generate multi-modal data.

| Datasets | Feature | Structure | HGNN | UniGNN | HGAT | GHCN(ours) | SHSC(ours) |
|---|---|---|---|---|---|---|---|
| | MVCNN | MVCNN | 80.11±0.38 | 75.25±0.17 | 80.40±0.47 | 81.37±0.63 | **82.56±0.39** |
| NTU | GVCNN | GVCNN | 84.26±0.30 | 84.63±0.21 | 84.45±0.12 | **85.15±0.34** | 83.35±0.30 |
| | BOTH | BOTH | 83.54±0.50 | 84.45±0.40 | 84.05±0.36 | 84.45±0.40 | **85.12±0.25** |
| | MVCNN | MVCNN | 91.28±0.11 | 90.36±0.10 | 91.29±0.15 | 91.99±0.16 | **92.01±0.08** |
| Model-Net40 | GVCNN | GVCNN | 92.53±0.06 | **92.88±0.10** | 92.44±0.11 | 92.66±0.10 | 92.69±0.06 |
| | BOTH | BOTH | 97.15±0.14 | 96.69±0.07 | 96.44±0.15 | 97.28±0.15 | **97.78±0.03** |

average performance improvement of 4.88% and 5.83% over the previous spectral-based methods, respectively. On the other hand, SHSC achieves the best result at a deep layer and gains better performance than GHCN on most datasets, which demonstrates the benefits of deep model and the long-range information in hypergraphs. It's worth noting that our methods gain superior performance on disconnected datasets compared to HGNN. HGNN shows poor performance on disconnected datasets, mainly due to the row in the adjacency matrix of equivalent undigraph corresponding to an isolated point is 0, resulting direct loss of its vertex information (an example can see Fig. 4 in appendix). And our methods utilize the renormalization trick, which can maintain the features of isolated vertices during aggregation. Another important observation is that HyperGCN has a high standard deviation, revealing its poor generalization. Conversely, SHSC has lower bias and standard deviation than others, showing better generalization.

## 4.2 Visual Object Classification

This experiment is about semi-supervised learning. We employ two public benchmarks: Princeton ModelNet40 dataset (Wu et al., 2015) and the National Taiwan University (NTU) 3D model dataset (Chen et al., 2003) to evaluate our methods. We follow HGNN (Feng et al., 2019) to preprocess the data by MVCNN (Su et al., 2015) and GVCNN (Feng et al., 2018). Finally, we use the datasets provided by its public Code (https://github.com/iMoonLab/HGNN). More details can be found in Appendix I.4.

**Results.** Table 2 depicts that our methods significantly outperform the image-input or point-input methods. These results demonstrate that our methods can capture the similarity of objects in the feature space to improve the performance of the classification task. Table 20 compares our methods with HGNN and uniGNN on NTU and ModelNet40. From the results, we can see that our methods outperform HGNN on both single modality and multi-modality (BOTH) datasets and SHSC achieves much better performance on multi-modality compared with others. These results reveal that our SHSC has the advantage of combining such multi-modal information through concatenating the weighted incidence matrices ($\mathbf{Q}$) of hypergraphs, which means merging the multi-level hyperedges.

## 4.3 Protein Quality Assessment (QA) and Fold Classification

Protein QA and fold classification are vital for mining the property of proteins. However, to the best of our knowledge, most researchers represented a protein as a sequence or a simple graph (Baldassarre et al., 2020; Hermosilla et al., 2021) in the deep learning area. So we'd like to investigate whether hypergraphs are better than simple graphs for protein modeling, and the following results confirm this conjecture.

**Protein hypergraphs.** A protein is a chain of amino acids (residues) that will fold to a 3D structure. To simultaneously model protein sequence and spatial structure information, we build sequence hyperedges and distance hyperedges: we choose $\tau$ consecutive amino acids $(v_i, v_{i+1}, \cdots, v_{i+\tau})$ to form a sequence hyperedge, and choose amino acids whose spatial Euclidean distance is less than

the threshold $\epsilon > 0$ to form a spatial hyperedge, where $v_i(i = 1, \cdots, |S|)$ represents the $i$-th amino acids in the sequence. Appendix I.5 contains more details.

**Protein Quality Assessment (QA).** Protein QA is used to estimate the quality of computational protein models in terms of divergence from their native structure. It is a regression task to predict how close the decoy is to the unknown, native structure.

Inspired by Baldassarre et al. (2020), we train our models on Global Distance Test Score (Zemla, 2003), which is the global-level score, and the Local Distance Difference Test (Mariani et al., 2013), an amino-acids-level score. The loss function of QA is defined as the Mean Squared Error (MSE): $\mathcal{L}_g = MSE(\mathcal{P}^g_{pred} - \text{GDT\_TS}), \mathcal{L}_l = \sum_{i=1}^{|S|} MSE(\mathcal{P}^l_{pred_i} - \text{LDDT}_i)$ where $\mathcal{P}^g_{pred}$ and $\mathcal{P}^l_{pred}$ denote predicted global and local score, respectively. We jointly learn node and graph embeddings and the losses are weighted as $\mathcal{L}_{total} = \mu\mathcal{L}_l + (1 - \mu)\mathcal{L}_g$, where $\mu$ is a hyper-parameter. We use the data from past years' editions of CASP, including CASP10-13. CASP10,11,13 are used for training and validation, and CASP12 is

Table 4: Comparison of our method to others on protein Quality Assessment task (CASP12). At the residue level, we report *Pearson correlation* across all residues of all decoys of all targets ($R$) and *Pearson correlation* across all residues of per decoys and then average all decoys ($R_{decoy}$) with LDDT scores. At the global level, we report *Pearson correlation* across all decoys of all targets ($R$) and *Pearson correlation* per target and then average over all targets ($R_{target}$) with GDT_TS scores.

| Methods | GDT_TS | | LDDT | |
|---|---|---|---|---|
| | $R$ | $R_{target}$ | $R$ | $R_{decoy}$ |
| Olechnovivc & Venclovas (2017) | - | 0.557 | - | - |
| Zhang & Zhang (2010) | - | 0.313 | - | - |
| Derevyanko et al. (2018) | - | 0.607 | - | - |
| Conover et al. (2019) | 0.651 | 0.439 | - | - |
| HGNN | 0.667 | 0.582 | 0.632 | 0.319 |
| GHCN (ours) | 0.737 | **0.609** | 0.656 | 0.340 |
| SHSC (ours) | **0.760** | 0.554 | **0.678** | **0.449** |

used for testing. **Results.** Table 4 shows that our methods outperform most of SOTAs, which demonstrates our methods can learn protein more efficiently. The outstanding performance of SHSC indicates that it is able to jointly learn better the node and graph level representation. But SHSC trained to predict only global scores obtains $R = 0.712$, which suggests that the local information can help the assessment of the global quality.

**Protein Fold Classification.** This task is critical for studying the relationship between protein structure and function, and protein evolution. We use the well-known SCOPe 1.75 dataset (Hou et al., 2018) and the cross-entropy loss. **Results.** Table 5 depicts that our proposed GHCN outperforms all others. It is manifest in the table that HGCNN-based methods GHCN and HGNN achieve much better performance compared to GCNN-based. However, it is worth noting that our proposed SHSC, does not seem to of-

Table 5: Comparison of our methods to others on fold classification. We report the *mean accuracy* (%) of all proteins.

| Methods | Architecture | #params | Fold | Super. | Fam. |
|---|---|---|---|---|---|
| Hou et al. (2018) | 1D ResNet | 41.7M | 17.0 | 31.0 | 77.0 |
| Rao et al. (2019)* | 1D Transformer | 38.4M | 21.0 | 34.0 | 88.0 |
| Bepler & Berger (2018)* | LSTM | 31.7M | 17.0 | 20.0 | 79.0 |
| Strodthoff et al. (2020)* | LSTM | 22.7M | 14.9 | 21.5 | 83.6 |
| Kipf & Welling (2017) | GCNN | 1.0M | 16.8 | 21.3 | 82.8 |
| Diehl (2019) | GCNN | 1.0 M | 12.9 | 16.3 | 72.5 |
| Gligorijevic et al. (2020)* | LSTM+GCNN | 6.2M | 15.3 | 20.6 | 73.2 |
| Baldassarre et al. (2020) | GCNN | 1.3M | 23.7 | 32.5 | 84.4 |
| HGNN | HGCNN | 2.1M | 24.2 | 34.4 | 90.0 |
| SHSC (ours) | HGCNN | 0.75M | 21.4 | 23.0 | 78.4 |
| GHCN (ours) | HGCNN | 2.1 M | **25.0** | **36.3** | **91.6** |

* Pre-trained unsupervised on 10-31 million protein sequences.

fer significant advantages over GHCN and HGNN. This can be understood in the following lens: SGHC is used to alleviate the over-smoothing problem, which has a large impact on the node-level task, so our SHSC is more suitable for node-level tasks, while for the pure graph-level tasks, SHSC is too simple to offer good performance.

## 5 DISCUSSIONS AND FUTURE WORK.

Despite the good results, there are some limitations which worth further explorations in the future: 1) We just provide sufficient conditions for the equivalency to undigraphs, but the necessary condition is still unclear. 2) With the equivalency between hypergraphs and undigraphs, we have extended the undigraph-based GCNNs to hypergraph learning in this paper. Meanwhile, the equivalency between hypergraphs and digraphs would allow one to extend digraph-based GCNNs to potentially learn richer information in hypergraphs. 3) One could use the proposed unified random walk on hypergraphs to devise new clustering algorithms for hypergraph partitioning. 4) We make every effort to include a thorough experimental study. Further results on other possible datasets and baselines will be added as a future work.

## ETHICS STATEMENT AND BROADER IMPACT

In the work, we proposed a unified random walk framework on hypergraphs, based on which we study the equivalence between hypergraphs and graphs, and show that the unified random walk on EDVW-hypergraphs can not be equivalent to undigraphs without some specific conditions. Meanwhile, we lead to the Laplacian of our framework and analyze the spectral properties of it, especially focusing on the condition equivalent to an undigraph. Furthermore, we develop a Generalized Hypergrpah Convolutional Network and a Simple Hypergraph Spectral Convolution for hypergraph learning, based on the condition equivalent to undigraph.

Our research of the equivalency conditions between hypergraph and graph opens up many more possibilities in designing more comprehensive graph-based methods to solve the hypergraph problem. The experimental tasks used in our work including social networks and academic networks analysis, visual object classification, and protein learning, suggest that GHCN and SHSC can be applied for beneficial purposes, such as Rumour Detection, Crime Identification, Protein Function Prediction, etc.

Meanwhile, we have to be aware of possible negative impacts, such as using our model for telecom fraud on social networks, placing a high degree of trust in our model and obtain an incorrect interpretation of the results, etc. In addition, we should be vigilant about the potential unemployment issue due to the reduced amount of the need of labeling by human beings.

## REPRODUCIBILITY STATEMENT

All the datasets are publicly available as described in the main text. Our efforts to ensure reproducibility include the following aspects: 1) A sampled code is provided in the supplementary material. 2) The proofs and derivations are provided in Appendix  G, F. 3) More details of experimental configurations are provided in Appendix I.

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

# Appendix

CONTENTS

## A    RELATED WORK

### A.1    SIMPLIFIED RELATED WORK

**Deep learning for hypergraphs.** For spectral-based methods, Feng et al. (2019) introduce the first hypergraph spectral convolution architecture HGNN, based on the hypergraph Laplacian proposed by Zhou et al. (2006). Yadati et al. (2019) use a non-linear Laplacian operator (Chan & Liang, 2020) to convert hypergraphs to simple graphs by reducing hyperedges into simple edges with mediators, and take advantage of graph neural networks to learning hypergraphs. Meanwhile, for spatial-based methods. Dong et al. (2020) propose a message-passing method that is divided into two processes from vertices to edges and from edges to vertices. Wendler et al. (2020) utilize signal processing theory to define the convolutional neural network on set functions. Yadati (2020) and Huang & Yang (2021) generalize the current graph *message passing* methods to hypergraphs and obtain several generalized models. Even though hypergraph neural networks have been well studied to process the the edge-independent vertex weights, to the best of our knowledge, we are the first to attempt on edge-dependent vertex weights.

**Hypergraph random walk and spectral theory.** In machine learning applications, Laplacian is an important tool to represent graphs or hypergraphs. While random walk-based graph Laplacian has a well-studied spectral theory, the spectral methods of hypergraphs are surprisingly lagged. A two-step random walk Laplaciain was proposed by Zhou et al. (2006) for the general hypergraph, and was improved by Chitra & Raphael (2019) and Carletti et al. (2020; 2021). On the other hand, Lu & Peng (2011) and Aksoy et al. (2020) define a $s$-th Laplacian through random $s$-walks on hypergraphs. More recently, non-linear Laplacian, as an important tool to represent hypergraphs, has been extended to several settings, including directed hypergraphs (Zhang et al., 2017; Chan et al., 2019), submodular hypergraphs (Li & Milenkovic, 2017), etc.

**Equivalency between hypergraphs and undigraphs.** A fundamental problem in the spectral theoretical study of hypergraphs is the equivalency with undigraphs. Once equipped with equivalency, it is possible to represent hypergraphs with lower-order graphs, thus reducing the difficulty of hypergraph learning. So far, equivalence is established from two perspectives: spectrum of Laplacian (Agarwal et al., 2006; Hayashi et al., 2020) or random walk (Chitra & Raphael, 2019).

**Hypergraph with edge-dependent vertex weights (EDVW-hypergraph)**

EDVW-hypergraphs have been widely adopted in machine learning applications, including 3D object classification (Zhang et al., 2018), image segmentation (Ding & Yilmaz, 2010), e-commerce (Li et al., 2018a), image search (Huang et al., 2010), hypergraph clustering (Hayashi et al., 2020) etc. Unfortunately, to the best of our knowledge, existing learning algorithms for processing EDVW-hypergraphs do not involve deep learning methods.

## A.2 Detailed Related Work

**Deep learning for hypergraphs** Feng et al. (2019) introduce the first hypergraph architecture *hypergraph neural network*(HGNN), based on the hypergraph Laplacian proposed by Zhou et al. (2006), however, a flaw in the theory makes the learning of unconnected networks very inefficient (details can see Appendix D.1). Yadati et al. (2019) use the non-linear Laplacian operators Chan & Liang (2020) to convert hypergraphs to simple graphs by reducing a hyperedge to a subgraph with edge weights related only to its degree, which causes the information loss of hypergraphs and limited the generalization. Jiang et al. (2019) proposed a *dynamic hypergraph neural networks* to update hypergraph structure during training. Wendler et al. (2020) utilize signal processing theory to define the convolutional neural network on set functions. Zhang et al. (2020) combined hypergraph label propagation with deep learning and introduced a *Hypergraph Label Propagation Network* to optimize the hypegraph learning. Yadati (2020) proposed a *generalised-MPNN* on hypergraph which unifies the existing MPNNs Gilmer et al. (2017) on simple graph and also raised a *MPNN-Recursive* framework for recursively-structured data processing, but it performs poorly for other high-order relationships. Huang & Yang (2021) generalized the current graph message passing methods to hypergraphs and obtain a UniGCN with self-loop and a deepen hypergrpah network UniGCNII. However, it does not give reasons why self-loop can assist UniGCN to gain the improvement of performance in the citation network benchmark. Moreover, its experimental results with various depths may show that the HGCNNs have over-smoothing issue, but theoretical analysis to explain this phenomenon is lacked. Hypergraph deep learning has also been applied to many areas, such as NLP (Ding et al., 2020), computer vision (Jin et al., 2019), recommendation system (Zhang et al., 2019), etc

**Hypergraph random walk and spectral theory.** In machine learning applications, Laplacian is an important tool to represent graphs or hypergraphs. While random walk-based graph Laplacian has a well-studied spectral theory, the spectral methods of hypergraphs are surprisingly lagged. The research of hypergraph Laplacian can probably be traced back to Chung (1993), which defines the Laplacian of k-uniform hypergraph (each hyperedge contains the same number of vertices). Then Zhou et al. (2006) defined a two-step random walk-based Laplacian for general hypergraphs. Based on it, many works try to design a more comprehensive random walk on hypergraphs. Chitra & Raphael (2019) designed a random walk which considered edge-dependent vertex weights of hypergraphs in the *second step* to replace the edge-independent weights in Zhou et al. (2006). Actually, the edge-dependent vertex weights could model the contribution of vertex $v$ to hyperedge $e$ and were widely used in machine learning such as 3D object classification (Zhang et al., 2018; Feng et al., 2019) and image segmentation (Ding & Yilmaz, 2010) etc., but without spectral guarantees.

On the other hand, Carletti et al. (2020; 2021) takes another perspective to gain more fine-grained information from a hypergraph by taking into account the degree of hyperedges to measure the importance between vertices in the *first step*. However, a comprehensive random walk should consider the above two aspects at the same time. Furthermore, different from the two-step random walk Laplacian of Zhou et al. (2006), Lu & Peng (2011) and Aksoy et al. (2020) define a $s$-th Laplacian through random $s$-walk on hypergraphs. More recently, non-linear Laplacian, as an important tool to represent hypergraphs, has been extended to several settings, including directed hypergraphs (Zhang et al., 2017; Chan et al., 2019), submodular hypergraph (Li & Milenkovic, 2017) etc. Meanwhile, liner Laplacian has also been developed. For example, Chitra & Raphael (2019) use a two-step random walk to develop a spectral theory for hypergraphs with edge-dependent vertex weights, and a two-step random walk on hypergraph proposed by Carletti et al. (2020; 2021) also designed a two-step random walk by involving the degree of hyperedges in the first step and lead a linear Laplacian. In this work, we design a unified two-step random walk framework to study the linear Laplacians.

**Equivalency between hypergraphs and undigraphs.** A core problem in the spectral theoretical study of hypergraphs is the equivalency between hypergraphs and graphs. Once equipped with equivalency, it is possible to represent hypergraphs with lower-order graphs, thus reducing the difficulty of hypergraph learning. So far, equivalence can be established from two perspectives: spectrum of Laplacian or random walks. For the aspect of Laplacian spectrum, Agarwal et al. (2006) firstly showed the Laplacian of Zhou et al. (2006) is equivalent to the Laplacian of the corresponding star expansion graph (having the same eigenvalue problem). Next, Hayashi et al. (2020) claims that the Laplacian defined by Chitra & Raphael (2019) is equal to an undigraph Laplacian, but the undi-

graph Laplacian needs to be constructed by invoking the stationary distribution, which restricts its application. Meanwhile, from the random walk perspective, Chitra & Raphael (2019) claims that hypergraph is equivalent to its clique graph (undirected) when the vertex weight of hypergraph is edge-independent. However, despite the edge-dependent weights were widely used (Ding & Yilmaz, 2010; Zeng et al., 2016; Zhang et al., 2018), the equivalency problem remains open when the vertex weight is edge-dependent.

**Protein learning.** Proteins are biological macromolecules with spatial structures formed by folding chains of amino acids, which have an important role in biology. A protein has a three-level structure: primary, secondary, and tertiary, where primary (sequence of amino acids) and tertiary (3D spatial) structures are often used to model proteins for representation learning. Based on amino acids sequence, there exist many related works of protein learning, such as Rao et al. (2019); Bepler & Berger (2018); Strodthoff et al. (2020); Conover et al. (2019), etc. Meanwhile, many 3D structure based models are also raised for protein learning, such as Olechnovivc & Venclovas (2017); Derevyanko et al. (2018); Baldassarre et al. (2020); Diehl (2019); Baldassarre et al. (2020); Gligorijevic et al. (2020); Hermosilla et al. (2021), etc. However, to the best of our knowledge, despite the hypergraphs have been developed to model proteins (Maruyama et al., 2001), there is still no related work to design the EDVW-hypergraph-based protein learning algorithm.

# B    RELATIONS WITH PREVIOUS RANDOM WALKS ON HYPERGRAPH

## B.1    A CLASSICAL RANDOM WALK ON HYPERGRAPH

It can be traced back to Zhou et al. (2006) in which they have given a view of random-walk to analyze the hypergraph normalized cut. The transition probability of Zhou et al. (2006) from current vertex $u$ to next vertex $v$ is denoted as:

$$P(u,v) = \sum_{e \in \mathcal{E}} w(e) \frac{H(u,e)}{d(u)} \frac{H(v,e)}{\delta(e)}. \tag{9}$$

It is easy to find $P(u,v)$ means: (i) choose an arbitrary hyperedge $e$ incident with $u$ with probability $w(e)/d(u)$ where $d(u) = \sum_{e \in \mathcal{E}} w(e)H(u,e)$; (ii) Then choose an arbitrary vertex $v \in e$ with probability $1/\delta(e)$ where $\delta(e) = \sum_{v \in \mathcal{V}} H(v,e)$, which means to select vertex randomly in the hyperedge.

## B.2    INTUITION AND ANALYSIS OF THE UNIFIED RANDOM WALK ON HYPERGRAPH

In this subsection, we would like to explain our intuition of developing the unified random walk framework in Definition 1 from a view of two-step random walk on a hypergraph. To generalize the notation of hypergraph random walk in the seminal paper Zhou et al. (2006), in this work, we tend to explain the process from another perspective. The fact in Zhou et al. (2006) that

$$d(u) = \sum_{e \in \mathcal{E}} w(e)H(u,e) = \sum_{e \in \mathcal{E}} \sum_{v \in \mathcal{V}} w(e) \frac{H(u,e)H(v,e)}{\delta(e)} = \sum_{v \in \mathcal{V}} \sum_{e \in E(u,v)} \frac{w(e)}{\delta(e)} \tag{10}$$

and

$$P(u,v) = \sum_{e \in \mathcal{E}} \frac{w(e)H(u,e)}{d(u)} \cdot \frac{H(v,e)}{\delta(e)} = \frac{1}{d(u)} \sum_{e \in E(u,v)} \frac{w(e)}{\delta(e)} = \frac{\sum_{e \in E(u,v)} \frac{w(e)}{\delta(e)}}{\sum_{b \in \mathcal{V}} \sum_{e \in E(u,b)} \frac{w(e)}{\delta(e)}} \tag{11}$$

show that the probability $P(u,v)$ also means a normalized weighted adjacency relationship between $u$ and $v$ (Here, $E(u,v)$ denotes the hyperedges contained vertices $u$ and $v$.). Furthermore, the relationship is actually measured by the sum of $\frac{w(e)}{\delta(e)}$ for all $e \in \mathcal{E}$ concluding pair $\{u,v\}$, which demonstrates that a hyperedge $e$ with larger degree ($\delta(e)$) linked to $\{u,v\}$ contribute less to the transition probability between $u$ and $v$.

From Carletti et al. (2021), which proposed a random walk on hypergraphs (a special case of our unified random walk with $\mathbf{Q}_1 = \mathbf{Q}_2 = \mathbf{H}$ and $\rho(\cdot) = (\cdot)^\sigma$), we further explain the intuition of our

purpose to design the $P(u, v)$ of our unified random walk. When $\mathbf{Q}_1 = \mathbf{Q}_2 = \mathbf{H}$, $P(u, v)$ in Eq. 3 can be expressed as:

$$P(u, v) = \frac{\sum\limits_{e \in E(u,v)} w(e)\rho(\delta(e))}{\sum\limits_{b \in \mathcal{V}} \sum\limits_{e \in E(u,b)} w(e)\rho(\delta(e))}$$

That is to say, the influence of hyperedge degree should not be limited to an inverse relationship by introducing the function $\rho(\cdot)$. Thus, by evolving more fine-grained vertexes weights $Q_1(u, e)$ and $Q_2(v, e)$ to replace $H(v, e)$ and $H(u, e)$, we obtain the final transition probability of our unified hypergraph random walk in equation 3.

### B.3 SPECIAL CASES OF THE UNIFIED RANDOM WALK FRAMEWORK AND LAPLACIAN

We show the existing random walks and Laplacians are the special cases of our unified random walks and our unified Laplacians, respectively. Under our framework, the existing random walk models can be seen as a special case with specific $p_1$, $p_2$. It is easy to construct the relations of our framework to previous works as follows:

**Remark 1 (Zhou et al. (2006))** *When we choose $\rho(\cdot) = (\cdot)^{-1}$ and $\mathbf{Q}_1 = \mathbf{Q}_2 = \mathbf{H}$ in our framework, the probabilities of unified random walk on hypergraph are reformulated as $p_1 = \frac{w(e)H(u,e)}{\sum_{e \in \mathcal{E}} w(e)H(u,e)}$ and $p_2 = \frac{H(v,e)}{\sum_{v \in \mathcal{V}} H(v,e)}$. As the special case of ours satisfying $\mathbf{Q}_1 = \mathbf{Q}_2 = \mathbf{H}$ and $\rho(\cdot) = (\cdot)^{-1}$, Zhou et al. (2006)'s random walk satisfies both condition (1) and condition (2) in Thm. 1. So we obtain from Corollary 1 that $\pi_{zhou}(v) = \frac{d(v)}{\sum_{u \in \mathcal{V}} d(u)}$ and $\mathbf{L}_{zhou} = \mathbf{I} - \mathbf{D}_v^{-1/2}\mathbf{H}\mathbf{W}\mathbf{D}_e^{-1}\mathbf{H}^{\top}\mathbf{D}_v^{-1/2}$ where $D_v(u, u) = d(u) = \sum_{e \in \mathcal{E}} w(e)H(u, e)$ and $D_e(e, e) = \delta(e) = \sum_{v \in e} H(v, e)$. The forms of $\pi_{zhou}$ and $\mathbf{L}_{zhou}$ are exactly the same in Zhou et al. (2006).*

**Remark 2 (Carletti et al. (2020; 2021))** *When we select $\rho(\cdot) = (\cdot)^{\sigma}$ and $\mathbf{Q}_1 = \mathbf{Q}_2 = \mathbf{H}$, the probabilities of unified random walk on hypergraph are reduced to $p_1 = \frac{w(e)H(u,e)(\sum_{v \in \mathcal{V}} H(v,e))^{\sigma+1}}{\sum_{e \in \mathcal{E}} w(e)H(u,e)(\sum_{v \in \mathcal{V}} H(v,e))^{\sigma+1}}$ and $p_2 = \frac{H(v,e)}{\sum_{v \in \mathcal{V}} H(v,e)}$. As the special case of ours satisfying $\mathbf{Q}_1 = \mathbf{Q}_2 = \mathbf{H}$ and $\rho(\cdot) = (\cdot)^{\sigma}$, Carletti et al. (2021)'s random walk satisfies both condition (1) and condition (2) in Thm. 1. So we obtain from Corollary 1 that $\pi_{car}(v) = \frac{d(v)}{\sum_{u \in \mathcal{V}} d(u)}$ and $\mathbf{L}_{car} = \mathbf{I} - \mathbf{D}_v^{-1/2}\mathbf{H}\mathbf{W}\mathbf{D}_e^{\sigma}\mathbf{H}^{\top}\mathbf{D}_v^{-1/2}$ where $D_v(u, u) = d(u) = \sum_{e \in \mathcal{E}} w(e)\delta(e)^{\sigma+1}H(u, e)$ and $\mathbf{D}_e(e, e) = \delta(e) = \sum_{v \in e} H(v, e)$. The forms of $\pi_{car}$ and $\mathbf{L}_{car}$ are exactly the same in Carletti et al. (2021). Note that the random walk proposed by Carletti et al. (2020) is a special case of Carletti et al. (2021) with $\sigma = 1$, the $\pi$ and $\mathbf{L}$ are easy to obtain from our conclusions.*

**Remark 3 (Chitra & Raphael (2019))** *When we set $\rho(\cdot) = (\cdot)^{-1}$ and $\mathbf{Q}_1 = \mathbf{H}$, the probabilities of unified random walk on hypergraph become $p_1 = \frac{w(e)H(u,e)}{\sum_{e \in \mathcal{E}} w(e)H(u,e)}$ and $p_2 = \frac{Q_2(v,e)}{\sum_{v \in \mathcal{V}} Q_2(v,e)}$ with an edge-dependent vertex weights $\mathbf{Q}_2$ in the second steps. Actually, as the special case of ours satisfying $\mathbf{Q}_1 = \mathbf{H}$ and $\rho(\cdot) = (\cdot)^{-1}$, Chitra & Raphael (2019)'s random walk on hypergraph generates the Laplacian $\mathbf{L}_{chi}$ which could build up a simple relation with our $\mathbf{L}_{rw}$ in equation 20 as*

$$\mathbf{L}_{chi} = \Phi^{1/2}\mathbf{L}_{rw}\Phi^{1/2} = \Phi - \frac{\Phi\mathbf{P} + \mathbf{P}^{\top}\Phi}{2}$$

*which means that $\mathbf{L}_{rw}$ denotes a form of symmetrization on $\mathbf{L}_{chi}$. Further, as the vertex weights is edge-independent in Chitra & Raphael (2019) which means they satisfies condition(1) in Thm. 1, the same $\pi$ as Chitra & Raphael (2019) can be obtained from Corollary 1, i.e. $\pi(v) = \frac{\hat{d}(v)}{\sum_v \hat{d}(v)}$.*

## C UNIFIED NON-LAZY RANDOM WALKS ON HYPERGRAPH

The non-lazy version is nontrivial to factor and analyze. To ease the studies, we first give the following definition and lemma.

**Definition 4 (Reversible Markov chain)** *Let M be a Markov chain with state space $\mathcal{X}$ and transition probabilities $P(u, v)$, for $u, v \in \mathcal{X}$. We say M is reversible if there exists a probability distribution $\pi$ over $\mathcal{X}$ such that*

$$\pi(u)P(u, v) = \pi(v)P(v, u). \tag{12}$$

**Lemma 2 (Chitra & Raphael (2019))** *Let M be an irreducible Markov chain with finite state space $\mathcal{S}$ and transition probabilities $P(u, v)$ for $u, v \in \mathcal{S}$. M is reversible if and only if there exists a weighted, undirected graph $\mathcal{G}$ with vertex set $\mathcal{S}$ such that a random walk on $\mathcal{G}$ and M are equivalent.*

**Proof.** Let $\pi$ be the stationary distribution of $M$ (suppose that is irreducible). Note that $\pi(u) \neq 0$ due to the irreducibility of $M$. Let $\mathcal{G}$ be a graph with vertices $\mathcal{S}$. Different from Chitra & Raphael (2019), we set the edge weights of $\mathcal{G}$ to be

$$\omega(u, v) = c\pi(u)P(u, v), \quad \forall\, u, v \in \mathcal{S} \tag{13}$$

where $c > 0$ is a constant for normalizing $\pi$. With reversibility, $\mathcal{G}$ is well-defined (i.e. $\omega(u, v) = \omega(v, u)$). In a random walk on $\mathcal{G}$, the transition probability from $u$ to $v$ in one time-step is: $\frac{\omega(u,v)}{\sum_{v \in \mathcal{V}} \omega(u,v)} = \frac{c\pi(u)P(u,v)}{\sum_{v \in \mathcal{V}} c\pi(u)P(u,v)} = P(u, v)$, since $\sum_{w \in \mathcal{S}} P(u, w) = 1$. Thus, if $M$ is reversible, recall Definition 3, the stated claim holds. The other direction follows from the fact that a random walk on an undirected graph is always reversible (Aldous & Fill,[70]).

Next, we generalize the non-lazy random walk of Carletti et al. (2021) to our unified non-lazy random walk with vertex weights $\mathbf{Q}_1, \mathbf{Q}_2$.

**Definition 5 (Unified Non-lazy Random Walk on Hypergraphs)** *Suppose $\delta(e) := \sum_{v \in \mathcal{V}} Q_2(v, e)$ is the degree of hyperedge, and $d_{nl}(v) := \sum_{e \in \mathcal{E}} w(e)(\delta(e) - Q_2(v, e))\rho(\delta(e) - Q_2(v, e))Q_1(v, e)$ is the degree of vertex v. The unified non-lazy random walk on a hypergraph $\mathcal{H}$ is defined in a two-step manner: Given the current vertex u,*

**Step I**: *choose a hyperedge e incident to u, with probability*

$$p_1 = \frac{w(e)(\delta(e) - Q_2(u, e))\rho(\delta(e) - Q_2(u, e))Q_1(u, e)}{d_{nl}(u)}; \tag{14}$$

**Step II**: *choose an arbitrary vertex $v \neq u$ from e, with probability*

$$p_2 = \frac{Q_2(v, e)}{\delta(e) - Q_2(u, e)}, \tag{15}$$

Thus, the transition probability of our unified non-lazy random walk from vertex $u$ to $v$ is:

$$P_{nl}(u, v) = \begin{cases} \sum_{e \in \mathcal{E}} \frac{w(e)\rho(\delta(e) - Q_2(u,e))Q_1(u,e)Q_2(v,e)}{d_{nl}(u)} & \text{if } u \neq v, \\ 0 & \text{if } u = v \end{cases} \tag{16}$$

Thus, the transition matrix of unified non-lazy random walk can be expressed as:

$$\mathbf{P}_{nl} = \mathbf{D}_{nl}^{-1}(\mathbf{Q}_1 \odot \rho(\mathbf{1}\mathbf{D}_e - \mathbf{Q}_2))\mathbf{W}\mathbf{Q}_2^\top, \tag{17}$$

where $\mathbf{D}_{nl}, \mathbf{D}_e$ are diagonal matrices with entries $D_{nl}(v, v) = d_{nl}(v)$ and $D_e(e, e) = \delta(e)$, respectively. $\odot$ denotes the Hadamard product and $\mathbf{1}$ denotes a $|\mathcal{V}| \times |\mathcal{E}|$ ones matrix(i.e. all elements equal to 1). $\rho(\mathbf{1}\mathbf{D}_e - \mathbf{Q}_2))$ represents the function $\rho$ acting on each element of $(\mathbf{1}\mathbf{D}_e - \mathbf{Q}_2)$.

Following Chitra & Raphael (2019), the modified version of the clique graph without self-loops is defined below:

**Definition 6 (Chitra & Raphael (2019))** *Let $\mathcal{H}(\mathcal{V}, \mathcal{E}, \mathbf{Q})$ be a hypergraph associated with the unified non-lazy random walk in Definition 5. The clique graph of $\mathcal{H}$ without self-loops, $\mathcal{G}_{nl}^C$, is a weighted, undirected graph with vertex set $\mathcal{V}$, and edges $\mathcal{E}'$ defined by*

$$\mathcal{E}' = \{(v, w) \in \mathcal{V} \times \mathcal{V} : v, w \in e \text{ for some } e \in \mathcal{E}, \text{ and } v \neq w\} \tag{18}$$

Different from the lazy random walk, a unified non-lazy random walk on a hypergraph with condition (1) or condition (2) in Thm. 1 is not guaranteed to satisfy reversibility (see Definition 4). However, if $\mathbf{Q}_1 = \mathbf{Q}_2 = \mathbf{H}$, then reversibility holds, and we obtain the result below.

**Theorem 2** *Let $\mathcal{H}(\mathcal{V}, \mathcal{E}, \mathbf{Q})$ be a hypergraph associated with the unified non-lazy random walk in Definition 5. Let $\mathbf{Q}_1 = \mathbf{Q}_2 = \mathbf{H}$, i.e. $Q_i(v, e) = 1, i \in \{1, 2\}$ for all vertices $v$ incident hyperedges $e$. Then, there exist weights $\omega(u, v)$ on the clique graph without self-loops $\mathcal{G}_{nl}^C$ such that a non-lazy random walk on $\mathcal{H}$ is equivalent to a random walk on $\mathcal{G}_{nl}^C$.*

**Proof.** We first prove that the unified non-lazy random walk on $\mathcal{H}$ is reversible. It is easy to verify the stationary distribution of the unified non-lazy random walk on $\mathcal{H}$ with $\mathbf{Q}_1 = \mathbf{Q}_2 = \mathbf{H}$ is $\pi(v) = \frac{d_{nl}(v)}{\sum_{v \in \mathcal{V}} d_{nl}(v)}$. Let $P(u, v)$ be the probability of going from u to v in a unified non-lazy random walk on $\mathcal{H}$, where $u \neq v$. Then,

$$
\begin{aligned}
\pi(u)P(u, v) &= \frac{d_{nl}(u)}{\sum_{u \in \mathcal{V}} d_{nl}(u)} \sum_{e \in \mathcal{E}} \frac{w(e)\rho(\delta(e) - Q_2(u, e))Q_1(u, e)Q_2(v, e)}{d_{nl}(u)} \\
&= \frac{1}{\sum_{u \in \mathcal{V}} d_{nl}(u)} \sum_{e \in \mathcal{E}} (w(e)\rho(\delta(e) - Q_2(u, e))Q_1(u, e)Q_2(v, e)) \\
&= \frac{1}{\sum_{u \in \mathcal{V}} d_{nl}(u)} \sum_{e \in \mathcal{E}} (w(e)\rho(\delta(e) - H(u, e))H(u, e)H(v, e)) \\
&= \frac{1}{\sum_{u \in \mathcal{V}} d_{nl}(u)} \sum_{e \in \mathcal{E}} (w(e)\rho(\delta(e) - H(v, e))H(v, e)H(u, e)) \\
&= \pi(v)P(v, u)
\end{aligned}
$$

So the unified non-lazy random walk is reversible. Thus, by Lemma 2, there exists a graph $\mathcal{G}$ with vertex set $\mathcal{V}$ and edge weights

$$
\omega(u, v) = d_{nl}(u)P(u, v)
$$

such that a random walk on $\mathcal{G}$ and the unified non-lazy random walk on $\mathcal{H}$ is equivalent. The equivalence of the random walks implies that $P(u, v) > 0$ if and only if $\omega(u, v) > 0$, so it follows that $\mathcal{G}$ is the clique graph of $\mathcal{H}$ without self-loops. $\blacksquare$

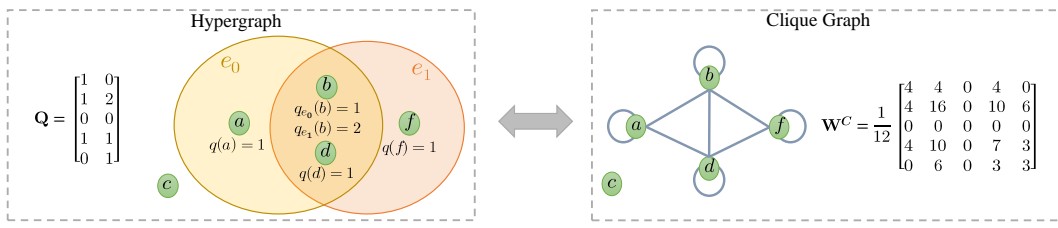

Figure 4: An example of the disconnected hypergraph and its equivalent weighted undirected graph. $c$ is an isolated vertex. The characteristics of hypergraph are encoded in the weight incidence matrix $\mathbf{Q}$. The elements in $\mathbf{W}^C := \mathbf{Q}\mathbf{D}_e^{-1}\mathbf{Q}^\top$ denotes the edge-weight matrix of the clique graph.

# D  TYPICAL HYPERGRAPH CONVOLUTION (HGNN) AND ITS SPECTRAL ANALYSIS

Recall Remark 1, the random walk and Laplaican proposed by Zhou et al. (2006) are special cases ($\rho(\cdot) = (\cdot)^{-1}, \mathbf{Q}_1 = \mathbf{Q}_2 = \mathbf{H}$) of our unified random walk and unified Laplacian, respectively. Thus, now the degree of a vertex $v \in \mathcal{V}$ is $d(v) = \sum_{e \in \mathcal{E}} w(e)H(v, e)$ and for a hyperedge $e \in \mathcal{E}$, its degree is $\delta(e) = \sum_{v \in \mathcal{V}} H(v, e)$. The edge-degree matrix and vertex-degree matrix can be denoted as diagonal matrices $\mathbf{D}_e \in \mathbb{R}^{|\mathcal{E}| \times |\mathcal{E}|}$ and $\mathbf{D}_v \in \mathbb{R}^{|\mathcal{V}| \times |\mathcal{V}|}$ respectively.

## D.1 Typical Spectral Hypergraph Convolution

From the seminal paper Zhou et al. (2006), a classical symmetric normalized hypergraph Laplacian is defined from the perspective of spectral hypergraph partition: $\mathbf{L} = \mathbf{I} - \mathbf{D}_v^{-1/2}\mathbf{H}\mathbf{W}\mathbf{D}_e^{-1}\mathbf{H}^\top\mathbf{D}_v^{-1/2}$. Based on it, Feng et al. (2019) follow Kipf & Welling (2017) to deduce the *Hypergraph Neural Network* HGNN. Specifically, they first perform an eigenvalue decomposition of $\mathbf{L}$: $\mathbf{L} = \mathbf{U}\Lambda\mathbf{U}^\top$, where $\Lambda$ is a diagonal matrix of eigenvalues and $\mathbf{U}$ is consisted of the corresponding regularized eigenvectors. Combining the hypergraph Laplacian with Fourier Transform of graph signal processing, they derive the primary form of hypergraph spectral convolution: $g * x = \mathbf{U}((\mathbf{U}^\top g) \odot (\mathbf{U}^\top x)) = \mathbf{U}g(\Lambda)\mathbf{U}^\top x$ where $g(\Lambda) = diag(g(\lambda_1), g(\lambda_2), \cdots, g(\lambda_N))$ is a function of the eigenvalues of $\mathbf{L}$. To avoid expensive computation of eigen decomposition (Defferrard et al., 2016), they use the truncated Chebyshev polynomials $T_k(x)$ to approximated $g(\Lambda)$, which is recursively deduced by $T_k(x) = 2xT_{k-1}(x) - T_{k-2}(x)$ with $T_0(x) = 1$ and $T_1(x) = x$. $g * x \approx \sum_{k=0}^{K-1} \theta_k T_k(\hat{\Lambda})x$, with scaled Laplacian $\hat{\Lambda} = \frac{2}{\lambda_{max}}\Lambda - \mathbf{I}$ limiting the eigenvalues in $[-1, 1]$ to guarantee requirement of the Chebyshev polynomial (Hammond et al., 2011). Since the parameters in the neural network could adapt to the scaled constant ($\lambda_{max} \approx 2$), the convolution operation is further simplified to be $g * x \approx \theta_0 x - \theta_1 \mathbf{D}_v^{-\frac{1}{2}}\mathbf{H}\mathbf{W}\mathbf{D}_e^{-1}\mathbf{H}^\top\mathbf{D}_v^{-\frac{1}{2}}x$.

However, in order to obtain the final convolutional layer similar to Kipf & Welling (2017) proposed, **Feng et al. (2019) used a specific matrix to fit the scalar parameter $\theta_0$, causing a flaw in the derivation.** In the end, they have managed to obtain a seemingly correct hypergraph spectral convolution (HGNN):

$$\mathbf{Y} = \mathbf{D}^{-1/2}\mathbf{H}\mathbf{W}\mathbf{D}_e^{-1}\mathbf{H}^\top\mathbf{D}_v^{-1/2}\mathbf{X}\Theta, \tag{19}$$

where $\Theta$ is the parameter to be learned during the training process. There is no doubt that HGNN can work properly, but it could have low efficiency when the hypergraph of the datasets is disconnected. Our experiments in subsection 4.1 and Figure 4 verify our conclusion.

## D.2 Spectral Analysis of HGNN

We give the definition of Rayleigh quotient to study the spectrum of Laplacian.

**Lemma 3 ( Rayleigh Quotient (Horn, 1986))** *Assume that $\mathbf{M} \in \mathbb{R}^{n \times n}$ is a real symmetric matrix and $\mathbf{x} \in \mathbb{R}^n$ is an arbitrary nonzero vector. Let $\lambda_1 \le \lambda_2 \le \cdots \le \lambda_n$ denote the eigenvalues of $\mathbf{M}$ in order. Then the Rayleigh quotient satisfies the property:*

$$\lambda_1 \le R(\mathbf{M}, \mathbf{x}) := \frac{\mathbf{x}^\top\mathbf{M}\mathbf{x}}{\mathbf{x}^\top\mathbf{x}} \le \lambda_n.$$

*The equality holds that $R(\mathbf{M}, \mathbf{u}_1) = \lambda_1$ and $R(\mathbf{M}, \mathbf{u}_n) = \lambda_n$ where $\mathbf{u}_i$ is the eigenvector related to $\lambda_i, i \in \{1, n\}$.*

The Lemma 3 is mainly used to get all exact or approximated eigenvalues and eigenvectors of $\mathbf{M}$.

In this part, we utilize it to prove that the smallest eigenvalue $\lambda_{min}$ of $\mathbf{L} = \mathbf{I} - \mathbf{D}_v^{-\frac{1}{2}}\mathbf{H}\mathbf{W}\mathbf{D}_e^{-1}\mathbf{H}^\top\mathbf{D}_v^{-\frac{1}{2}}$ is 0 and figure out the corresponding eigenvector $\mathbf{u}_{min}$ is $\mathbf{D}_v^{\frac{1}{2}}\mathbf{1}$.

Specifically, let $g$ denote an arbitrary column vector which assigns to each vertex $v \in \mathcal{V}$ a real value $g(v)$. We demonstrate $R(\mathbf{L}, g)$ as below in order to describe our conclusion distinctly:

$$\begin{aligned}
\lambda_{min} \le \frac{g^\top\mathbf{L}g}{g^\top g} &= \frac{g^\top(\mathbf{I} - \mathbf{D}_v^{-\frac{1}{2}}\mathbf{H}\mathbf{W}\mathbf{D}_e^{-1}\mathbf{H}^\top\mathbf{D}_v^{-\frac{1}{2}})g}{g^\top g} \\
&= \frac{(\mathbf{D}_v^{\frac{1}{2}}f)^\top(\mathbf{I} - \mathbf{D}_v^{-\frac{1}{2}}\mathbf{H}\mathbf{W}\mathbf{D}_e^{-1}\mathbf{H}^\top\mathbf{D}_v^{-\frac{1}{2}})(\mathbf{D}_v^{\frac{1}{2}}f)}{(\mathbf{D}_v^{\frac{1}{2}}f)^\top(\mathbf{D}_v^{\frac{1}{2}}f)} \\
&= \frac{f^\top(\mathbf{D}_v - \mathbf{H}\mathbf{W}\mathbf{D}_e^{-1}\mathbf{H}^\top)f}{f^\top\mathbf{D}_v f} \\
&= \frac{\sum_e\sum_v\sum_u \frac{w(e)h(v,e)h(u,e)}{\delta(e)}(f(u) - f(v))^2}{2\sum_v f(v)^2 d(v)}
\end{aligned}$$

where $g = \mathbf{D}_v^{1/2} f$ and $f \in \mathbb{R}^{|\mathcal{V}| \times 1}$. It is easy to see the fact: $R(\mathbf{L}, g) \geq 0$, and $R(\mathbf{L}, g) = 0$ if and only if $f = \mathbf{1}$ (i.e. $g = \mathbf{D}_v^{1/2} \mathbf{1}$). Then with lemma 3 we get $\lambda_{min} = \min_g R(\mathbf{L}, g) = R(\mathbf{L}, \mathbf{D}_v^{1/2} \mathbf{1}) = 0$ and $\mathbf{u}_{min} = \mathbf{D}_v^{\frac{1}{2}} \mathbf{1}$.

Actually, since the Laplacian of Zhou et al. (2006) is our unified Laplacian in Corollary 1, the property above can be directly led by Thm. 5.

## E    THE RELATIONSHIP BETWEEN GENERALIZED HYPERGRAPHS AND DIGRAPHS

In this part, we would like to illustrate that our unified random walk on a hypergraph is always equivalent to a random walk on the weighted directed clique graph, As long as the equivalent conditions are established, the graphs can be viewed as the low-order encoders of hypergraphs, and many machine learning methods of digraphs could be easily generalized to hypergraphs.

Chitra & Raphael (2019), whose random walk is a special case of our framework with $\rho(\cdot) = (\cdot)^{-1}, \mathbf{Q}_1 = \mathbf{H}$, claims that the hypergraph with edge-dependent weights (i.e. $\mathbf{Q}_2$ is edge-dependent) is equivalent to a directed graph. Here we extend this conclusion to a more general version by the above definition.

**Theorem 3** *(**Equivalency between hypergraph and weighted digraph**) Let $\mathcal{H}(\mathcal{V}, \mathcal{E}, \mathbf{W}, \mathbf{Q}_1, \mathbf{Q}_2)$ denote the generalized hypergraph in Definition 2. There exists a weighted directed clique graph $\mathcal{G}^C$ such that the random walk on $\mathcal{H}$ is equivalent to a random walk on $\mathcal{G}^C$.*

Thm. 3 indicates that any hypergraph is equivalent to a digraph under our framework (Fig. 1) and implies that the existing hypergraph random walks are modeling pairwise relationships (proof in Appendix G.1 ).

Adopting results of Thm. 3, we design the Laplacian for our unified random walk framework and explore its spectral properties. Thm. 3 says that the unified random walk framework is equivalent to directed graph, so we extend the definition of directed graph Laplacian (Chung, 2005) to obtain the normalized Laplacian matrix as follows.

**Corollary 3** *(**Random Walk-based Hypergraph Laplacian**) Let $\mathcal{H}(\mathcal{V}, \mathcal{E}, \mathbf{W}, \mathbf{Q}_1, \mathbf{Q}_2)$ be the generalized hypergraph Let $\mathbf{P}$ be the transition matrix of our random walk framework on $\mathcal{H}$ with stationary distribution $\pi$ (suppose that exists). Let $\mathbf{\Phi}$ be a $|\mathcal{V}| \times |\mathcal{V}|$ diagonal matrix with $\Phi(v, v) = \pi(v)$. The unified random walk-based hypergraph Laplacian matrix $\mathbf{L}_{rw}$ is defined as:*

$$\mathbf{L}_{rw} = \mathbf{I} - \frac{\mathbf{\Phi}^{1/2} \mathbf{P} \mathbf{\Phi}^{-1/2} + \mathbf{\Phi}^{-1/2} \mathbf{P}^{\top} \mathbf{\Phi}^{1/2}}{2}. \tag{20}$$

Following Chung (2005), it is easy to study many properties based on Rayleigh quotient of $\mathbf{L}_{rw}$, such as positive semi-definite, Cheeger inequality, etc. The Laplacian $\mathbf{L}_{rw}$ has a great expressive power thanks to the comprehensive random walk, which unifies most of existing random walk-based Laplacian variants due to the generalization ability of the framework (see Appendix B.3). Note that the distribution $\pi$ is not easy to obtain and do not admit a unitized expression that limits the application of $\mathbf{L}_{rw}$ in equation 20. In this paper, we focus on the case when hypergraph is equivalent to undigraph (Thm. 1) and deduce the explicit expression of $\mathbf{L}_{rw}$ (Corollary 1).

Based on our equivalency Theorem and Lapalacian, we can design directed graph neural networks for generalized hypergraph learning, which we will do as a future work.

## F    DERIVATION AND ANALYSIS OF GHCN AND SHSC

### F.1    DERIVATION OF GHCN

We define $\mathbf{K}$ as $\mathbf{Q}_2 \mathbf{W} \rho(\mathbf{D}_e) \mathbf{Q}_2^{\top}$. Then the unified hypergraph Laplacian matrix $\mathbf{L}$ in Thm. 1 can be denoted as

$$\mathbf{L} = \mathbf{I} - \hat{\mathbf{D}}_v^{-1/2} \mathbf{K} \hat{\mathbf{D}}_v^{-1/2}, \tag{21}$$

Similar to GCN (Kipf & Welling, 2017) and Appendix D.1, we perform eigenvalue decomposition of $\mathbf{L}$, then obtain the convolution kernel by using Fourier transform, and approximate it with the truncated Chebyshev polynomial to get the convolution form:

$$\mathbf{g} \star \mathbf{x} = \theta_0 \mathbf{x} - \theta_1 \hat{\mathbf{D}}_v^{-1/2} \mathbf{K} \hat{\mathbf{D}}_v^{-1/2} \mathbf{x}. \tag{22}$$

Then we adopt the same method proposed by Kipf & Welling (2017) to set the parameter $\theta = \theta_0 = -\theta_1$ and obtain the following expression:

$$\mathbf{g} \star \mathbf{x} = \theta \left( \mathbf{I} + \hat{\mathbf{D}}_v^{-1/2} \mathbf{K} \hat{\mathbf{D}}_v^{-1/2} \right) \mathbf{x}. \tag{23}$$

According to Theorem 5, it is easy to know the eigenvalues of $\mathbf{I} + \hat{\mathbf{D}}_v^{-1/2} \mathbf{K} \hat{\mathbf{D}}_v^{-1/2}$ range in $[0, 2]$, which may cause the unstable numerical instabilities and gradient explosion problem. So we use the renormalization trick (Kipf & Welling, 2017):

$$\mathbf{I} + \hat{\mathbf{D}}_v^{-1/2} \mathbf{K} \hat{\mathbf{D}}_v^{-1/2} \to \tilde{\mathbf{D}}_v^{-1/2} \tilde{\mathbf{K}} \tilde{\mathbf{D}}_v^{-1/2} \triangleq \tilde{\mathbf{T}}, \tag{24}$$

where $\tilde{\mathbf{K}} = \mathbf{K} + \mathbf{I}$ can be regarded as the weighted adjacency matrix of the equivalent weighted graph with self-loops, and $\tilde{\mathbf{D}}(v, v) = \sum_u \tilde{\mathbf{K}}(v, u)$ is a $|\mathcal{V}| \times |\mathcal{V}|$ diagonal matrix. This self-loops makes the methods more robust to process disconnected hypergraphs datasets. Finally, we drive the generalized hypergraph convolutional network(**GHCN**):

$$\mathbf{X}^{(l+1)} = \psi(\tilde{\mathbf{T}} \mathbf{X}^{(l)} \boldsymbol{\Theta}) \tag{25}$$

$\boldsymbol{\Theta}$ is a learnable parameter and $\psi(\cdot)$ denotes an activation function.

## F.2 DERIVATION OF THE DISCOUNTED MARKOV DIFFUSION KERNEL

**Discounted Markov Diffusion Kernel for hypergraph**    Recall that the *discounted average visiting rate*:

$$\bar{v}_{ik}(t) = \frac{1}{t} \sum_{\tau=1}^{t} \alpha^\tau \Pr(s(\tau) = k | s(0) = i) = \frac{1}{t} \sum_{\tau=1}^{t} \alpha^\tau (\mathbf{P}^\tau)_{ik} = \mathbf{Z}(t)_{ik} \tag{26}$$

i.e. $\mathbf{Z}(t) = \sum_{\tau=1}^{t} \frac{\alpha^\tau}{t} \mathbf{P}^\tau$.
Then, $\bar{\mathbf{v}}_i(t) = (\bar{v}_{i1}(t), \cdots, \bar{v}_{in}(t))^\top$ denotes as:

$$\bar{\mathbf{v}}_i(t) = (\mathbf{e}_i^\top \mathbf{Z}(t))^\top = \mathbf{Z}^\top(t) \bar{\mathbf{v}}_i(0)$$

Recall the definition of $\bar{\mathbf{v}}_i(0)$ in subsection 3.2: $\bar{v}_{ik}(0) = 1$ if and only if $i = k$, otherwise $\bar{v}_{ik}(0) = 0$. Then, the *discounted Markov diffusion kernel* $\mathbf{K}_{MD}$ can be derived from the $l_2$-norm diffusion distance as follows:

$$\begin{aligned} d_{ij}(t) &= \|\bar{\mathbf{v}}_i(t) - \bar{\mathbf{v}}_j(t)\|_2^2 = \left\|\mathbf{Z}^\top(t)(\bar{\mathbf{v}}_i(0) - \bar{\mathbf{v}}_j(0))\right\|_2^2 \\ &= (\bar{\mathbf{v}}_i(0) - \bar{\mathbf{v}}_j(0))^\top \mathbf{Z}(t) \mathbf{Z}^\top(t) (\bar{\mathbf{v}}_i(0) - \bar{\mathbf{v}}_j(0)) \\ &= (\mathbf{e}_i - \mathbf{e}_j)^\mathrm{T} \mathbf{K}_{MD} (\mathbf{e}_i - \mathbf{e}_j) \end{aligned}$$

## F.3 DERIVATION OF SHSC

Inspired by the diffusion kernel-based methods from Zhu & Koniusz (2021) and Klicpera et al. (2019), and based on the discounted Markov diffusion kernel defined in subsection 3.2, the hypergraph convolution designed in our work is shown as below general form:

$$\mathbf{Y} = \psi \left( \left( \sum_{k=1}^{K} \frac{\alpha^k}{K} \tilde{\mathbf{T}}^k \right) \mathbf{X} \boldsymbol{\Theta} \right), \tag{27}$$

where $\tilde{\mathbf{T}}$ is the generalized transition matrix in equation 24, $\boldsymbol{\Theta}$ is the parameter matrix to be learned during training, and $\psi(\cdot)$ denotes an activation function. Since $\tilde{\mathbf{T}}^k$ can be viewed as a $k$-steps transition matrix, in order to weaken the neighbor vertices in long diffusion steps, we introduce

a discounting factor $\frac{\alpha^k}{K}$ for $\tilde{\mathbf{T}}^k$, which can be deduced from the underlying feature map $\mathbf{Z}(t)$ of discounted Markov diffusion kernel defined in subsection 3.2. We use a well-known trick (Zhu & Koniusz, 2021; Chen et al., 2020) to balance the global information aggregation and the vertex's own information by introducing a hyper-parameter $\beta \in [0, 1]$ to control the self-loop $\tilde{\mathbf{T}}^0 = \mathbf{I}$. Finally, we proposed a **Simple Hypergraph Spectral Convolution** (SHSC) as:

$$\mathbf{Y} = \psi \left( \left( \beta \sum_{k=1}^{K} \frac{\alpha^k}{K} \tilde{\mathbf{T}}^k + (1 - \beta)\mathbf{I} \right) \mathbf{X}\mathbf{\Theta} \right) \tag{28}$$

Actually, SHSC is a spatial-based model while we can analyze it from a spectral-based view. Let $\mathbf{L} = \mathbf{I} - \tilde{\mathbf{T}}$ denote the normalized Laplacian matrix. Then the SHSC can be viewed as a special spectral-based polynomial filter through $\sum_{i=0}^{I} \xi_i \mathbf{L}^i = \sum_{k=0}^{K} \theta_k \tilde{\mathbf{T}}^k$ where $\theta_k = \frac{\alpha^k}{K}$. Finally, we can deduce the coefficients of the special polynomial filter as $\xi_i = (-1)^i \sum_{k=i}^{K} \binom{k}{i} \frac{\alpha^k}{K}$ showing the strong relationships between SHSC and spectral-based model. The explicit form of $\xi_i$ can be derived as:

$$\sum_{k=0}^{K} \theta_k \tilde{\mathbf{T}}^k = \sum_{k=0}^{K} \theta_k (\mathbf{I} - \mathbf{L})^k = \sum_{k=0}^{K} \theta_k \sum_{i=0}^{k} \binom{k}{i}(-1)^i \mathbf{L}^i = \sum_{i=0}^{K} \left( \sum_{k=i}^{K} (-1)^i \binom{k}{i} \frac{\alpha^k}{K} \right) \mathbf{L}^i$$

### F.4 OVER-SMOOTHING ANALYSIS OF GHCN AND SHSC

For real-world datasets, we always use non-Euclidean graphs with finite-dimensional features of vertices. It is difficult to use the absolute *diffusion distance* (Masuda et al., 2017) which measures the distance walked from starting vertex in the diffusion process. Therefore, we use the reciprocal of the mean $l_1$-norm error between $t$-step transition probability $\mathbf{P}^t$ and stationary distribution $\pi$ as the measure of over-smoothing in arbitrary starting vertex. Formally, we name the measure by **over-smoothing energy** as follows:

$$e(i, t) = \frac{1}{\frac{1}{N}\|\mathbf{f}(i)\mathbf{P}^t - \pi\|_1}$$

where $i$ denotes the sign of starting vertex $i$ and $t$ denotes the number of diffusion step. As mentioned in Corollary. 2, $\mathbf{f}(i)$ is an initial distribution which means the walker diffuse starting from vertex $i$(i.e. $\mathbf{f}(i)_i = 1$ and $\mathbf{f}(i)_j = 0, \forall j \neq i$ ). It is easy to observe that if the limit of $e(i, t)$ with respect to $t$ converges to infinity, it indicates the model based on $\mathbf{P}$ undergo severe over-smoothing issue. Recall Corollary 2, we get a low-bounded quantity of $e(i, t)$ as $e_{low}(i, t)$:

$$e(i, t) \geq \frac{N}{\sum_{j=1}^{N}(1 - \lambda_H)^t \frac{\sqrt{\hat{d}(j)}}{\sqrt{\hat{d}(i)}}} = \frac{N\sqrt{\hat{d}(i)}}{(1 - \lambda_H)^t \varphi(H)} = e_{low}(i, t)$$

where $\varphi(H) = \sum_{j=1}^{N} \sqrt{\hat{d}(j)}$ can be seen as a constant associated with $\mathcal{H}$ which is independent with $i$ or $t$. Recall Corollary 2 that $\lambda_H$ is smallest nonzero eigenvalue of $\mathbf{L}$.

**Over-smoothing analysis of GHCN.** As shown in Table 14, we can see our proposed GHCN model also face with the over-smoothing phenomenon. Assume that we get a $K$-layers GHCN model with the form: $\mathbf{X}^{(l+1)} = \text{ReLU}(\tilde{\mathbf{T}}\mathbf{X}^{(l)}\mathbf{\Theta})$. To get a simple mathematical form from above incremental convolution formula (remove the activation layer), we could get approximated $K$-layers GHCN as:

$$\widetilde{\mathbf{X}}^{(K)} = \tilde{\mathbf{T}}^K \mathbf{X}^{(0)} \mathbf{\Theta}$$

Intuitively, from the simple form we find the *over-smoothing energy* of GHCN in vertex $i$ can be represented by $e_{low}(i, K)$. It is easy to analyze that as $K \to \infty$, $e_{low}(i, K)$ converge to infinity which implies that the *over-smoothing energy* $e(i, t)$ converge to infinity. Furthermore, the convergence of *over-smoothing energy* to infinity means the error between initial signal and $\pi$ converge to 0 as the number of layers increasing, which describes the reason of over-smoothing underwent by GHCN from a quantitative perspective. HGNN (Feng et al., 2019) also suffers from over-smoothing as shown in Table 14. Meanwhile, the theoretical analysis of over-smoothing on HGNNcan be covered by the analysis of GHCN due to HGNN is a special case of GHCN.

**Over-smoothing analysis of SHSC.** Similarly, from the form of SHSC, we use $\frac{1}{K}\sum_{k=0}^{K}\alpha^k e_{low}(i,k)$ to represent SHSC's *over-smoothing energy*. In order to explain the reason why SHSC could relieve over-smoothing, we provide the idea of equivalent infinitesimals to compare the over-smoothing energy between muti-layers GHCN and SHSC as follows: As $K \to \infty$:

$$
\begin{aligned}
\lim_{K\to\infty} \frac{e_{SHSC}(i,K)}{e_{GHCN}(i,K)} &= \lim_{K\to\infty} \frac{\frac{1}{K}\sum_{k=0}^{K}\alpha^k e_{low}(i,k)}{e_{low}(i,K)} \\
&= \lim_{K\to\infty} \frac{(1-\lambda_H)^K \sum_{k=0}^{K}(\frac{\alpha}{1-\lambda_H})^k}{K} \\
&= \lim_{K\to\infty} \frac{(1-\lambda_H)^{K+1} - \alpha^{K+1}}{K(1-\lambda_H-\alpha)}(\text{ suppose } 1-\lambda_H-\alpha \neq 0) \\
&= 0
\end{aligned}
\tag{29}
$$

where the last equation depends on $\alpha \in (0,1]$ and $\lambda_H \in (0,2]$ proved by Theorem 5. If $1-\lambda_H-\alpha = 0$, $1-\lambda_H = \alpha < 1$. Thus, the equation 29 also holds. The result describes why SHSC can relieve over-smoothing compared to multi-layer GHCN that the over-smoothing energy of multi-layers GHCN has been sparse by the designed diffusion kernel of SHSC as $K$ growing.

Actually, we can directly analyse the *over-smoothing energy* of SHSC:

$$
\begin{aligned}
&\lim_{K\to\infty} \frac{1}{K}\sum_{k=0}^{K}\alpha^k e_{low}(i,k) \\
&= \lim_{K\to\infty} \frac{1}{K}\sum_{k=0}^{K}\alpha^k \frac{N\sqrt{\hat{d}(i)}}{(1-\lambda_H)^k \varphi(H)} \\
&= \lim_{K\to\infty} \frac{N\sqrt{\hat{d}(i)}}{\varphi(H)}\left(\frac{1-\lambda_H}{K(1-\lambda_H-\alpha)} - \frac{\alpha}{K(1-\lambda_H-\alpha)}\left(\frac{\alpha}{1-\lambda_H}\right)^K\right)
\end{aligned}
\tag{30}
$$

If $\alpha \leq |1-\lambda_H|$ and $1-\lambda_H-\alpha \neq 0$, equation 30 equals to 0; If $1-\lambda_H-\alpha = 0$, equation 30 equals to $\frac{N\sqrt{\hat{d}(i)}}{\varphi(H)}$; Otherwise, equation 30 converges to infinity. Notably, this result shows that the over-smoothing property of SHSC is related to the magnitude of $\alpha$ and $\lambda_H$. Moreover, this analysis further reveals that the introduced hyperparameter $\alpha$ is meaningful for SHSC to alleviate over-smoothing.

## G  DETAILS OF THEORIES AND PROOFS

### G.1  PROOF OF THEOREM 3

**Theorem 3** *(Equivalency between hypergraph and weighted digraph)  Let $\mathcal{H}(\mathcal{V},\mathcal{E},\mathbf{W},\mathbf{Q}_1,\mathbf{Q}_2)$ denote the generalized hypergraph in Definition 2. There exists a weighted directed clique graph $\mathcal{G}^C$ such that the random walk on $\mathcal{H}$ is equivalent to a random walk on $\mathcal{G}^C$.*

**Proof.**  Let $\omega(u,v)$ be the weight of edge $(u,v)$ in $\mathcal{G}^C$. We set $\omega(u,v) = d(u)P(u,v)$ in Definition 1, then the weighted clique graph is a directed graph, where $P(u,v)$ is the transition probabilities in Eq equation 3. The random walk on the directed weighted $\mathcal{G}^C$ with transition probabilities

$$
P^C(u,v) = \frac{\omega(u,v)}{\sum_{v\in\mathcal{V}}\omega(u,v)} = \frac{d(u)P(u,v)}{\sum_{v\in\mathcal{V}}d(u)P(u,v)} = P(u,v)
$$

is equivalent to $\mathcal{H}$ with transition probabilities $\mathbf{P}(u,v)$. The matrix form of edge weight of $\mathcal{G}^C$ is :

$$
\mathbf{W}^C = \mathbf{Q}_1\mathbf{W}\rho(\mathbf{D}_e)\mathbf{Q}_2^\top
$$

Suppose $\mathbf{W} = \mathbf{I}$ and let $\rho(\cdot) = (\cdot)^{-1}$, we can obtain the equivalent weighted clique graph showing in Figure 5.

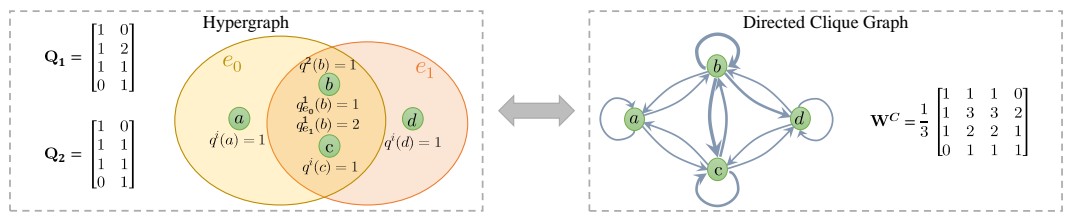

Figure 5: An example of hypergraph and its equivalent weighted directed graph, where $q^i(\cdot) = Q_i(\cdot, e), i \in \{1, 2\}$. Here, $\mathbf{Q}_1$ is edge-dependent and $\mathbf{Q}_2$ is edge-independent. The characteristics of the hypergraph are encoded in the weighted incidence matrices (i.e. vertex weights) $\mathbf{Q}_1, \mathbf{Q}_2$. $\mathbf{W}^C := \mathbf{Q}_1\mathbf{D}_e^{-1}\mathbf{Q}_2^\top$ denotes the edge-weight matrix of the clique graph and can be viewed as the embedding of high-order relationships.

### G.2  PROOF OF LEMMA 1

**Lemma 1** *Let* $\mathcal{H}(\mathcal{V}, \mathcal{E}, \mathbf{W}, \mathbf{Q}_1, \mathbf{Q}_2)$ *denote the generalized hypergraph in Definition 2. Let* $\mathcal{F}_{(Q_1,Q_2)}(u, v) := \sum_{e \in \mathcal{E}} w(e)\rho(\delta(e))Q_1(u, e)Q_2(v, e)$ *and* $\mathcal{T}_{(Q)}(u) := \sum_{e \in \mathcal{E}} w(e)\delta(e)\rho(\delta(e))Q(u, e)$. *When* $\mathbf{Q}_1, \mathbf{Q}_2$ *satisfies the following equation*

$$\mathcal{T}_{(Q_2)}(u)\mathcal{T}_{(Q_1)}(v)\mathcal{F}_{(Q_1,Q_2)}(u, v) = \mathcal{T}_{(Q_2)}(v)\mathcal{T}_{(Q_1)}(u)\mathcal{F}_{(Q_1,Q_2)}(v, u), \forall u, v \in \mathcal{V} \tag{4}$$

*there exists a weighted undirected clique graph* $\mathcal{G}^C$ *such that a random walk on* $\mathcal{H}$ *is equivalent to a random walk on* $\mathcal{G}^C$ *with edge weights* $\omega(u, v) = \mathcal{T}_{(Q_2)}(u)\mathcal{F}_{(Q_1,Q_2)}(u, v)/\mathcal{T}_{(Q_1)}(u)$ *if* $\mathcal{T}_{(Q_1)}(u) \neq 0$ *and 0 otherwise.*

*And the symmetrical Laplacian of* $\mathcal{G}^C$ *is*

$$L(u, v) = \mathcal{I}_{\{u=v\}} - \mathcal{T}_{(Q_2)}(u)^{-1/2}\omega(u, v)\mathcal{T}_{(Q_2)}(v)^{-1/2} \tag{31}$$

*where* $\mathcal{I}_{\{\cdot\}}$ *denotes the indicator function.*

**Proof.**  Form Lemma 2, we just need to prove Markov chain with state space $\mathcal{V}$ is reversible. The transition probabilities of the unified random walk on $\mathbf{H}$ are:

$$P(u, v) = \sum_{e \in \mathcal{E}} \frac{w(e)\rho(\delta(e))Q_1(u, e)Q_2(v, e)}{d(u)} = \frac{\mathcal{F}_{(Q_1,Q_2)}(u, v)}{\mathcal{T}_{(Q_1)}(u)}$$

By equation 4, we have:

$$\mathcal{T}_{(Q_2)}(u)\frac{\mathcal{F}_{(Q_1,Q_2)}(u, v)}{\mathcal{T}_{(Q_1)}(u)} = \mathcal{T}_{(Q_2)}(v)\frac{\mathcal{F}_{(Q_1,Q_2)}(v, u)}{\mathcal{T}_{(Q_1)}(v)},$$

i.e.

$$\mathcal{T}_{(Q_2)}(u)P(u, v) = \mathcal{T}_{(Q_2)}(v)P(v, u),$$

Comparing with  equation 12, we normalize $G(Q_2)$ to define the stationary distribution of $M$:

$$\pi(u) := \frac{\mathcal{T}_{(Q_2)}(u)}{\sum_{u \in \mathcal{V}} \mathcal{T}_{(Q_2)}(u)}, \forall u \in \mathcal{V}, \tag{32}$$

which can be easy to verify by $\sum_u \pi(u)P(u, v) = \pi(v)$. Thus, $\pi(u)P(u, v) = \pi(v)P(v, u)$, which suggests that $M$ is reversible. By Lemma 2, the edge weight $\omega(u, v)$ of $\mathcal{G}^C$ is

$$\omega(u, v) = c\pi(u)P(u, v) = \mathcal{T}_{(Q_2)}(u)P(u, v) = \frac{\mathcal{T}_{(Q_2)}(u)}{\mathcal{T}_{(Q_1)}(u)}\mathcal{F}_{(Q_1,Q_2)}(u, v)$$

To gain the Laplacian of $\mathcal{G}^C$, we calculate the sum of row of edge weights matrix,

$$\sum_{v \in \mathcal{V}} \omega(u, v) = \frac{\mathcal{T}_{(Q_2)}(u)}{\mathcal{T}_{(Q_1)}(u)} \sum_{v \in \mathcal{V}} \mathcal{F}_{(Q_1,Q_2)}(u, v) = \mathcal{T}_{(Q_2)}(u)$$

Then we get the symmetrical Laplacian of $\mathcal{G}^C$:

$$L(u, v) = \mathcal{I}_{\{u=v\}} - \mathcal{T}_{(Q_2)}(u)^{-\frac{1}{2}}\omega(u, v)\mathcal{T}_{(Q_2)}(v)^{-\frac{1}{2}} \tag{33}$$

### G.3 PROOF OF THEOREM 1

**Theorem 1 (Equivalency between generalized hypergraph and weighted undigraph)** *Let $\mathcal{H}(\mathcal{V}, \mathcal{E}, \mathbf{W}, \mathbf{Q}_1, \mathbf{Q}_2)$ denote the generalized hypergraph in Definition 2. When $\mathcal{H}$ satisfies any of the condition bellow:*

*Condition (1) $\mathbf{Q}_1$ and $\mathbf{Q}_2$ are both edge-independent;     Condition (2) $\mathbf{Q}_1 = k\mathbf{Q}_2$ ($k \in \mathbb{R}$),*

*there exists a weighted undirected clique graph $\mathcal{G}^C$ such that a random walk on $\mathcal{H}$ is equivalent to a random walk on $\mathcal{G}^C$.*

**Proof.**

**1) Proof based on Lemma 1**     For condition (1),

$$
\begin{aligned}
\mathcal{F}_{(Q_1, Q_2)}(u, v) &= \sum_{e \in \mathcal{E}} w(e) \rho(\delta(e)) Q_1(u, e) Q_2(v, e) = q_1(u) q_2(v) \sum_{e \in \mathcal{E}} w(e) \rho(\delta(e)) H(u, e) H(v, e) \\
\mathcal{T}_{(Q_i)}(u) &= \sum_{e \in \mathcal{E}} w(e) \delta(e) \rho(\delta(e)) Q_i(u, e) = q_i(u) \sum_{e \in \mathcal{E}} w(e) \delta(e) \rho(\delta(e)) H(u, e), \ i = \{1, 2\}
\end{aligned}
\tag{34}
$$

Substituting the above equations into equation 4, then equation 4 holds.

For condition (2),

$$
\begin{aligned}
\mathcal{F}_{(Q_1, Q_2)}(u, v) &= \sum_{e \in \mathcal{E}} w(e) \rho(\delta(e)) Q_1(u, e) Q_2(v, e) = k \sum_{e \in \mathcal{E}} w(e) \rho(\delta(e)) Q_2(u, e) Q_2(v, e) \\
\mathcal{T}_{(Q_1)}(u) &= \sum_{e \in \mathcal{E}} w(e) \delta(e) \rho(\delta(e)) Q_1(u, e) = k \sum_{e \in \mathcal{E}} w(e) \delta(e) \rho(\delta(e)) Q_2(u, e) = k\mathcal{T}_{(Q_2)}(u)
\end{aligned}
\tag{35}
$$

Similarly, substituting the above equations into equation 4, then equation 4 holds.

**2 ) Proof based on Lemma 2**     I) For Condition (1):
Because $\mathbf{Q}_1$ and $\mathbf{Q}_2$ are both edge-independent, we set $Q_1(u, e) = q_1(u)$ and $Q_2(v, e) = q_2(v)$ for all hyperedge $e$. From Lemma 2, the key is to prove the unified hypergraph random walk on $\mathcal{H}$ under condition (1) is reversible (see Definition 4). It is hard to find the explicit form of $\pi$ before we get Corollary 4. Fortunately, by Kolmogorov's criterion, that is equal to prove:

$$
p_{v_1, v_2} p_{v_2, v_3} \cdots p_{v_n, v_1} = p_{v_1, v_n} p_{v_n, v_{n-1}} \cdots p_{v_2, v_1}
$$

for arbitrary subset $\{v_1, \cdots, v_n\} \subseteq \mathcal{V}$. The transition probabilities of the generalized random walk on $\mathbf{H}$ are:

$$
\begin{aligned}
p(u, v) &= \sum_{e \in \mathcal{E}} \frac{w(e) \rho(\delta(e)) Q_1(u, e) Q_2(v, e)}{d(u)} = \frac{q_1(u) q_2(v)}{d(u)} \sum_{e \in \mathcal{E}} w(e) \rho(\delta(e)) H(u, e) H(v, e) \\
&= \frac{q_1(u) q_2(v)}{d(u)} \sum_{e \in E(u, v)} w(e) \rho(\delta(e))
\end{aligned}
$$

where $E(u,v) = \{e \in \mathcal{E} : u \in e, \ v \in e\}$ to be the set of hyperedges incident to both $v$ and $u$. Then we have:

$$p_{v_1,v_2} \cdots p_{v_n,v_1}$$

$$= \left( \frac{q_1(v_1)q_2(v_2)}{d(v_1)} \sum_{e \in E(v_1,v_2)} w(e)\rho(\delta(e)) \right) \cdots \left( \frac{q_1(v_n)q_2(v_1)}{d(v_n)} \sum_{e \in E(v_n,v_1)} w(e)\rho(\delta(e)) \right)$$

$$= \left( \prod_{i=1}^{n} \frac{q_1(v_i)q_2(v_i)}{d(v_i)} \right) \cdot \left( \prod_{i=1}^{n} \sum_{e \in E(v_i,v_{i+1})} w(e)\rho(\delta(e)) \right), \text{ where set } v_{n+1} = v_1$$

$$= \left( \frac{q_1(v_1)q_2(v_n)}{d(v_1)} \sum_{e \in E(v_n,v_1)} w(e)\rho(\delta(e)) \right) \cdots \left( \frac{q_1(v_2)q_2(v_1)}{d(v_2)} \sum_{e \in E(v_1,v_2)} w(e)\rho(\delta(e)) \right)$$

$$= p_{v_1,v_n} \cdots p_{v_2,v_1}$$

So the unified random walk on $\mathcal{H}$ under condition (1) is reversible.

As a matter of fact, we can succinctly obtain the reversibility via the stationary distribution. Specifically, we can verify the stationary distribution is

$$\pi(u) = \frac{\sum_{e \in \mathcal{E}} w(e)\delta(e)\rho(\delta(e))Q_2(u,e)}{\sum_{v \in \mathcal{V}} \sum_{e \in \mathcal{E}} w(e)\delta(e)\rho(\delta(e))Q_2(v,e)} \tag{36}$$

by $\sum_u \pi(u)P(u,v) = \pi(v)$. Then with $\pi(u)P(u,v) = \pi(v)P(v,u)$ holding, we know our unified random walk on $\mathcal{H}$ under condition (1) is reversible. Since $\mathcal{H}$ is connected, the unified random on $\mathcal{H}$ is irreducible. From Lemma.2, We know that the current unified random walk on $\mathcal{H}$ is equivalent to a random walk on an undirected graph $\mathcal{G}$ with vertex set $\mathcal{V}$. By equation 13, the edge weights of $\mathcal{G}$ are:

$$\omega(u,v) = c\pi(u)P(u,v) = \sum_{e \in \mathcal{E}} w(e)\rho(\delta(e))Q_2(u,e)Q_2(v,e) \tag{37}$$

where $c = \sum_{v \in \mathcal{V}} \sum_{e \in \mathcal{E}} w(e)\delta(e)\rho(\delta(e))Q_2(v,e)$. Note that $P(u,v) > 0$ if and only if $\omega(u,v) > 0$, so $\mathcal{G}$ is the clique graph of $\mathcal{H}$.

II) For condition (2) that $\mathbf{Q}_1 = k\mathbf{Q}_2$,
we have $\mathbf{P} = k\mathbf{D}_v^{-1}\mathbf{Q}_2\mathbf{W}\rho(\mathbf{D}_e)\mathbf{Q}_2^{\top}$. Thanks to the underlying symmetry adjacency matrix of $\mathbf{P}$ (i.e. $\mathbf{Q}_2\mathbf{W}\rho(\mathbf{D}_e)\mathbf{Q}_2$ is symmetrical), we have: $\mathbf{1}^{\top}\hat{D}_v\mathbf{P} = \mathbf{1}^{\top}\hat{D}_v^{\top}$. Thus, the stationary distribution is

$$\pi(u) = \frac{\hat{d}(u)}{\sum_v \hat{d}(v)} = \frac{\sum_{e \in \mathcal{E}} w(e)\delta(e)\rho(\delta(e))Q(u,e)}{\sum_{v \in \mathcal{V}} \sum_{e \in \mathcal{E}} w(e)\delta(e)\rho(\delta(e))Q(v,e)} \tag{38}$$

Further, we would prove that the unified random walk on $\mathcal{H}$ under current condition (2) is reversible.

$$\pi(u)p(u,v) = \frac{\hat{d}(u)}{\sum_{v'} \hat{d}(v')} \sum_{e \in \mathcal{E}} \frac{w(e)\rho(\delta(e))Q(u,e)Q(v,e)}{d(u)}$$

$$= \frac{d(v)}{\sum_{v'} d(v')} \sum_{e \in \mathcal{E}} \frac{w(e)\rho(\delta(e))Q(v,e)Q(u,e)}{d(v)} = \pi(v)p(v,u)$$

So the unified hypergraph random walk on $\mathcal{H}$ under condition (2) is reversible. Since $\mathcal{H}$ is connected, the unified random on $\mathcal{H}$ is irreducible. From Lemma.2, We know that the current unified random walk is equivalent to a random walk on a weighted, undirected graph $\mathcal{G}$ with vertex set $\mathcal{V}$. By equation 13, the edge weights of $\mathcal{G}$ are:

$$\omega(u,v) = c\pi(u)P(u,v) = \sum_{e \in \mathcal{E}} w(e)\rho(\delta(e))Q(u,e)Q(v,e) \tag{39}$$

where $c = \sum_{v'} d(v') = \sum_{v \in \mathcal{V}} \sum_{e \in \mathcal{E}} w(e)\delta(e)\rho(\delta(e))Q(v, e)$. Note that $P(u, v) > 0$ if and only if $\omega(u, v) > 0$, so $\mathcal{G}$ is the clique graph of $\mathcal{H}$.

On the whole, we get the specific equivalent weighted undigraphs under our condition (1) and condition (2). Note that they have the same edge weights expression ( equation 37, equation 39):

$$\mathbf{W}^C = \mathbf{Q}_2 \mathbf{W} \rho(\mathbf{D}_e) \mathbf{Q}_2^\top$$

which further suggests the connection between condition (1) and condition (2) stated in Corollary 4 and Thm. 1.

**Remark:** Let $\rho(\cdot) = (\cdot)^{-1}$ and $\mathbf{W} = \mathbf{I}$, we can get the undigraph shown in Figure 1.

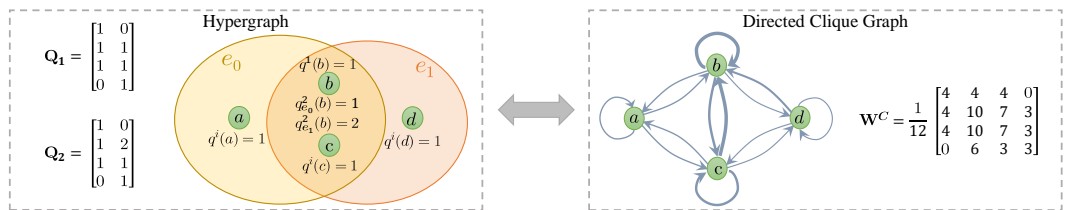

Figure 6: An example of hypergraph and its equivalent weighted directed graph, where $q^i(\cdot) = Q_i(\cdot, e), i \in \{1, 2\}$. Here, $\mathbf{Q}_1$ is edge-independent and $\mathbf{Q}_2$ is edge-dependent. The characteristics of the hypergraph are encoded in the weighted incidence matrices (i.e. vertex weights) $\mathbf{Q}_1, \mathbf{Q}_2$. $\mathbf{W}^C := \mathbf{Q}_1 \mathbf{D}_e^{-1} \mathbf{Q}_2^\top$ denotes the edge-weight matrix of the clique graph and can be viewed as the embedding of high-order relationships.

## G.4 NONEQUIVALENT CONDITION.

**Theorem 4** *There at least exists one generalized hypergraph $\mathcal{H}(\mathcal{V}, \mathcal{E}, \mathbf{W}, \mathbf{Q}_1, \mathbf{Q}_2)$ (Definition 2) with edge-dependent weights and $\mathbf{Q}_1 \neq k\mathbf{Q}_2$, such that a random walk on $\mathcal{H}$ is not equivalent to a random walk on its undirected clique graph $\mathcal{G}^C$ for any choice of edge weights on $\mathcal{G}^C$.*

**Remark.** Theorem 4 tells that the random walk on a hypergraph can not be always equivalent to a random walk on an undigraph, and open up the possibility to construct a hypergraph is not equivalent to an undigraph, which may inspire researcher to further study for higher-order interactions and other insurmountable bottlenecks on hypergraphs.

**Proof.** When the vertex weights are edge-dependent, the reversibility is not always satisfied in our unified random walk on a hypergraph. By Lemma 2, if the unified random walk is not time-reversible, it could not equivalent to random walks on digraphs for any choice of edge weights.

A irreversible example can see Figure 6. The probability transition matrix of the unified random walk on the left hypergraph (with $\mathbf{W} = \mathbf{I}, \rho(\cdot) = (\cdot)^{-1}$) is given below:

$$\mathbf{P} = \mathbf{D}_v^{-1}\mathbf{Q}_1\mathbf{W}\rho(\mathbf{D}_e)\mathbf{Q}_2^\top = \mathbf{D}_v^{-1}\mathbf{Q}_1\mathbf{D}_e^{-1}\mathbf{Q}_2^\top = \begin{bmatrix} \frac{1}{3} & \frac{1}{3} & \frac{1}{3} & 0 \\ \frac{1}{6} & \frac{5}{12} & \frac{7}{24} & \frac{1}{8} \\ \frac{1}{6} & \frac{5}{12} & \frac{7}{24} & \frac{1}{8} \\ 0 & \frac{1}{2} & \frac{1}{3} & \frac{1}{3} \end{bmatrix}, \tag{40}$$

and the stationary distribution of the unified random walk on hypergraph with $\mathbf{P}$ is:

$$\pi = [\frac{3}{17}, \frac{7}{17}, \frac{5}{17}, \frac{2}{17}] \tag{41}$$

We verify $\pi(1)P'(1, 2) = \frac{3}{17} \times \frac{1}{3} \neq \frac{7}{17} \times \frac{1}{6} = \pi(2)P(2, 1)$, which indicates that the Markov chain represented by the unified random walk in Figure 6 with transition probabilities $\mathbf{P}$ is irreversible.

## G.5 FAILURE CONDITION OF FUNCTION $\rho$.

As a matter of fact, the $\rho$ in our random walk framework can not always work and we give its failure condition as follows.

**Proposition 1** *Let $\mathcal{H}(\mathcal{V}, \mathcal{E}, \mathbf{W}, \mathbf{Q})$ associated with the unified random walk in Definition 1. $\rho(\cdot)$ fails to work in our unified random walk if the hyperedge degrees are edge-independent (i.e. $\delta(e) = \delta(e'), \forall e' \in \mathcal{E}$).*

**Proof.**   Given $\delta(e)$ are edge-independent, let us suppose $\delta(e) = k$, then

$$
\begin{aligned}
P(u,v) &= \sum_{e \in \mathcal{E}} \frac{w(e)Q_1(u,e)Q_2(v,e)\rho(\delta(e))}{d(u)} = \sum_{e \in \mathcal{E}} \frac{w(e)Q_1(u,e)Q_2(v,e)\rho(\delta(e))}{\sum_{e \in \mathcal{E}} w(e)\delta(e)\rho(\delta(e))Q_1(v,e)} \\
&= \sum_{e \in \mathcal{E}} \frac{w(e)Q_1(u,e)Q_2(v,e)\rho(k)}{\sum_{e \in \mathcal{E}} w(e)\delta(e)\rho(k)Q_1(v,e)} = \sum_{e \in \mathcal{E}} \frac{\rho(k)w(e)Q_1(u,e)Q_2(v,e)}{\rho(k)\sum_{e \in \mathcal{E}} w(e)\delta(e)Q_1(v,e)} \\
&= \sum_{e \in \mathcal{E}} \frac{w(e)Q_1(u,e)Q_2(v,e)}{\sum_{e \in \mathcal{E}} w(e)\delta(e)Q_1(v,e)}
\end{aligned}
$$

which indicates $P(u,v)$ is independent to $\rho(\cdot)$, i,e. $\rho(\cdot)$ fails to work.

It is obvious that the $k$-uniform hypergraph has edge-independent hyperedge degrees. Formally, we have the following proposition.

**Proposition 2** *Let $\mathcal{H}(\mathcal{V}, \mathcal{E}, \mathbf{W}, \mathbf{Q}_1, \mathbf{Q}_2)$ denote the generalized hypergraph in Definition 2 with $\rho(\cdot) = (\cdot)^\sigma$ and $\mathbf{Q}_1 = \mathbf{Q}_2 = \mathbf{H}$, then $\mathcal{H}$ is a $k$-uniform hypergraph(i.e each hyperedge contains the same number of vertices) if and only if $P(u,v)$ in equation 3 is independent with $\sigma$.*

**Proof.**   For $\mathbf{Q} = \mathbf{H}$, then $\delta(e) = \sum_{v \in e} h(v,e) = k$.
So, for all $u, v \in \mathcal{V}$, we have:

$$
K^{(\sigma)}(u,v) := \sum_{e \in \mathcal{E}} w(e)(\delta(e))^\sigma q_e(u)q_e(v) = k^\sigma \sum_{e \in \mathcal{E}} w(e)H(u,e)H(v,e) \tag{42}
$$

Then we get the entries of the transition matrix $\mathbf{P}^{(\sigma)}$:

$$
P^{(\sigma)}(u,v) := \frac{K^{(\sigma)}(u,v)}{\sum_b K^{(\sigma)}(u,b)} = \frac{k^\sigma \sum_{e \in \mathcal{E}} w(e)H(u,e)H(v,e)}{\sum_{b \in \mathcal{V}} k^\sigma \sum_{e \in \mathcal{E}} w(e)H(u,e)H(b,e)} \tag{43}
$$

$$
= \frac{\sum_{e \in \mathcal{E}} w(e)H(u,e)H(v,e)}{\sum_{b \in \mathcal{V}} \sum_{e \in \mathcal{E}} w(e)H(u,e)H(b,e)} \tag{44}
$$

It is easy to see that $P^{(\sigma)}(u,v)$ is independent with $\sigma$.
The other direction, we find when $P^{(\sigma)}(u,v)$ in is independent with $\sigma$,then:

$$
\frac{\partial P^{(\sigma)}(u,v)}{\partial \sigma} = \frac{\frac{\partial d(u)}{\partial \sigma}K^{(\sigma)}(u,v) - \frac{\partial K^{(\sigma)}(u,v)}{\partial \sigma}d(u)}{d^2(u)} \tag{45}
$$

$$
= \frac{\sum_b \sum_e \sum_{e'} w(e)w(e')h(u,e)h(u,e')h(v,e')h(b,e)\delta(e)^\sigma \delta(e')^\sigma (\ln(\delta(e)) - \ln(\delta(e')))}{d^2(u)} \tag{46}
$$

$$
= 0, \forall\, u, v \in \mathcal{V} \tag{47}
$$

It is equal to $\exists k \in \mathbf{Z}^{++}$ s.t.:

$$
\delta(e) = \delta(e') = k\ for\ all\ e \in \mathcal{E}
$$

.i.e. the hypergraph is k-uniform. With the condition that $\mathbf{Q}_1 = \mathbf{Q}_2 = \mathbf{H}$, our unified random walk on hypergraph reduces to a random walk on hypergraph that leaving only incident relationships between hyperedges and vertices. Actually, the reduced random walk mentioned is a common structure in real-world applications. The proposition is to say when the hypergraph based on the reduced random walk is uniform, the function $\rho(\cdot)$ does not impact the importance between vertices any more.

### G.6 PROOF OF COROLLARY 1.

Before proving the Corollary 1, we further illustrate the fact that Lemma 4 states.

**Lemma 4** *Let $\mathcal{H}(\mathcal{V}, \mathcal{E}, \mathbf{W}, \mathbf{Q}_1, \mathbf{Q}_2)$ be the generalized hypergraph in Definition 2. Given a $\mathcal{H}$ satisfying condition (1) in Thm. 1, there exists a $\mathcal{H}'$ satisfying the condition (2) such that the random walk on $\mathcal{H}$ is equivalent to that on $\mathcal{H}'$.*

**Proof of Lemma 4.** When the condition (2) in Thm. 1 holds (i.e. $\mathbf{Q}_1 = \mathbf{Q}_2$), the transition probabilities of the unified random walk on $\mathcal{H}$ are

$$P^{c2}(u, v) = \frac{\sum_{e \in \mathcal{E}} w(e)\rho(\delta(e))Q_2(u, e)Q_2(v, e)}{\sum_{e \in \mathcal{E}} w(e)\rho(\delta(e)\delta(e)Q_2(u, e)} \tag{48}$$

Meanwhile, if the condition (1) in Thm. 1 holds (i.e. $\mathbf{Q}_1, \mathbf{Q}_2$ are both edge-independent), for all hyperedge $e$, we represent $Q_1(u, e)$, $Q_2(v, e)$ as $q_1(u)$, $q_2(v)$ respectively, and the transition probabilities are

$$P^{c1}(u, v) = \frac{\sum_{e \in \mathcal{E}} w(e)\rho(\delta(e))Q_1(u, e)Q_2(v, e)}{\sum_{e \in \mathcal{E}} w(e)\rho(\delta(e)\delta(e)Q_1(u, e)} = \frac{q_1(u)q_2(v)\sum_{e \in \mathcal{E}} w(e)\rho(\delta(e))H(u, e)H(v, e)}{q_1(u)\sum_{e \in \mathcal{E}} w(e)\rho(\delta(e)\delta(e)H(u, e)}$$
$$= \frac{q_2(u)q_2(v)\sum_{e \in \mathcal{E}} w(e)\rho(\delta(e))H(u, e)H(v, e)}{q_2(u)\sum_{e \in \mathcal{E}} w(e)\rho(\delta(e)\delta(e)H(u, e)} = \frac{\sum_{e \in \mathcal{E}} w(e)\rho(\delta(e))Q_2(u, e)Q_2(v, e)}{\sum_{e \in \mathcal{E}} w(e)\rho(\delta(e)\delta(e)Q_2(u, e)} \tag{49}$$

It can be observed that $P^{c1}(u, v) = P^{c2}(u, v), \forall u, v \in \mathcal{V}$. By definition 3, the Lemma 4 holds.

Next, based on Lemma 4, we prove the following main conclusion.

**Corollary 1** *Let $\mathcal{H}(\mathcal{V}, \mathcal{E}, \mathbf{W}, \mathbf{Q}_1, \mathbf{Q}_2)$ be the generalized hypergraph in Definition 2. Let $\hat{\mathbf{D}}_v$ be a $|\mathcal{V}| \times |\mathcal{V}|$ diagonal matrix with entries $\hat{D}_v(v, v) := \hat{d}(v) := \sum_{e \in \mathcal{E}} w(e)\delta(e)\rho(\delta(e))Q_2(v, e)$. No matter $\mathcal{H}$ satisfies condition (1) or condition (2) in Thm. 1, it obtains the unified explicit form of stationary distribution $\pi$ and Laplacian matrix $\mathbf{L}$ as:*

$$\pi = \frac{\mathbf{1}^\top \hat{\mathbf{D}}_v}{\mathbf{1}^\top \hat{\mathbf{D}}_v \mathbf{1}} \text{ and } \mathbf{L} = \mathbf{I} - \hat{\mathbf{D}}_v^{-1/2} \mathbf{Q}_2 \mathbf{W} \rho(\mathbf{D}_e) \mathbf{Q}_2^\top \hat{\mathbf{D}}_v^{-1/2}. \tag{5}$$

**Proof of Corollary 1.**

**1) Proof based on Lemma 1** When the generalized hypergraph satisfies Theorem 1, we obtain the equation 34 and equation 35 in Appendix G.2. substituting the equations into equation 33 and equation 32, we can obtain the Corollary 1.

**2) Proof based on Lemma 4** We know whether $\mathcal{H}$ satisfying condition (1) or condition (2) in Theorem 1, there exists a weighted undirected clique graph $\mathcal{G}^C$ equivalent to $\mathcal{H}$. Recall Lemma 4, when $\mathcal{H}$ satisfies any of the two conditions, the probability transition matrix $\mathbf{P}$ of $\mathcal{H}$ can be denoted as:

$$P(u, v) = \frac{\sum_{e \in \mathcal{E}} w(e)\rho(\delta(e))Q_2(u, e)Q_2(v, e)}{\sum_{e \in \mathcal{E}} w(e)\rho(\delta(e)\delta(e)Q_2(u, e)}$$

Then, we obtain the unified form of probability transition matrix $\mathbf{P}$ under any of the two conditions:

$$\mathbf{P} = \hat{\mathbf{D}}_v^{-1} \mathbf{Q}_2 \mathbf{W} \rho(\mathbf{D}_e) \mathbf{Q}_2^\top \tag{50}$$

where $\hat{\mathbf{D}}_v$ is a $|\mathcal{V}| \times |\mathcal{V}|$ diagonal matrix with entries $\hat{D}_v(v, v) = \sum_{e \in \mathcal{E}} w(e)\delta(e)\rho(\delta(e))Q_2(v, e)$ Remark the $\mathbf{P}$ is also the probability transition matrix of the equivalent undigraph in Theorem 1. Due to the symmetric adjacency matrix underlying $\mathbf{P}$ (i.e. $\mathbf{Q}_2 \mathbf{W} \rho(\mathbf{D}_e) \mathbf{Q}_2^\top$ is symmetrical), it is easy to derive the unified stationary distribution $\pi$ by $\mathbf{1}^\top \hat{\mathbf{D}}_v \mathbf{P} = \mathbf{1}^\top \hat{\mathbf{D}}_v$. Thus, the stationary distribution is:

$$\pi = \frac{\mathbf{1}^\top \hat{\mathbf{D}}_v}{\mathbf{1}^\top \hat{\mathbf{D}}_v \mathbf{1}} \quad \left( \pi(u) = \frac{\hat{d}(u)}{\sum_v \hat{d}(v)} = \frac{\sum_{e \in \mathcal{E}} w(e)\delta(e)\rho(\delta(e))Q_2(u, e)}{\sum_v \sum_{e \in \mathcal{E}} w(e)\delta(e)\rho(\delta(e))Q_2(v, e)} \right) \tag{51}$$

Substitute $\mathbf{P}$ and $\pi$ into equation 20, we finally get the unified hypergraph Laplacian $\mathbf{L}$ as:

$$\mathbf{L} = \mathbf{I} - \hat{\mathbf{D}}_v^{-1/2} \mathbf{Q}_2 \mathbf{W} \rho(\mathbf{D}_e) \mathbf{Q}_2^\top \hat{\mathbf{D}}_v^{-1/2}$$

**Remark.** The stationary distribution with edge-independent vertex weights in Chitra & Raphael (2019) can be seen as a special case of $\pi$ in equation 51 by setting $\mathbf{Q}_1 = \mathbf{H}, Q_2(u, e) = \gamma(u)$ and $\rho(\cdot) = (\cdot)^{-1}$:

$$\pi^{Ch}(u) = \frac{\gamma(u) \sum_{e \in \mathcal{E}} w(e) H(u, e)}{\sum_v \gamma(v) \sum_{e \in \mathcal{E}} w(e) H(v, e)} = \frac{\gamma(u) d_{Ch}(u)}{\sum_v \gamma(v) d_{Ch}(v)}$$

where $d_{Ch}(u) = \sum_{e \in \mathcal{E}} w(e) h(u, e)$ is the definition of vertex degree in Chitra & Raphael (2019).

### G.7 SPECTRAL RANGE OF LAPLACIAN MATRIX

The following Theorem proves the upper bound for eigenvalues of $\mathbf{L}$ which meets the requirement of Chebyshev polynomials to enable the derivation for our proposed GHCN in Appendix F.1 and leads to Corollary . 2 in the paper.

**Theorem 5** *Let* $\mathbf{L} = \mathbf{I} - \hat{\mathbf{D}}_v^{-1/2} \mathbf{Q}_2 \mathbf{W} \rho(\mathbf{D}_e) \mathbf{Q}_2^\top \hat{\mathbf{D}}_v^{-1/2}$ *denote the unified hypergraph Laplacian matrix in Corollary 1 and* $\lambda_1 \le \lambda_2 \le \cdots \le \lambda_n$ *denote the eigenvalues of* $\mathbf{L}$ *in order.*

1) $\lambda_{min} = \lambda_1 = 0$ *and* $\mathbf{u}_1 = \hat{\mathbf{D}}_v^{\frac{1}{2}} \mathbf{1}$(*the eigenvector associated with* $\lambda_1$ )

2) *For* $k = 2, 3, \cdots, n$*, we have*

$$\lambda_k = \inf_{f \perp \hat{\mathbf{D}}_v^{\frac{1}{2}} S_{k-1}} \frac{\sum_{e,u,v} \beta(e, u, v)(f(u) - f(v))^2}{\sum_v f(v)^2 d(v)},$$

*here,* $S_{k-1}$ *is the subspace spanned by eigenvectors* $\{\mathbf{u}_1, \cdots, \mathbf{u}_{k-1}\}$ *where* $\mathbf{u}_i$ *related to* $\lambda_i$ *and* $\beta(e, u, v) = w(e)\rho(\delta(e))Q_2(v, e)Q_2(u, e)$.

3) $\lambda_{max} = \lambda_n \le 2$

**Proof:** To analyze the generalized Laplacian proposed in our work, we use the Rayleigh quotient mentioned in the Lemma 3 to give out some conclusions about the eigenvalues and eigenvectors of the Laplacian.

1) Deduce the Rayleigh quotient of $\mathbf{L}_2$:

$$\begin{aligned}
\frac{g^\top \mathbf{L} g}{g^\top g} &= \frac{g^\top (\mathbf{I} - \hat{\mathbf{D}}_v^{-\frac{1}{2}} \mathbf{Q}_2 \mathbf{W} \rho(\mathbf{D}_e) \mathbf{Q}_2^\top \hat{\mathbf{D}}_v^{-\frac{1}{2}}) g}{g^\top g} \\
&= \frac{(\hat{\mathbf{D}}_v^{\frac{1}{2}} f)^\top (\mathbf{I} - \hat{\mathbf{D}}_v^{-\frac{1}{2}} \mathbf{Q}_2 \mathbf{W} \rho(\mathbf{D}_e) \mathbf{Q}_2^\top \hat{\mathbf{D}}_v^{-\frac{1}{2}})(\hat{\mathbf{D}}_v^{\frac{1}{2}} f)}{(\hat{\mathbf{D}}_v^{\frac{1}{2}} f)^\top (\hat{\mathbf{D}}_v^{\frac{1}{2}} f)} \\
&= \frac{f^\top (\hat{\mathbf{D}}_v - \mathbf{Q} \mathbf{W} \rho(\mathbf{D}_e) \mathbf{Q}^\top) f}{f^\top \hat{\mathbf{D}}_v f} \\
&= \frac{\sum_u \sum_v \sum_e w(e) \rho(\delta(e)) Q_2(v, e) Q_2(u, e)(f(u) - f(v))^2}{2 \sum_v f(v)^2 \hat{d}(v)}
\end{aligned}$$

where $g = \hat{\mathbf{D}}_v^{1/2} f$ and $f \in \mathbb{R}^{|\mathcal{V}| \times 1}$. It is easy to see the fact: $R(\mathbf{L}, g) \ge 0$ and $R(\mathbf{L}, g) = 0$ if and only if $f = \mathbf{1}$ (i.e. $g = \hat{\mathbf{D}}_v^{1/2} \mathbf{1}$), Then with Lemma 3 we get $\lambda_1 = \lambda_{min} = \min_g R(\mathbf{L}, g) = R(\mathbf{L}, \hat{\mathbf{D}}_v^{1/2} \mathbf{1}) = 0$ and $\mathbf{u}_1 = \hat{\mathbf{D}}_v^{\frac{1}{2}} \mathbf{1}$.

2) From Courant–Fischer theorem (Horn, 1986), we have:

$$\begin{aligned}
\lambda_k &= \inf_{g \perp S_{k-1}} R(\mathbf{L}, g) = \inf_{g \perp S_{k-1}} \frac{g^\top \mathbf{L} g}{g^\top g} \\
&= \inf_{f \perp \hat{\mathbf{D}}_v^{\frac{1}{2}} S_{k-1}} \frac{\sum_u \sum_v \sum_e w(e) \rho(\delta(e)) Q_2(v, e) Q_2(u, e)(f(u) - f(v))^2}{2 \sum_v f(v)^2 \hat{d}(v)}
\end{aligned}$$

3) We have the fact:

$$(f(u) - f(v))^2 \leq 2(f^2(u) + f^2(v))$$

And from Lemma 3, we get:

$$
\begin{aligned}
\lambda_n = \lambda_{max} &= R(\mathbf{L}, \mathbf{u}_{max}) \\
&= \inf_{f \perp \hat{\mathbf{D}}_v^{\frac{1}{2}} S_{n-1}} \frac{\sum_u \sum_v \sum_e w(e)\rho(\delta(e))Q_2(u,e)Q_2(v,e)(f(u)-f(v))^2}{2\sum_v f(v)^2 \hat{d}(v)} \\
&\leq \inf_{f \perp \hat{\mathbf{D}}_v^{\frac{1}{2}} S_{n-1}} \frac{\sum_u \sum_e w(e)\rho(\delta(e))\delta(e)Q_2(u,e)f(u)^2 + \sum_v \sum_e w(e)\rho(\delta(e))\delta(e)Q_2(v,e)f(v)^2}{2\sum_v f(v)^2 \hat{d}(v)} \\
&\leq \inf_{f \perp \hat{\mathbf{D}}_v^{\frac{1}{2}} S_{n-1}} \frac{2\sum_u \hat{d}(u)f^2(u)}{\sum_v \hat{d}(v)f^2(v)} = 2
\end{aligned}
$$

## G.8 PROOF OF COROLLARY 2

**Corollary 2** *Let $\mathcal{H}(\mathcal{V}, \mathcal{E}, \mathbf{W}, \mathbf{Q}_1, \mathbf{Q}_2)$ be the generalized hypergraph in Definition 1. When $\mathcal{H}$ satisfies any of two conditions in Thm. 1, let $\mathbf{L}$ and $\pi$ be the hypergraph Laplacian matrix and stationary distribution from Corollary 1. Let $\lambda_H$ denote the smallest nonzero eigenvalue of $\mathbf{L}$. Assume an initial distribution $\mathbf{f}$ with $f(i) = 1$ ($f(j) = 0, \forall\, j \neq i$) which means the corresponding walk starts from vertex $v_i$. Let $\mathbf{p}^{(k)} = \mathbf{f}\mathbf{P}^k$ be the probability distribution after $k$ steps unified random walk where $\mathbf{P}$ denotes the transition matrix, then $p^{(k)}(j)$ denotes the probability of finding the walker in vertex $v_j$ after $k$ steps. We have:*

$$\left| p^{(k)}(j) - \pi(j) \right| \leq \sqrt{\frac{\hat{d}(j)}{\hat{d}(i)}}(1 - \lambda_H)^k. \tag{6}$$

**Proof:** From equation 50:

$$\mathbf{P} = \hat{\mathbf{D}}_v^{-1}\mathbf{Q}\mathbf{W}\rho(\mathbf{D}_e)\mathbf{Q}^\top = \hat{\mathbf{D}}_v^{-\frac{1}{2}}(\mathbf{I} - \mathbf{L})\hat{\mathbf{D}}_v^{\frac{1}{2}}$$

Then we have:

$$\mathbf{1}^\top \hat{\mathbf{D}}_v \mathbf{P} = \mathbf{1}^\top \hat{\mathbf{D}}_v$$

Then the stationary distribution $\pi \in \mathbb{R}^{1 \times |\mathcal{V}|}$ of $\mathbf{P}$ can be denoted as:

$$\pi = \frac{\mathbf{1}^\top \hat{\mathbf{D}}_v}{c} \tag{52}$$

, where $c = \mathbf{1}^\top \hat{\mathbf{D}}_v \mathbf{1} = \sum_{v \in \mathcal{V}} \hat{d}(v) > 0$ is the sum of $\mathbf{1}^\top \hat{\mathbf{D}}_v$. Let $\lambda_1 \leq \lambda_2 \leq \cdots \leq \lambda_n$ denote the eigenvalues of $\mathbf{L}$ in order. Then we assume that $\mathbf{f}\hat{\mathbf{D}}_v^{-\frac{1}{2}} = \sum_{i=1}^n a_i \tilde{\mathbf{u}}_i^\top \in \mathbb{R}^{1 \times |\mathcal{V}|}$, where $\tilde{\mathbf{u}}_i \in \mathbb{R}^{|\mathcal{V}| \times 1}$ denotes the $l_2$-norm orthonormal eigenvector related with $\lambda_i$. From Thm .5, we know $\tilde{\mathbf{u}}_1 = \frac{\mathbf{u}_1}{\|\mathbf{u}_1\|_2} = \frac{\hat{\mathbf{D}}_v^{\frac{1}{2}}\mathbf{1}}{\sqrt{c}}$. Then $a_1$ can be computed as:

$$a_1 = \frac{\tilde{\mathbf{u}}_1^\top (\mathbf{f}\hat{\mathbf{D}}_v^{-\frac{1}{2}})^\top}{\tilde{\mathbf{u}}_1^\top \cdot \tilde{\mathbf{u}}_1} = \tilde{\mathbf{u}}_1^\top (\mathbf{f}\hat{\mathbf{D}}_v^{-\frac{1}{2}})^\top = \frac{(\hat{\mathbf{D}}_v^{\frac{1}{2}}\mathbf{1})^\top (\mathbf{f}\hat{\mathbf{D}}_v^{-\frac{1}{2}})^\top}{\sqrt{c}} = \frac{1}{\sqrt{c}}$$

with the fact that $\mathbf{f} \in \mathbb{R}^{1\times|\mathcal{V}|}$ is a probability distribution(i.e. $\mathbf{f1} = 1$).
Then,

$$
\begin{aligned}
|p_j(k) - \pi_j| &= \left|(\mathbf{fP}^k)_j - \pi_j\right| = \left|(\mathbf{fP}^k - \frac{\mathbf{1}^\top \hat{\mathbf{D}}_v}{c})_j\right| = \left|(\mathbf{fP}^k - a_1\tilde{\mathbf{u}}_1^\top \hat{\mathbf{D}}_v^{\frac{1}{2}})_j\right| \\
&= \left|(\mathbf{f}\hat{\mathbf{D}}_v^{-\frac{1}{2}}(\mathbf{I}-\mathbf{L})^k \hat{\mathbf{D}}_v^{\frac{1}{2}} - a_1\tilde{\mathbf{u}}_1^\top \hat{\mathbf{D}}_v^{\frac{1}{2}})_j\right| = \left|(\sum_{i=1}^n a_i \tilde{\mathbf{u}}_i^\top (\mathbf{I}-\mathbf{L})^k \hat{\mathbf{D}}_v^{\frac{1}{2}} - a_1\tilde{\mathbf{u}}_1^\top \hat{\mathbf{D}}_v^{\frac{1}{2}})_j\right| \\
&= \left|(\sum_{i=1}^n a_i \tilde{\mathbf{u}}_i^\top (1-\lambda_i)^k \hat{\mathbf{D}}_v^{\frac{1}{2}} - a_1\tilde{\mathbf{u}}_1^\top \hat{\mathbf{D}}_v^{\frac{1}{2}})_j\right| = \left|(\sum_{i=2}^n a_i \tilde{\mathbf{u}}_i^\top (1-\lambda_i)^k \hat{\mathbf{D}}_v^{\frac{1}{2}})_j\right| \\
&\le (1-\lambda_H)^k \sqrt{\hat{d}(j)} \left|(\sum_{i=1}^n a_i \tilde{\mathbf{u}}_i^\top)_j\right| = (1-\lambda_H)^k \sqrt{\hat{d}(j)} \left|(\mathbf{f}\hat{\mathbf{D}}_v^{-1/2})_j\right| \\
&\le (1-\lambda_H)^k \frac{\sqrt{\hat{d}(j)}}{\sqrt{\hat{d}(i)}}
\end{aligned}
$$

## H  GENERALIZED HYPERGRAPH PARTITION

For arbitrary vertex subset $S \subset \mathcal{V}$, $S^c$ denotes the compliment of $S$. We define the hyperedge boundary $\partial S$ as $\partial S := \{e \in E \mid e \cap S \ne \emptyset, e \cap S^c \ne \emptyset\}$. We will generalize the hypergraph partition problem in Zhou et al. (2006) to our unified hypergraph framework. Hypergraph partition is to cut vertex set $\mathcal{V}$ into two parts $S$ and $S^c$. As shown in Theorem 3, we are inspired to adopt the custom of directed graph normalized cut to formulate the hypergraph partition in a mathematical expression. We need to reuse the definition of $\mathrm{vol}(S)$ and $\mathrm{vol}(\partial S)$ for directed graphs as:

$$
\mathrm{vol}(S) = \sum_{u\in S} \pi(u), \quad \mathrm{vol}(\partial S) = \sum_{u\in S}\sum_{v\in S^c} \pi(u)P(u,v)
$$

And the objective function of the hypergraph partition can be expressed as:

$$
\underset{\emptyset \ne S \subset \mathcal{V}}{\arg\min}\, c(S) := \mathrm{vol}\,\partial S \left(\frac{1}{\mathrm{vol}(S)} + \frac{1}{\mathrm{vol}(S^c)}\right). \tag{53}
$$

As we need the explicit form of stationary distribution to formulate this problem, we set the unified random walk on a hypergraph that satisfies with the conditions in Thm. 1. Then we have:

$$
\mathrm{vol}(S) = \frac{\sum_{u\in S} \hat{d}(u)}{\mathrm{vol}(\mathcal{V})},
$$

$$
\mathrm{vol}(\partial S) = \frac{1}{\mathrm{vol}(\mathcal{V})} \sum_{e\in\partial S}\sum_{u\in e\cap S}\sum_{v\in e\cap S^c} w(e)\rho(\delta(e))Q_2(u,e)Q_2(v,e)
$$

$$
= \frac{1}{\mathrm{vol}(\mathcal{V})} \sum_{e\in\partial S} w(e)\rho(\delta(e))m(e\cap S)m(e\cap S^c)
$$

where $m(e\cap S) = \sum_{u\in e\cap S} Q_2(u,e)$ is a measure on the subset of $V$ which explicitly demonstrates the difference between Zhou et al. (2006) (where $m(e \cap S) = \sum_{u\in e\cap S} H(u,e)$) and our work. We considerate more fine-grained information on vertices when we establish a measure on arbitrary subset of $\mathcal{V}$ and involve the degree of hyperedges to impact the corresponding hyperedge weights rather than use the a priori hyperedge weights.

## I  DETAILS OF EXPERIMENTS

Note that our GHCN and SHSC are both based on the Laplacian led by the equivalent condition in Thm .1, which means the vertex weights we use in our experimental datasets should satisfy the condition (1) or condition (2) in Thm. 1.

### I.1 HYPER-PARAMETER STRATEGY

We use grid search strategies to adjust the hyper-parameters of our GHCN and SHSC. The range of hyper-parameters listed in Table 6, Table 7, Table 8, and Table 9.

Table 6: Hyper-parameter search range for citation network classification and visual object classification.

| Methods | Hyper-parameter | Range |
|---|---|---|
| GHCN | $\sigma$ | {-2,-1,-0.5,0,0.5,1,2} |
| | $\gamma$(visual object classification) | {0.1, 0.2, 0.4, 0.5, 0.8,1.0} |
| | Learning rate | {0.001, 0.005, 0.01} |
| | Hidden dimension | {64,128} |
| | Layers | {2,4} |
| | Weight decay | {1e-3,1e-4,5e-4, 1e-5} |
| | Dropout rate | {0.1,0.2,0.3, 0.4, 0.5} |
| | Optimizer | Adam |
| | Epoch | 1000 |
| | Early stopping patience | 100 |
| | GPU | GTX 2080Ti |
| SHSC | $\alpha$ | {1,0.98,0.95,0.93,0.92,0.9,0.85,0.8} |
| | $\beta$ | {1,0.95,0.9,0.85,0.80, 0.75} |
| | $\gamma$(visual object classification) | {0.1, 0.2, 0.4, 0.5, 0.8,1.0} |
| | Learning rate | {0.001, 0.005, 0.01} |
| | Hidden dimension | {64,128} |
| | Layers | {2,4,6,8,16,32,64 } |
| | Weight decay | {1e-3,1e-4, 1e-5} |
| | Dropout rate | {0.1,0.2,0.3, 0.4, 0.5} |
| | Optimizer | Adam |
| | Epochs | 1000 |
| | Early stopping patience | 100 |
| | GPU | GTX 2080Ti |

Table 7: Hyper-parameter search range for GHCN and SHSC model in fold classification.

| Methods | Hyper-parameter | Range | best |
|---|---|---|---|
| GHCN | Amino-acids type input size | {32,64,128} | 64 |
| | Secondary-structure input size | {32,64,128} | 64 |
| | Readout layer input size | {256,512,1024} | 1024 |
| | Layers L | {2,3,4,5,6} | 4 |
| | Batch size | {64,128,256} | 128 |
| | Weight decay | {1e-3,1e-4, 1e-5} | 1e-3 |
| | Learning rate | {0.0001,0.001, 0.005, 0.01} | 0.0001 |
| | Dropout rate | {0.1,0.2,0.3, 0.4, 0.5} | 0.2 |
| | $\sigma$ | {-1} | -1 |
| | $\gamma$ | {0.1,0.2,,0.4,0.5,0.8,1.0} | 0.1 |
| | Optimizer | Adam | - |
| | Epochs | 300 | - |
| | Early stopping patience | 30 | - |
| | GPU | GTX 2080Ti | - |
| SHSC | Amino-acids type input size | {32,64,128} | 64 |
| | Secondary-structure input size | {32,64,128} | 64 |
| | Readout layer input size | {256,512,1024} | 512 |
| | Layers L | {2,4,6,8,12,16,32} | 12 |
| | Batch size | {64,128,256} | 64 |
| | Weight decay | {1e-3,1e-4, 1e-5} | 1e-5 |
| | Learning rate | {0.0001,0.001, 0.005, 0.01} | 0.0001 |
| | Dropout rate | {0.1,0.2,0.3, 0.4, 0.5} | 0.3 |
| | $\sigma$ | {-1} | -1 |
| | $\gamma$ | {0.1,0.2,,0.4,0.5,0.8,1.0} | 0.4 |
| | $\alpha$ | {1,0.97,0.95,0.9,0.85,0.8,0.75,0.7,0.65} | 0.97 |
| | $\beta$ | { 1,0.95,0.90,0.85,0.8 } | 0.85 |
| | Optimizer | Adam | - |
| | Epochs | 300 | - |
| | Early stopping patience | 30 | - |
| | GPU | GTX 2080Ti | - |

Extra parameter settings for Visual Object Classification include $k = 6$.

Table 8: Hyper-parameter search range for GHCN and SHSC model in protein Quality Assessment (CASP10,CASP11,CASP13 for training, CASP12 for testing).

| Methods | Hyper-parameter | Range | best |
|---|---|---|---|
| GHCN | Amino-acids type input size | {128,256,512} | 512 |
| | Secondary-structure input size | {32,64,128} | 64 |
| | Readout layer input size | {256,512,1024} | 1024 |
| | Layers L | {2,3,4,5,6} | 4 |
| | Batch size | {64,128,256} | 64 |
| | Weight decay | {1e-3,1e-4, 1e-5} | 1e-5 |
| | Learning rate | {0.0001,0.001, 0.005, 0.01} | 0.0001 |
| | Dropout rate | {0.1,0.2,0.3, 0.4, 0.5} | 0.5 |
| | $\sigma$ | {-1} | -1 |
| | $\gamma$ | {0.1,0.2,,0.4,0.5,0.8,1.0} | 0.1 |
| | Optimizer | Adam | - |
| | Epochs | 200 | - |
| | Early stopping patience | 20 | - |
| | GPU | Tesla P40 | - |
| SHSC | Amino-acids type input size | {128,256,512} | 512 |
| | Secondary-structure input size | {32,64,128} | 128 |
| | Readout layer input size | {256,512,1024} | 256 |
| | Layers L | {2,4,6,8,12,16,18,32} | 18 |
| | Batch size | {64,128,256} | 128 |
| | Weight decay | {1e-3,1e-4, 1e-5} | 1e-5 |
| | Learning rate | {0.0001,0.001, 0.005, 0.01} | 0.0005 |
| | Dropout rate | {0.1,0.2,0.3, 0.4, 0.5} | 0.3 |
| | $\sigma$ | {-1} | -1 |
| | $\gamma$ | {0.1,0.2,,0.4,0.5,0.8,1.0} | 0.2 |
| | $\alpha$ | {1,0.97,0.95,0.9,0.85,0.8,0.75,0.7,0.65,0.6} | 0.65 |
| | $\beta$ | { 1,0.95,0.90,0.85,0.8 } | 0.95 |
| | Optimizer | Adam | - |
| | Epochs | 200 | - |
| | Early stopping patience | 20 | - |
| | GPU | Tesla P40 | - |

## I.2 BASELINES OF SPECTRAL CONVOLUTIONS

For HyperGCN (Yadati et al., 2019), we reproduce it by public code with the data already split (10 splits). But the result in its original paper is via running 1000 times with randomly split to get the best 100 times, which is not very common. Therefore, despite the HyperGCN is the SOTA method on Table 10, it can not achieve a competitive performance under the 10-splits.

For HGNN (Feng et al., 2019), in addition to the flaw during the derivation (i.e. use a specific matrix to fit the scalar parameter $\theta_0$), when doing the Visual Object Classification experiment in their public code, it also takes Gaussian kernel as the incidence matrix for hypergraph learning, which lacks theoretical guarantee. So in our experiment, we just use the incidence matrix $\mathbf{H}$ which only consists of elements $\{0, 1\}$ as the incidence matrix using in HGNN.

HGNN and HyperGCN we use the same grid search strategy as our method GHCN.

## I.3 CITATION NETWORK CLASSIFICATION

**Datasets.** The datasets we use for citation network classification include co-authorship and co-citation datasets: PubMed, Citeseer, Cora (Sen et al., 2008) and DBLP (Rossi & Ahmed, 2015). We adopt the hypergraph version of those datasets directly from Yadati et al. (2019), where hypergraphs are created on these datasets by assigning each document as a node and each hyperedge represents (a) all documents co-authored by an author in co-authorship dataset and (b) all documents cited together by a document in co-citation dataset. The initial features of each document (vertex) are represented by bag-of-words features. The details about vertices, hyperedges and features are shown in Table 10.

Table 9: Hyper-parameter search range for GHCN model in protein Quality Assessment (CASP10-12 for training, and CASP13 for testing).

| Methods | Hyper-parameter | Range | best |
|---------|-----------------|-------|------|
| GHCN | Amino-acids type input size | {128,256,512} | 128 |
| | Secondary-structure input size | {32,64,128} | 64 |
| | Readout layer input size | {256,512,1024} | 1024 |
| | Layers L | {2,3,4,5,6} | 4 |
| | Batch size | {64,128,256} | 1024 |
| | Weight decay | {1e-3,1e-4, 1e-5} | 1e-5 |
| | Learning rate | {0.0001,0.001, 0.005, 0.01} | 0.001 |
| | Dropout rate | {0.1,0.2,0.3, 0.4, 0.5,0.6,0.7} | 0.7 |
| | $\sigma$ | {-1} | -1 |
| | $\gamma$ | {0.1,0.2,,0.4,0.5,0.8,1.0} | 0.1 |
| | Optimizer | Adam | - |
| | Epochs | 200 | - |
| | Early stopping patience | 20 | - |
| | GPU | Tesla P40 | - |
| SHSC | Amino-acids type input size | {128,256,512} | 128 |
| | Secondary-structure input size | {32,64,128} | 64 |
| | Readout layer input size | {256,512,1024} | 1024 |
| | Layers L | {2,4,6,8,12,16,18,32} | 8 |
| | Batch size | {64,128,256} | 1024 |
| | Weight decay | {1e-3,1e-4, 1e-5} | 1e-5 |
| | Learning rate | {0.0001,0.001, 0.005, 0.01} | 0.001 |
| | Dropout rate | {0.1,0.2,0.3, 0.4, 0.5} | 0.2 |
| | $\sigma$ | {-1} | -1 |
| | $\gamma$ | {0.1,0.2,,0.4,0.5,0.8,1.0} | 0.2 |
| | $\alpha$ | {1,0.97,0.95,0.9,0.85,0.8,0.75,0.7,0.65,0.6} | 0.65 |
| | $\beta$ | { 1,0.95,0.90,0.85,0.8 } | 0.95 |
| | Optimizer | Adam | - |
| | Epochs | 200 | - |
| | Early stopping patience | 20 | - |
| | GPU | Tesla P40 | - |

Table 10: Real-world hypergraph datasets used in our citation network classification task.

| Dataset | # vertices | # Hyperedges | # Features | # Classes | # isolated vertices |
|---------|-----------|--------------|------------|-----------|---------------------|
| **Cora** (co-authorship) | 2708 | 1072 | 1433 | 7 | 320(11.8%) |
| **DBLP** (co-authorship) | 43413 | 22535 | 1425 | 6 | 0 (0.0%) |
| **Pubmed** (co-citation) | 19717 | 7963 | 500 | 3 | 15877 (80.5%) |
| **Cora** (co-citation) | 2708 | 1579 | 1433 | 7 | 1274 (47.0%) |
| **Citeseer** (co-citation) | 3312 | 1079 | 3703 | 6 | 1854 (55.9%) |

**Settings and baselines.** We adopt the same dataset and train-test splits (10 splits) as provided by in their publically available implementation[1]. Note that this dataset just has the edge-independent vertex weights $\mathbf{H}$, which is also called the incidence matrix. So this experiment can be regarded as a special case of the specific application of our model (i.e. $\mathbf{Q} = \mathbf{H}$). For baselines, we include Multi-layer perceptron (MLP) which can be considered as the model without hypergraph structure, current state-of-the-art message-passing hypergraph framework UniGNN (Huang & Yang, 2021) and two recent spectral-based hypergraph convolutonal networks: Hypergraph neural networks(HGNN) (Feng et al., 2019) and HyperGCN (Yadati et al., 2019). For UniGNN, we reuse the results of UniGCN reported in (Huang & Yang, 2021), and for HGNN and HyperGCN, we reproduce them following Appendix I.2.

We use cross-entropy loss and Adam SGD optimizer with early stopping with patience of 100 epochs to train GHCN and SHSC. For other hyper-parameters, we use the grid search strategy. It is worth noting that the layer number of our model is $K$ in equation 8. More details of hyper-parameters can be found in Table 6.

Table 11: summary of the ModelNet40 and NTU datasets

| Dataset | ModelNet40 | NTU |
|---|---|---|
| Objects | 12311 | 2012 |
| MVCNN Feature | 4096 | 4096 |
| GVCNN Feature | 2048 | 2048 |
| Training node | 9843 | 1639 |
| Testing node | 2468 | 373 |
| Classes | 40 | 67 |

## I.4 VISUAL OBJECT CLASSIFICATION

**Datasets and Settings.** We employ two public benchmarks: Princeton ModelNet40 dataset (Wu et al., 2015) and the National Taiwan University (NTU) 3D model dataset (Chen et al., 2003), as shown in Table 11.

In this experiment, each 3D object is represented by the feature vectors which are extracted by Multi-view Convolutional Neural Network (MVCNN) (Su et al., 2015) or Group-View Convolutional Neural Network (GVCNN) (Feng et al., 2018). The features generated by different methods can be considered as multi-modality features. The hypergraph structure we designed is similar to Zhang et al. (2018)( but they did not give spectral guarantees for supporting the rationality of their practices). We represent the hypergraph structure as a edge-dependent vertex weight $\mathbf{Q}$ to satisfy the condition (2) in Thm. 1 (i.e. $\mathbf{Q}_1 = \mathbf{Q}_2 = \mathbf{Q}$). Specifically, we firstly generate hyperedges by $k$-NN approach, i.e. each time one object can be selected as centroid and its $k$ nearest neighbors are used to generate one hyperedge including the centroid itself (in our experiment, we set $k = 10$). Then, given the features of data, the vertex-weight matrix $\mathbf{Q}$ is defined as

$$\mathbf{Q}(v, e) = \begin{cases} \exp(\frac{-d(v, v_c)}{\gamma \hat{d}^2}), & \text{if } v \in e \\ 0, & \text{otherwise,} \end{cases} \tag{54}$$

where $d(v, v_c)$ is the euclidean distance of features between an object $v$ and the centroid object $v_c$ in the hyperedge and $\hat{d}$ is the average distance between objects. $\gamma$ is a hyper-parameter to control the flatness. Because we have two-modality features generated by MVCNN and GVCNN, we can obtain the matrix $\mathbf{Q}_{\{i\}}$ which corresponds to the data of the $i$-th modality ($i \in \{1, 2\}$). After all the hypergraphs from different features have been generated, these matrices $\mathbf{Q}_{\{i\}}$ can be concatenated to build the multi-modality hypergraph matrix $\mathbf{Q} = [\mathbf{Q}_{\{1\}}, \mathbf{Q}_{\{2\}}]$. The features generated by GVCNN or MVCNN can be singly used, or concatenated to a multi-modal feature for constructing the hypergraphs.

For baselines, we just compare with HGNN method for multi-modality learning, following the settings of Appendix I.2. We also compare our methods using two-modality features to recent SOTA methods on ModelNet40 dataset. And we use the datasets provided by its public Code [2]. We use cross-entropy loss and Adam SGD optimizer with early stopping with patience of 100 epochs to train GHCN and SHSC. It is worth noting that the layer number of our model is $K$ in equation 8. More details of hyper-parameters can be found in Table 6.

## I.5 PROTEIN QUALITY ASSESSMENT AND FOLD CLASSIFICATION

**Protein hypergraph modeling.** At the high level, a protein is a chain of amino acids (residues) that will form 3D structure by spatial folding. In order to simultaneously consider protein sequence and spatial structure information, we build sequence hyperedge and distance hyperedge. Specifically, given a protein with $|S|$ amino acids, we choose $\tau$ consecutive amino acids $(v_i, v_{i+1}, \cdots, v_{i+\tau})$ to connect to form a sequence hyperedge and choose amino acids whose spatial Euclidean distance is less than a threshold $\epsilon > 0$ to connect to form a spatial hyperedge, where $v_i(i = 1, \cdots, |S|)$ represent the $i$-th amino acid in the sequence. Let $\mathcal{E}_s$ and $\mathcal{E}_e$ denote sequence hyperedge and spatial hyperedge, respectively. Then, we design an edge-dependent vertex-weight matrix $\mathbf{Q}$ for capturing the more granular high-order relationships of proteins below (actually, our

---

[1]https://github.com/malllabiisc/HyperGCN, Apache License
[2]https://github.com/iMoonLab/HGNN, MIT License

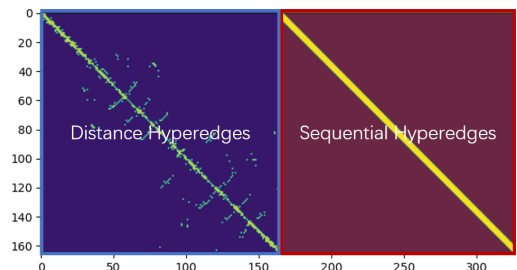

Figure 7: An expample of edge-dependent vertex weight matrix $\mathbf{Q}$ of protein.

models GHCN and SHSC allow one to design a more comprehensive $\mathbf{Q}$ for learning proteins better):

$$
Q(v, e) = \begin{cases} 1, & \text{if } v \in e \text{ and } e \in \mathcal{E}_s \\ \exp(\frac{-d(v,v_c)}{\gamma \hat{d}_{v_c}^2}), & \text{if } v \in e \text{ and } e \in \mathcal{E}_e \\ 0, & \text{otherwise} \end{cases} \tag{55}
$$

where $d(v, v_c) < \epsilon$ is the euclidean distance between an amino acid $v$ and the centroid amino acid $v_c$ in the hyperedge and $\hat{d}_{v_c}$ is the average distance between $v_c$ and the other amino acids $\{v_i\}_{i \neq c}$. $\gamma$ is a hyper-parameter to control the flatness. Here we just design an edge-dependent vertex-weights matrix $\mathbf{Q}$ for satisfying the condition (2) in Thm. 1 (i.e. $\mathbf{Q}_1 = \mathbf{Q}_2 = \mathbf{Q}$). Note that this $\mathbf{Q}$ matrix pays more attention to the information of the sequence (Figure 7).

**Experimental settings**  In our experiment, we set $\tau = 6$ and $\epsilon = 8\mathring{A}$. The initial node features, following Baldassarre et al. (2020), are composed of amino acid types and 3D spatial features including dihedral angles, surface accessibility and secondary structure type generated by DSSP (Kabsch & Sander, 1983). Note that both Quality Assessment and Fold Classification include graph-level tasks, which means we should add global pooling layers to readout the node representations from our GHCN or SHSC.

Here, for the sake of simplicity, we adopt the permutation-invariant operator *mean pooling* and a *single layer MLP* as **Readout** layers to obtain a global hypergraph embedding and add the *softmax* (classification) or *sigmoid* (regression) activation function before output.

Table 12: Comparison of our method to others on protein Quality Assessment tasks. At the residue level, We report *Pearson correlation* across all residues of all decoys of all targets ($R$) and *Pearson correlation* all residues of per decoys and then average all decoys ($R_{decoy}$) with LDDT scores. At the global level, we report *Pearson correlation* across all decoys of all targets ($R$) and *Pearson correlation* per target and then average over all targets ($R_{target}$) with GDT_TS scores.

| Test set | Methods | GDT_TS | | LDDT | |
|---|---|---|---|---|---|
| | | $R$ | $R_{target}$ | $R$ | $R_{model}$ |
| CASP13 | HGNN | 0.714 | 0.622 | 0.651 | 0.337 |
| | GHCN | **0.718** | 0.620 | **0.657** | 0.352 |
| | SHSC | 0.628 | 0.565 | 0.654 | **0.435** |
| CASP12 | VoroMQA | - | 0.557 | - | - |
| | RWplus | - | 0.313 | - | - |
| | 3D CNN | - | 0.607 | - | - |
| | AngularQA | 0.651 | 0.439 | - | - |
| | HGNN | 0.667 | **0.582** | 0.632 | 0.319 |
| | GHCN (ours) | 0.737 | **0.609** | 0.656 | 0.340 |
| | SHSC (ours) | **0.760** | 0.554 | **0.678** | **0.449** |

Protein QA is used to estimate the quality of computational protein models in terms of divergence from their native structure. It is a regression task aiming to predict how close the decoys to the unknown, native structure. Inspired by Baldassarre et al. (2020), we train our models on Global Distance Test Score (Zemla, 2003), which is the global-level score, and the Local Distance Difference Test (Mariani et al., 2013), an amino-acids-level score. The loss function of QA is defined as the

Mean Squared Error (MSE) losses:

$$\mathcal{L}_g = MSE(\mathcal{P}^g_{pred} - \text{GDT\_TS}) \qquad \mathcal{L}_l = \sum_{i=1}^{|S|} MSE(\mathcal{P}^l_{pred_i} - \text{LDDT}_i) \qquad (56)$$

where $\mathcal{P}^g_{pred}$ and $\mathcal{P}^l_{pred}$ denote predicted score of global and local respectively.

### I.5.1 PROTEIN QUALITY ASSESSMENT (QA)

Table 13: summary of CASP datasets

| Dataset | Targets | Decoys | Usage |
|---------|---------|--------|-------|
| CASP 10 | 103 | 26254 | Train |
| CASP 11 | 85 | 12563 | Train |
| CASP 12 | 40 | 6924 | Test |
| CASP 13 | 82 | 12336 | Train |

**Dataset and settings** We use the data from past years' editions of CASP, including CASP10-13. We randomly split the CASP10, CASP11, CASP13 for training and validation, with ratio training: validation = 9:1. CASP 12 is set aside for testing against other methods. More details about the datasets can be found in Table 13. For the baseline, we compare our methods with other start-of-the-art methods, including random walk-based methods : RWplus (Zhang & Zhang, 2010), sequence-based methods: AngularQA (Conover et al., 2019), and 3D structrue-based methods: VOroMQA (Olechnovivc & Venclovas, 2017), 3DCNN (Derevyanko et al., 2018). The results of these baselines we reused from Baldassarre et al. (2020) reports. Another baseline HGNN (Feng et al., 2019) is reproduced by us with same training strategy as our methods.

Because our hypergraph based methods can jointly learn node and graph embeddings, the losses in equation 56 can be weighted as $\mathcal{L}_{total} = \mu\mathcal{L}_l + (1-\mu)\mathcal{L}_g$ and co-optimized by Adam Optimizer with $L_2$ regularization, where $\mu = 0.5$ in our experiment. We use grid search to select hyper-parameters and more details can be found in Table 8.

In addition, we use CASP10-12 for training and valuation, CASP 13 for testing to further evaluate the efficiency of our methods (the range of hyper-parameters can see Table 9), and the results are shown in Table 12.

### I.5.2 PROTEIN FOLD CLASSIFICATION

**Datasets and settings** The dataset that we used for training, validation and test is SCOPe 1.75 data set of Hou et al. (2018). This dataset includes 16,712 proteins covering 7 structural classes with total of 1195 folds. The 3D structures of proteins are obtained from SCOpe 1.75 database (Murzin et al., 1995), in which each protein save in a PDB file. The datasets have three test sets: 1) Fold, where proteins from the same superfamily do not appear in the training set; 2) Family, in which proteins from the same family are not present in the training set; 3) Family, where proteins from the same family that are present in the training set.

For baselines, we include sequence-based methods pre-trained unsupervised on millions of protein sequences: Rao et al. (2019); Bepler & Berger (2018); Strodthoff et al. (2020), 3D structure based model: Kipf & Welling (2017); Diehl (2019); Baldassarre et al. (2020) and Gligorijevic et al. (2020), who process the sequence with LSTM first and then apply GCNN. The accuracy of the above baselines is reused from Hermosilla et al. (2021) reported. We adopt Adam optimizer to minimize the cross-entropy loss of our methods. Hyper-parameters search range can see Table 7.

### I.6 OVER-SMOOTHING ANALYSIS.

Table 14 shows that our proposed SHSC significantly alleviates the performance descending with the increase of layers. Given the same layers, it can be observed that our SHSC almost outperforms the other methods for all cases, especially at a deep layer, which demonstrates the benefits of deep model and the long-range information around hypergraph. Furthermore, It should be noted that the optimal layer numbers $K$ of our SHSC are generally larger than other models, due to the only one

linear layer avoids the over-fitting problem. These results also reveal that HyperGCN, HGNN and our GHCN all suffer from severe over-smoothing issue, limiting the power of neural network to capture high-order relationships.

Table 14: Summary of classification accuracy (%) results with various depths. In our SHSC, the number of layers is equivalent to $K$ in equation 8. We report mean test accuracy over 10 train-test splits.

| Dataset | Method | Layers | | | | | |
|---|---|---|---|---|---|---|---|
| | | 2 | 4 | 8 | 16 | 32 | 64 |
| Cora (co-authorship) | HyperGCN | 60.66 | 57.50 | 31.09 | 31.10 | 30.09 | 31.09 |
| | HGNN | 69.23 | 67.23 | 60.17 | 29.28 | 27.15 | 26.62 |
| | GHCN (ours) | **74.79** | 72.86 | 63.99 | 31.03 | 30.46 | 31.09 |
| | SHSC (ours) | 74.60 | **75.78** | **75.70** | **75.04** | **75.26** | **74.79** |
| DBLP (co-authorship) | HyperGCN | 84.82 | 54.65 | 22.37 | 23.96 | 23.04 | 24.13 |
| | HGNN | 88.55 | 88.28 | 85.38 | 27.64 | 27.62 | 27.56 |
| | GHCN (ours) | **89.04** | **88.90** | 85.15 | 27.61 | 27.61 | 27.62 |
| | SHSC (ours) | 86.63 | 88.26 | **89.00** | **89.17** | **89.05** | **88.60** |
| Cora (co-citation) | HyperGCN | 62.35 | 58.29 | 31.09 | 31.17 | 31.09 | 29.68 |
| | HGNN | 55.60 | 55.72 | 42.10 | 26.16 | 24.40 | 24.43 |
| | GHCN (ours) | **69.03** | **69.45** | 57.37 | 28.21 | 26.27 | 26.95 |
| | SHSC (ours) | 62.21 | 64.57 | **67.59** | **68.96** | **69.37** | **68.15** |
| Pubmed (co-citation) | HyperGCN | 68.12 | 63.59 | 39.99 | 39.97 | 40.01 | 40.02 |
| | HGNN | 46.41 | 47.16 | 40.93 | 40.24 | 40.30 | 40.29 |
| | GHCN (ours) | **75.37** | 74.76 | 60.65 | 40.38 | 40.31 | 40.42 |
| | SHSC (ours) | 74.39 | **74.91** | **74.41** | **73.90** | **72.79** | **71.49** |
| Citeseer (co-citation) | HyperGCN | 56.94 | 36.75 | 20.72 | 20.41 | 20.16 | 18.95 |
| | HGNN | 39.93 | 38.98 | 36.67 | 19.91 | 19.86 | 19.79 |
| | GHCN (ours) | **62.67** | 61.50 | 49.94 | 21.95 | 21.84 | 21.93 |
| | SHSC (ours) | 61.63 | **62.75** | **63.86** | **64.62** | **65.14** | **65.10** |

## I.7 SENSITIVITY ANALYSIS

The proposed SHSC model performance on co-authorship Cora, co-citation cora and co-citation Pubmed with different $\sigma$, $\beta$ and $\alpha$ is reported in Figure 8. For $\sigma$, we can see that the best choice will vary depending on the data set but mainly concentrated around $-0.5$, which verify that the effect of hyperedges degree is various and it has a negative effect in most cases. For $\beta$, with the growth of $\beta$, the performance of SHSC stably increases, which means that the diffusion kernel is useful for information aggregation. Moreover, the fact that the performance remains stable on cora (coauthorship) and pubmed (cocitation) when $\beta$ is at 0.8-1.0 suggests we can only adjust this hyper-parameter at a range of large value. For $\alpha$, the tendency of it is similar to $\beta$, so we can also adjust this hyper-parameter at a range of large value to obtain a satisfying performance.

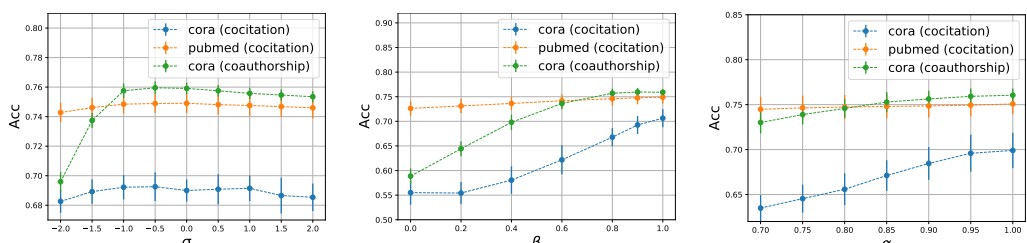

Figure 8: Test accuracy by varying the hyper-parameters $\sigma$ (left), $\beta$ (middle) and $\alpha$ (right) of SHSC.

## I.8 THE SPECTRUM ANALYSIS OF SHSC

We calculate the eigenvalues of $\left( \beta \sum_{k=1}^{K} \frac{\alpha^k}{K} \tilde{\mathbf{T}}^k + (1-\beta)\mathbf{I} \right)$ with various $(\alpha, \beta)$ on NTU2012 dataset ($|\mathcal{V}| = 2012$) and count the number of eigenvalues in different size ranges, which is shown

in the Table 15. The table suggests that our SHSC is able to capture both low and high frequency information of the graph signal, depending on the selection of the appropriate hyper-parameters $(\alpha, \beta)$.

Table 15: The number of eigenvalues of SHSC in different size ranges on NTU2012 dataset.

| $\alpha, \beta$ | $\lambda \geq 0.0$ | $\lambda \geq 0.1$ | $\lambda \geq 0.2$ | $\lambda \geq 0.3$ | $\lambda \geq 0.4$ | $\lambda \geq 0.5$ | $\lambda \geq 0.6$ | $\lambda \geq 0.7$ | $\lambda \geq 0.8$ | $\lambda \geq 0.9$ |
|---|---|---|---|---|---|---|---|---|---|---|
| (1,1) | 2012 | 364 | 163 | 108 | 82 | 66 | 52 | 38 | 26 | 13 |
| (1,0.8) | 2012 | 327 | 140 | 90 | 71 | 51 | 33 | 19 | 1 | 0 |
| (0.8,1) | 2012 | 161 | 40 | 0 | 0 | 0 | 0 | 0 | 0 | 0 |
| (0.8,0.8) | 2012 | 139 | 8 | 0 | 0 | 0 | 0 | 0 | 0 | 0 |

## I.9 ABLATION ANALYSIS

In order to verify the effectiveness of our proposed edge-dependent vertex weight $\mathbf{Q}$ and the necessity of re-normalization trick, we conduct an ablation analysis and report the results in Table 16. Specially, *w/o* $\mathbf{Q}$ is a variant of our methods that replaces the edge-dependent vertex weight $\mathbf{Q}$ with the edge-independent vertex weight matrix $\mathbf{H}$. *w/o renormalization* represents the variants without re-normalization trick. Form the reported results we can learn that both $\mathbf{Q}$ and renormalization are efficient for hypergraph learning, respectively. Moreover, on Cora and Pubmed, the performance gap between *w/o renormalization* and *with both* GHCN reveals the advantage of renormalization on disconnected hypergraph dataset. This phenomenon is mainly caused by the row in the adjacent matrix corresponding to an isolated point is 0 (Figure 4), resulting in a direct loss of its vertex information (Figure 4). And the renormalization trick adding the self-loop to matrix $\mathbf{K}$ can maintain the features of isolated vertices during aggregation. Compared with GHCN, the performance impact of renormalization on SHSC seems to be smaller. This is because we add initial features to the information gathered by the neighbors, which reduces the information loss of isolated vertices.

Table 16: Test accuracy (%) of our methods for ablation analysis. We report mean ±standard deviation. Cora and Pubmed are the datasets that do not contain $\mathbf{Q}$, so we just report the w/o renormalization results. NTU and ModelNet40 constructed by Feng et al. (2019) are both connected hypergraph networks in this work.

| methods | - | Cora (co-authorship) | Pubmed (co-citation) | NTU | ModelNet40 |
|---|---|---|---|---|---|
| GHCN | with both | **74.79±0.91** | **75.37±1.2** | **85.15±0.34** | **97.28±0.15** |
| | w/o $\mathbf{Q}$ | - | - | 84.93±0.31 | 97.18±0.20 |
| | w/o renormalization | 69.23±1.6 | 46.41±0.70 | 84.85±0.30 | 97.20±0.15 |
| | w/o both | - | - | 84.21±0.25 | 97.15±0.14 |
| SHSC | with both | **75.91±0.75** | **74.60±1.4** | **83.35±0.30** | **97.74±0.05** |
| | w/o $\mathbf{Q}$ | - | - | 82.84±0.40 | 97.65±0.08 |
| | w/o renormalization | 75.76±0.69 | 73.83±2.1 | 82.92±0.47 | **97.74±0.05** |
| | w/o both | - | - | 82.55±0.46 | 97.69±0.07 |

## I.10 EXAMPLES OF $\rho$

We give many examples for $\rho$, including $random$, $x^{-1}$, $sigmoid(x)$ ,$\frac{1}{\sqrt{2}\sigma}exp(-\frac{(x-\bar{x})^2}{2\sigma^2})$, $log(x)$ ,$exp(x)$ and $exp(-x)$, and the experimental results on citation networks can be seen the table 17 . Regarding learnable $\rho$, we will do as a future work.

Table 17: classifificaiton accuracy(standard deviation) with different $\rho$

| dataset | model | random | $x^{-1}$ | $sigmoid(x)$ | $\frac{1}{\sqrt{2}\sigma}exp(-\frac{(x-\bar{x})^2}{2\sigma^2})$ | $log(x)$ | $exp(x)$ | $exp(-x)$ |
|---|---|---|---|---|---|---|---|---|
| pubmed | GHCN | 0.74(±0.01) | 0.75(±0.01) | 0.74(±0.02) | 0.74(±0.01) | 0.74(±0.02) | 0.73(±0.02) | 0.73(±0.01) |
| pubmed | SHSC | 0.75(±0.01) | 0.75(±0.01) | 0.75(±0.01) | 0.75(±0.01) | 0.75(±0.01) | 0.74(±0.01) | 0.74(±0.01) |
| cora | GHCN | 0.72(±0.02) | 0.73(±0.01) | 0.72(±0.02) | 0.72(±0.01) | 0.71(±0.02) | 0.71(±0.01) | 0.60(±0.02) |
| cora | SHSC | 0.72(±0.01) | 0.73(±0.01) | 0.72(±0.01) | 0.73(±0.01) | 0.72(±0.01) | 0.72(±0.01) | 0.64(±0.01) |

### I.11 RUNNING TIME AND COMPUTATIONAL COMPLEXITY

Firstly, we analyze the theoretical computational complexity of our GHCN and SHSC: For GHCN, the computational cost is the $\mathcal{O}(|E|d)$, where $|E|$ is the total edge count in equivalent undigraph. Each sparse matrix multiplication $\tilde{\mathbf{T}}\mathbf{X}$ costs $|E|d$. And the computational of HGNN is the same as GHCN. For SHSC, the computational cost is the $\mathcal{O}(K|E|d + K|\mathcal{V}|d)$, which includes $K$ sparse matrix multiplication and $K$ summation over filters($|\mathcal{V}|d$ is the cost of adding features $X$). Then, we compare the running time between our GHCN and SHSC with existing models in Table 18 and the results illustrate that our methods are of the same order of magnitude as SOTA's approach UniGNN and outperform the HyperGCN and HGAT.

Table 18: The average training time per epoch with different methods on citation network classification task is shown below and timings are measured in seconds. The float in parentheses is the standard deviation.

| Methods | cora coauthorship | dblp coauthorship | cora cocitation | pubmed cocitation | citeseer cocitation |
|---|---|---|---|---|---|
| HyperGCN | 0.150($\pm$0.058) | 1.181($\pm$0.071) | 0.151($\pm$0.029) | 1.203($\pm$0.104) | 0.130($\pm$0.029) |
| HGNN | 0.005($\pm$0.002) | 0.081($\pm$0.006) | 0.005($\pm$0.040) | 0.008($\pm$0.002) | 0.005($\pm$0.002) |
| UniGNN | 0.014($\pm$0.044) | 0.042($\pm$0.040) | 0.014($\pm$0.042) | 0.023($\pm$0.043) | 0.0168($\pm$0.043) |
| HNHN | 0.001($\pm$0.0026) | 0.007($\pm$0.014) | 0.0010($\pm$0.004) | 0.009($\pm$0.006) | 0.001($\pm$0.003) |
| ChebNet | 0.027($\pm$0.005) | 0.063($\pm$0.001) | 0.073($\pm$0.008) | 0.050($\pm$0.019) | 0.067($\pm$0.0198) |
| SSGC | 0.055($\pm$0.001) | 0.291($\pm$0.001) | 0.205($\pm$0.003) | 0.135($\pm$0.056) | 0.193($\pm$0.057) |
| HGAT | 0.381($\pm$0.080) | OOM | 0.279($\pm$0.083) | 1.329($\pm$0.016) | 0.286($\pm$0.087) |
| GHCN(ours) | 0.005($\pm$0.036) | 0.020($\pm$0.079) | 0.005($\pm$0.039) | 0.011($\pm$0.075) | 0.081($\pm$0.091) |
| SHSC(ours) | 0.016($\pm$0.010) | 0.072($\pm$0.001) | 0.038($\pm$0.005) | 0.011($\pm$0.002) | 0.027($\pm$0.001) |

### I.12 THE EDGE-DEPENDENT VERTEX WEIGHTS VISUALIZATION

#### I.12.1 VISUAL OBJECT.

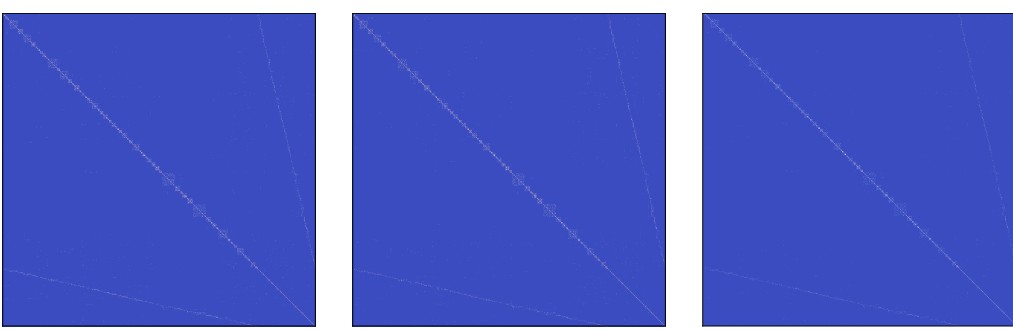

Figure 9: The edge-dependent vertex weights of NTU2012 dataset (MVCNN feature + MVCNN structure). The figure on the left represents the distance-based vertex weights matrix $\mathbf{Q}$ used in our GHCN and SHSC. The middle ($\mathbf{Q}_1$) and right ($\mathbf{Q}_2$) denote the node-level and edge-level attention coefficient matrix of HGAT (Ding et al., 2020), respectively.

## I.13 ADDITIONAL EXPERIMENTS OF BASELINES.

Table 19: Summary of classificaiton accuracy(%) results. We report the average test accuracy and its standard deviation over 10 train-test splits. The number in parentheses corresponds to the number of layers of the model. (OOM: our of memory)

| Dataset | Architecture | Cora (co-authorship) | DBLP (co-authorship) | Cora (co-citation) | Pubmed (co-citation) | Citeseer (co-citation) |
|---|---|---|---|---|---|---|
| MLP | - | 52.02±1.7 | 78.72±0.6 | 52.02±1.7 | 69.86±1.6 | 55.03±1.3 |
| HyperGCN | spectral-based | 60.66±10.8 | 84.82±9.7 | 62.35±9.3 | 68.12±9.7 | 56.94±6.3 |
| HGNN | spectral-based | 69.23±1.6 | 88.55±0.18 | 55.60±1.8 | 46.41±0.7 | 38.98±1.1 |
| HNHN | message-passing | 63.95±2.4 | 84.43±0.3 | 41.59±3.1 | 41.94±4.7 | 33.60±2.1 |
| HGAT | message-passing | 65.42±1.5 | OOM | 52.21±3.5 | 46.28±0.53 | 38.32±1.5 |
| UniGNN | message-passing | 75.30±1.2 | 88.80±0.2 | 70.10±1.4 | 74.40±1.0 | 63.60±1.3 |
| ChebNet | spectral-based | 42.27±2.2 | 85.81±5.22 | 45.55±2.37 | 65.73±1.3 | 48.44±0.91 |
| APPNP | spectral-based | 68.25±1.9 | 86.99±0.33 | 64.50±1.7 | 72.02±1.4 | 55.05±2.0 |
| SSGC | spectral-based | 72.04±1.2 | 88.61±0.16 | 68.79±2.1 | 74.49±1.3 | 60.52±1.7 |
| GHCN (ours) | spectral-based | 74.79±0.91 | 89.04±0.19 | 69.45±2.0 | **75.37±1.2** | 62.67±1.2 |
| SHSC (ours) | spectral-based | **76.05±0.75(6)** | **89.17±0.21(16)** | **70.64±1.8 (32)** | 75.08±1.1(4) | **65.14±0.97(32)** |

Table 20: Test accuracy on visual object classification. Each model we ran 10 random seeds and report the mean ± standard deviation. BOTH means GVCNN+MVCNN, which represents combining the features or structures to generate multi-modal data.

| Datasets | Feature | Structure | HGNN | UniGNN | HGAT | ChebNet | SSGC | GHCN(ours) | SHSC(ours) |
|---|---|---|---|---|---|---|---|---|---|
| NTU | MVCNN | MVCNN | 80.11±0.38 | 75.25±0.17 | 80.40±0.47 | 72.17±3.0 | 81.23±0.24 | 81.37±0.63 | **82.56±0.39** |
| | GVCNN | GVCNN | 84.26±0.30 | 84.63± 0.21 | 84.45±0.12 | 82.52±1.0 | 84.26±0.12 | **85.15±0.34** | 83.35±0.30 |
| | BOTH | BOTH | 83.54±0.50 | 84.45±0.40 | 84.05±0.36 | 79.17±1.8 | 84.13±0.34 | 84.45±0.40 | **85.12±0.25** |
| Model-Net40 | MVCNN | MVCNN | 91.28±0.11 | 90.36±0.10 | 91.29±0.15 | 85.81±5.2 | 91.21±0.11 | 91.99±0.16 | **92.01±0.08** |
| | GVCNN | GVCNN | 92.53±0.06 | **92.88±0.10** | 92.44±0.11 | 92.08±0.17 | 92.74±0.04 | 92.66±0.10 | 92.69±0.06 |
| | BOTH | BOTH | 97.15±0.14 | 96.69±0.07 | 96.44±0.15 | 85.98±17 | 97.07±0.07 | 97.28±0.15 | **97.78±0.03** |

