# OpenReview forum: " Hypergraph Convolutional Networks via Equivalency  between  Hypergraphs and Undirected Graphs"
_ICLR.cc/2022/Conference — ICLR 2022 Submitted_

### Official Review · Reviewer_RmER · 2021-10-26

**Correctness:** 4
**Technical Novelty And Significance:** 2
**Empirical Novelty And Significance:** 2
**Recommendation:** 5
**Confidence:** 5

**Main Review:**

### **Strengths**

1. **Quality of Paper**

The paper is technically sound and claims are well supported by theoretical analysis.
Extensive experimentation across different domains adds to the quality of the paper.
The paper also discusses limitations in addition to highlighting its strengths.


2. **Clarity of Presentation**

The paper is clearly written and well organised.
The appendix is comprehensive with both theoretical proofs and empirical details (e.g., dataset construction, hyperparameters, etc.).
The authors have also released the source code as part of the supplementary material.


### **Weaknesses**

1. **Originality of Contributions**

The theory of equivalence between EDVW hypergraph and undirected graph is a small contribution of the paper in which vertex weights are added into the first step of the random walk. Empirically, however, edge-dependent vertex weights have been previously explored  in the form of attention mechanisms (node-level and edge-level attention mechanisms), e.g., the HGAT model proposed in (i) Be More with Less: Hypergraph Attention Networks for Inductive Text Classification, EMNLP'20, and (ii) Session-based Recommendation with Hypergraph Attention Networks, SDM'21. Another weakness of the paper is that once the Laplacian is defined, the proposed methods, viz., GHCN and SHSC are straightforward applications of existing spectral convolutional methods viz., ChebNet [Defferrard et al. NeurIPS'16] and SSGC [Zhu & Koniusz, ICLR'21] to hypergraphs.


2. **Significance of Empirical Evaluation**

The paper can be significantly improved by comparing with the HGAT baseline, i.e., a baseline with node-level, edge-level attention mechanisms.
Moreover, SSGC and ChebNet on the clique expansion are also important missing baselines in all the experiments.
Empirical insights (e.g., comparison of attention weights and weights in the proposed $Q$ matrix for visual and protein data) would improve the paper.

**Summary Of The Paper:**

Real-world relational datasets (e.g., academic data, protein data) contain group relationships and can be modelled via hypergraphs.  One way to capture group higher-order relationships is to define a hypergraph with edge-dependent vertex weights (EDVWs) and such a hypergraph has been shown to be equivalent to directed graphs [Chitra et al., ICML'19]. The main contributions of this paper are:
1) Proposal of a generalised hypergraph capturing EDVWs with equivalence to undirected graphs,
2) Proposal of a Laplacian based on the generalised EDVW and exploration of existing spectral convolutional methods, and
3) Empirical evaluation on academic, protein, and visual object datasets.

**Summary Of The Review:**

Overall the paper is clear and of good quality with claims being well supported by theoretical analysis. However, the novelty is incremental and experimental results are marginally significant. Positioning with important prior work and evaluation with missing baselines would significantly improve the paper.

---

> ### Author Response · Authors · 2021-11-17
> **Response to Reviewer RmER(Part 1)**
>
> Thank you very much for your detailed comments and suggestions！
> Note that there are several misunderstandings regarding  this
> work, for which we will clarify in detail below.
>
> ### Comment:
> >The theory of equivalence between the EDVW hypergraph and the undirected graph is a small contribution of the
> paper in which vertex weights are added into the first step of the random walk.
>
>
> **ANSWER:**
>
>   This may be a misunderstanding of our main contribution. Our core contribution is building up the equivalency between the defined generalized hypergraphs and undirected graphs, not just adding the vertex weights in the first steps.
>
>  We clarify the theoretical as well as practical implications of the equivalency condition.
>
> - i) The equivalency led by the random walk is meaningful and valuable. As we stated in the analysis of theorem 1, the equivalency conditions provide new insights for hypergraph foundation, hypergraph application, and hypergraph learning.
>
> - ii) This equivalency provides a plug-and-play framework that not only can be used to deduce the two spectral convolutions but also could seamlessly incorporate existing undirected graph models for (EDVW-)hypergraph learning.
>
> ### Comment:
> >edge-dependent vertex weights have been previously explored in the form of attention mechanisms (node-level
> and edge-level attention mechanisms), e.g., the HGAT model.
>
> **ANSWER:**
> We clarify the difference between our methods and HGAT in terms of EDVWs.
>
> - i) Our methods enjoy a solid theoretical foundation. HGAT utilizes the learnable edge-dependent vertex-weights for hypergraph learning, but it heuristically embeds EDVWs into the message passing without a theoretical guarantee. Conversely, we embed the EDVWs into the undirected graph Laplacian via the equivalency condition and then get the weighted equivalent undirected graph. As a result, the EDVW-hypergraph learning problem can be handled with solid theoretical equivalency guarantees by utilizing the mature undirected graph CNNs.
>
> - ii ) Experimentally, our methods achieve better performance. We compare our models with HGAT and the analysis of results can be seen in the last part. Overall, HGAT does not achieve better performance than our methods but requires higher computational complexity and running time.
>
>
> ### Comment:
> >Another weakness of the paper is that once the Laplacian is
> defined, the proposed methods, viz., GHCN and SHSC are straightforward applications of existing spectral
> convolutional methods.
>
> **ANSWER:**
>  This is a misunderstanding of our main technique route.
>
> The direct usage of existing spectral convolutional models is not a weakness, but rather is the advantage of our approach (this insight is also confirmed by Reviewer NQ7e).  Thanks to the equivalency theory, we can apply advanced techniques of undigraph NNs to hypergraph learning as a convenient research route, which would largely ease hypergraph learning in real-world scenarios. By the way, our SHSC is different from SSGC by introducing the discount factor $\alpha$, which can alleviate long-range information aggregation in the hypergraphs.
>
> Our approach is non-trivial in that it provides a solution to the hypergraph problem from a theoretical equivalence perspective, and we experimentally validate that this idea is very practical and effective. It even surpasses the learnable HGAT model in the exploration of EDVW-hypergraphs.

---

> > ### Author Response · Authors · 2021-11-17
> > **Response to Reviewer RmER(Part 2)**
> >
> > ### Comment:
> > >The paper can be significantly improved by comparing with the HGAT baseline, i.e., a baseline with node-level, edge-level attention mechanisms. Moreover, SSGC and ChebNet on the clique expansion are also important missing baselines in all the experiments
> >
> > **ANSWER:**
> >
> > Thank you for your thoughtful considerations of the missing baselines.
> >
> >
> > i) For HGAT:
> >
> > We obtain another perspective to understand the edge-independent vertex-weights, since in the paper we just consider how to utilize the predetermined vertex weights and embed them into the undirected graphs, thus ignoring whether the learnable vertex-weights will be more informative. We add the experiments of HGAT shown in the following table.
> >
> > ii) For SSGC and ChebNet:
> >  Actually, any undirected graph convolutional methods can be used for hypergraph learning via our equivalency theory frameworks. Sorry that we just provide two cases (GHCN, SHSC) in the main text, and the two advised models (SSGC, ChebNet) we will add as baselines in the final version.  The results on citation classification are shown as follows. (We update Table 19 in Appendix I.13 of our manuscript)（OOM means Out Of Memory.）
> >
> > |Methods|cora coauthorship(11.8\%)|dblp coauthorship(0%)|Cora cocitation(47.0%)|Pubmed cocitation(80.5%)|citeseer cocitation(55.9%)|
> > |-|-|-|-|-|-|
> > |HGAT|65.42$\pm$1.5 | OOM |52.21$\pm$3.52 | 46.28$\pm$0.53|38.32$\pm$1.5
> > |ChebNet|42.27$\pm$2.17 |85.81$\pm$5.22| 45.55$\pm$2.37 | 65.73$\pm$1.3| 48.44$\pm$0.91 |
> > |SSGC | 72.04$\pm$1.2 | 88.61$\pm$0.16 | 68.79$\pm$2.1  | 74.49$\pm$1.3 | 60.52$\pm$1.7
> > |GHCN(ours) | 74.79$\pm$0.91 | 89.04$\pm$ 0.19 |69.45$\pm$ 2.0 |**75.37$\pm$1.2** | 62.67$\pm$1.2
> > |SHSC(ours) | **76.05$\pm$0.75** | **89.17$\pm$0.21**  | **70.64$\pm$1.8** | 75.08$\pm$ 1.1 |**65.14$\pm$0.97**
> >
> > The HGAT and ChebNet fail to capture the isolated vertices in hypergrpah~(the ratio of isolated vertices is shown in the parentheses). So they cannot get a competitive performance on the citation classification task. On the other hand, they can work when the hypergraph network is connected, as we can see on visual object classification tasks in the following table (We also update Table 20 in Appendix I.13 of our manuscript).
> >
> > |Datasets|featue|Structure|HGNN|UniGNN|GHCN(ours)|SHSC(ours)| HGAT | SSGC | ChebNet |
> > |-|-|-|-|-|-|-|-|---|--|
> > |NTU|MVCNN|MVCNN|80.11$\pm$0.38|75.25$\pm$0.17|81.37$\pm$0.63|**82.56$\pm$0.39**| 80.40$\pm$0.47 |81.23$\pm$0.24 | 72.17$\pm$3.0
> > |NTU|GVCNN|GVCNN|84.26$\pm$0.30| 84.63$\pm$ 0.21|**85.15$\pm$0.34**|83.35$\pm$0.30 | 84.45$\pm$0.12 |84.26$\pm$0.12 |82.52$\pm$1.0
> > |NTU|BOTH|BOTH|83.54$\pm$0.50|83.78$\pm$0.18|84.45$\pm$0.40|**85.12$\pm$0.25** |84.05$\pm$0.36 |84.13$\pm$0.34  | 79.17$\pm$1.8
> > |Model-Net40| MVCNN|MVCNN|91.28$\pm$0.11|90.36$\pm$0.10|91.99$\pm$0.16|**92.01$\pm$0.08** |91.29$\pm$0.15 |91.21$\pm$0.11 | 85.81$\pm$5.2
> > |Model-Net40|GVCNN|GVCNN|92.53$\pm$0.06|**92.88$\pm$0.10**|92.66$\pm$0.10|92.69$\pm$0.06 |92.44$\pm$0.11 |92.74$\pm$0.04 | 92.08$\pm$0.17
> > |Model-Net40|BOTH|BOTH|97.15$\pm$0.14|96.69$\pm$0.07|97.28$\pm$0.15|**97.78$\pm$0.03** |96.44$\pm$0.15|97.07$\pm$0.07|85.98$\pm$17.8

---

> > > ### Author Response · Authors · 2021-11-17
> > > **Response to Reviewer RmER(Part 3)**
> > >
> > > ### Comment:
> > > > Empirical insights (e.g., comparison of attention weights and weights in the proposed matrix for visual and protein data) would improve the paper.
> > >
> > > **ANSWER:**
> > >
> > > HGAT does not achieve better performance than our methods but requires higher computational complexity and running time.
> > >
> > > We compare $\mathbf{Q}$ constructed by distance matrix with node-level and edge-level attention coefficients on visual object classification task~(see I.12 in the revised version). The results suggest  HGAT learns two different vertex-weights (i.e. attention coefficients), but their Mean Absolute Distance is only 0.0801. On the other hand, the MAD between $\mathbf Q$ and the attention coefficients matrix is 0.8363, which implies that the EDVWs designed in our methods are significantly different from HGAT. Furthermore, according to the results of SHSC(ours) and HGAT in the table above, the learnable vertex-weights do not achieve better performance than fixed vertex weights $\mathbf Q$. Therefore, we have not yet been able to see the advantage of the learnable vertex weights. Moreover, the learnable attention coefficients require expensive computational effort, while our methods embed the vertex weights into Laplacian, thus avoiding additional computational overhead during training and providing a more robust way to utilize the vertex weights.  The experiment of proteins needs more time so we would try to provide these experiments in the final version when possible. Thank you again for your constructive suggestions to improve the paper.
> > >
> > >
> > > The average training time per epoch with different methods on citation classification task is shown below (We also update Table 18 in Appendix I.11 of the manuscript) and timings are measured in seconds. The float in parentheses is the standard deviation.
> > >
> > >
> > > |   Methods    |cora coauthorship | dblp coauthorship | cora cocitation | pubmed cocitation  |citeseer cocitation|
> > > |-------|-------|-------|-------|-------|--------|
> > > |HyperGCN | 0.149495(0.058368) |  1.181070(0.070887)  | 0.150609(0.029319)  | 1.202762(0.104372)  |0.129688(0.028550)|
> > > |HGNN | 0.004606(0.002452)  | 0.080851(0.005504)  | 0.005047(0.040228)  | 0.007616(0.002257)  |0.005490(0.002278)|
> > > |UniGNN | 0.013877(0.044469)  |  0.042261(0.040026)|  0.014216(0.042498)| 0.023298(0.043278)  |0.016777(0.043392)|
> > > |HNHN| 0.00108(0.00257) | 0.006846(0.014287) |0.00116(0.00352) |0.00870(0.00588)| 0.00135(0.00337)|
> > > |ChebNet|0.027281(0.00542)|0.063258(0.000616)|0.073447(0.008295) |0.050099(0.018820) | 0.066752(0.019770)
> > > |SSGC| 0.055385(0.001355) | 0.291338(0.000311)|0.204895(0.003073) |0.134543(0.055787) | 0.193085(0.057415)|
> > > |HGAT| 0.380661(0.079753) |OOM |0.279292(0.083447) | 1.329564(0.016568) |0.286309(0.087310) |
> > > |GHCN(ours)|  0.005190(0.036183)| 0.020323(0.078726)  | 0.004871(0.039286)  | 0.011252(0.075203) |0.080663(0.09119)|
> > > |SHSC(ours)|  0.016022(0.009814)  | 0.071777(0.001198) |  0.037806(0.004691)  | 0.010635(0.001673 |0.026667(0.001307) |
> > >
> > > The table shows that the average training time for HGAT is about an order of magnitude larger than our methods.

---

> > > > ### Comment · Reviewer_RmER · 2021-11-24
> > > > **Thanks for the Response**
> > > >
> > > > Thanks to the authors for the response and the additional experiments. Please clarify the following queries
> > > >
> > > > * The variance of ChebNet is very high in some cases (e.g., 5.22 on DBLP coauthorship, 17.8 on ModelNet40) while it is comparatively lower in other cases and much lower for competing methods. Please clarify if there was an obvious reason for this behaviour of ChebNet.
> > > > * HGAT/HyperGAT proposed in [1]  and UniGAT proposed in [2] both use attention mechanisms to determine the importance of incident hyperedges. UniGAT is shown to perform without OOM issues on DBLP in [2]. Please clarify why there was an OOM issue with HGAT on DBLP.
> > > >
> > > > [1] Be More with Less: Hypergraph Attention Networks for Inductive Text Classification, EMNLP'20
> > > >
> > > > [2] UniGNN: a Unified Framework for Graph and Hypergraph Neural Networks, IJCAI'21

---

> > > > > ### Author Response · Authors · 2021-11-26
> > > > > **Thanks for the further review！**
> > > > >
> > > > > ### Comment:
> > > > > >The variance of ChebNet is very high in some cases (e.g., 5.22 on DBLP co-authorship, 17.8 on ModelNet40) while it is comparatively lower in other cases and much lower for competing methods. Please clarify if there was an obvious reason for this behavior of ChebNet.
> > > > >
> > > > > **ANSWER:**
> > > > >
> > > > > Thanks for pointing this out! We did not notice this issue due to a large number of additional experiments in the limited rebuttal time. After carefully checking our training procedure, we find that the issue is caused by the learning rate. In the previous experiments, for ChebNet, we adopt the default lr (0.01) used by other models, such as UniGNN, APPNP, and HyperGCN. By tuning the default lr to 0.001, ChebNet can get competitive results on NTU and ModelNet40 datasets. We have updated the table of Part 2 for these datasets (Italicized values indicate updated results). The experiments on DBLP take too long time and we will update the table as soon as it finishes running.
> > > > >
> > > > > It's worth noting that such updates do not change the leading role of our models.  Thanks again for your careful reviews.
> > > > >
> > > > > |Datasets|featue|Structure|HGNN|UniGNN|GHCN(ours)|SHSC(ours)| HGAT | SSGC | ChebNet |
> > > > > |-|-|-|-|-|-|-|-|---|--|
> > > > > |NTU|MVCNN|MVCNN|80.11$\pm$0.38|75.25$\pm$0.17|81.37$\pm$0.63|**82.56$\pm$0.39**| 80.40$\pm$0.47 |81.23$\pm$0.24 | *78.04$\pm$0.46*
> > > > > |NTU|GVCNN|GVCNN|84.26$\pm$0.30| 84.63$\pm$ 0.21|**85.15$\pm$0.34**|83.35$\pm$0.30 | 84.45$\pm$0.12 |84.26$\pm$0.12 | *83.51$\pm$0.40*
> > > > > |NTU|BOTH|BOTH|83.54$\pm$0.50|83.78$\pm$0.18|84.45$\pm$0.40|**85.12$\pm$0.25** |84.05$\pm$0.36 |84.13$\pm$0.34  | *83.16$\pm$0.46*
> > > > > |Model-Net40| MVCNN|MVCNN|91.28$\pm$0.11|90.36$\pm$0.10|91.99$\pm$0.16|**92.01$\pm$0.08** |91.29$\pm$0.15 |91.21$\pm$0.11 | *90.86$\pm$0.29*
> > > > > |Model-Net40|GVCNN|GVCNN|92.53$\pm$0.06|**92.88$\pm$0.10**|92.66$\pm$0.10|92.69$\pm$0.06 |92.44$\pm$0.11 |92.74$\pm$0.04 | *92.46$\pm$0.15*
> > > > > |Model-Net40|BOTH|BOTH|97.15$\pm$0.14|96.69$\pm$0.07|97.28$\pm$0.15|**97.78$\pm$0.03** |96.44$\pm$0.15|97.07$\pm$0.07| *96.95$\pm$0.09*
> > > > >
> > > > > ### Comment:
> > > > > >HGAT/HyperGAT proposed in [1] and UniGAT proposed in [2] both use attention mechanisms to determine the importance of incident hyperedges. UniGAT is shown to perform without OOM issues on DBLP in [2]. Please clarify why there was an OOM issue with HGAT on DBLP.
> > > > >
> > > > > **ANSWER:**
> > > > >
> > > > > Nice comment! There are two main reasons for the difference in GPU memory consumption between UniGAT and HGAT.
> > > > >
> > > > > 1. The number of parameters to be learned is different. UniGAT only needs to learn the Edge-level attention coefficients, which is about half of the learning parameters than HGAT.
> > > > >
> > > > > 2. The original implementation framework is different. The public source code of UniGAT uses the PYG framework, which is very friendly to the processing of sparse graphs (The sparsity rate of incidence matrix in DBLP is 0.00011), while HGAT public source code uses Pytorch directly, resulting in the need to generate a mask matrix of the same size as the incidence matrix when masking non-incident vertices(or hyperedges) on the dense attention coefficients matrix. This is the main reason for OOM.

---

> > > > > > ### Author Response · Authors · 2021-11-29
> > > > > > **The updated result of ChebNet on dblp(coauthorship) dataset ( reply to the further comments of Reviewer RmER ).**
> > > > > >
> > > > > > Thanks for your patience. The updated result of ChebNet on the dblp(co-authorship) dataset is shown in the table below.
> > > > > > We are wondering whether your concerns have been addressed properly. We would be glad to answer any further questions you may have after reviewing the answers.
> > > > > >
> > > > > > |Methods|cora coauthorship((11.8\%))|dblp coauthorship(0%)|Cora cocitation(47.0%)|Pubmed cocitation(80.5%)|citeseer cocitation(55.9%)|
> > > > > > |-|-|-|-|-|-|
> > > > > > |HGAT|65.42$\pm$1.5 | OOM |52.21$\pm$3.52 | 46.28$\pm$0.53|38.32$\pm$1.5
> > > > > > |ChebNet|42.27$\pm$2.17 |*87.87$\pm$0.24*| 45.55$\pm$2.37 | 65.73$\pm$1.3| 48.44$\pm$0.91 |
> > > > > > |SSGC | 72.04$\pm$1.2 | 88.61$\pm$0.16 | 68.79$\pm$2.1  | 74.49$\pm$1.3 | 60.52$\pm$1.7
> > > > > > GHCN(ours)| 74.79$\pm$0.91 | 89.04$\pm$ 0.19 |69.45$\pm$ 2.0 |**75.37$\pm$1.2** | 62.67$\pm$1.2
> > > > > > SHSC(ours) | **76.05$\pm$ 0.75** | **89.17$\pm$0.21**  | **70.64$\pm$1.8** | 75.08$\pm$ 1.1 |**65.14$\pm$0.97**

---

> > > > > > > ### Author Response · Authors · 2021-11-30
> > > > > > > **Thanks again for your comments!**
> > > > > > >
> > > > > > > Dear Reviewer:
> > > > > > >
> > > > > > > Thanks again for your valuable comments! It has been very helpful for improving the work.
> > > > > > > We are looking forward to your feedback!
> > > > > > >
> > > > > > > Best,
> > > > > > >
> > > > > > > The authors.

---

> > > > > > > > ### Comment · Reviewer_RmER · 2021-11-30
> > > > > > > > **Thanks again for the response**
> > > > > > > >
> > > > > > > > Dear authors,
> > > > > > > >
> > > > > > > > Thanks again for the additional experiments. Please clarify if learning rate was the only sensitive hyperparameter for ChebNet and other models or there were others (e.g., hidden size, dropout, depth, the value of K used in ChebNet which denotes the maximum number of hops, etc.).
> > > > > > > >
> > > > > > > > Since GHCN is GCN with the Hypergraph Laplacian proposed by the authors, another important baseline is GCN on the clique expansion which is known to be very similar to / same as HGNN [1] in the literature. Please clarify why the results of HGNN in this paper differ significantly, especially on PubMed and Citeseer co-citation datasets, from those reported in the UniGNN paper [2].
> > > > > > > >
> > > > > > > > [1] Hypergraph Neural Networks, AAAI'19.
> > > > > > > > [2] UniGNN: a Unified Framework for Graph and Hypergraph Neural Networks, IJCAI'21
> > > > > > > >
> > > > > > > > Best,
> > > > > > > > RmER

---

> > > > > > > > > ### Author Response · Authors · 2021-11-30
> > > > > > > > > **Answers to further comments**
> > > > > > > > >
> > > > > > > > > Thanks for your careful comments posted around one hour before the deadline!
> > > > > > > > > Due to super limited time, we can only give a rough answer here.  We will elaborate more in the final version.
> > > > > > > > >
> > > > > > > > > ### Comment:
> > > > > > > > > >Please clarify if learning rate was the only sensitive hyperparameter for ChebNet and other models or there were others (e.g., hidden size, dropout, depth, the value of K used in ChebNet which denotes the maximum number of hops, etc.).
> > > > > > > > >
> > > > > > > > > **ANSWER:**
> > > > > > > > > Yes, the learning rate is the most significant factor affecting the variance of ChebNet. For other  models, We have not been able to verify all hyper-parameter in the limited rebuttal time.
> > > > > > > > > ### Comment:
> > > > > > > > > >Since GHCN is GCN with the Hypergraph Laplacian proposed by the authors, another important baseline is GCN on the clique expansion which is known to be very similar to / same as HGNN [1] in the literature. Please clarify why the results of HGNN in this paper differ significantly, especially on PubMed and Citeseer co-citation datasets, from those reported in the UniGNN paper [2].
> > > > > > > > >
> > > > > > > > > **ANSWER:**
> > > > > > > > >
> > > > > > > > > We think this is due to the following reasons that lead to the difference between our HGNN results and [2].
> > > > > > > > >
> > > > > > > > > 1) The results of [2] are copied from HYPERSAGE [3]. However,  we  doubted about the results of HGNN from [3]. Therefore we had emailed the authors of HYPERSAGE asking about this  on April 2021,  but got no response. In a word, the results from [2] (copied HYPERSAGE) are not reproducible.
> > > > > > > > >
> > > > > > > > > 2) This may be caused by different random seeds adopted.
> > > > > > > > >
> > > > > > > > >
> > > > > > > > >
> > > > > > > > > [1] Hypergraph Neural Networks, AAAI'19.
> > > > > > > > >
> > > > > > > > > [2] UniGNN: a Unified Framework for Graph and Hypergraph Neural Networks, IJCAI'21
> > > > > > > > >
> > > > > > > > > [3] Arya, Devanshu, et al. "HyperSAGE: Generalizing Inductive Representation Learning on Hypergraphs." arXiv preprint arXiv:2010.04558 (2020).

---

> > > > > > > > > > ### Comment · Reviewer_RmER · 2021-11-30
> > > > > > > > > > **Thanks for the clarification**
> > > > > > > > > >
> > > > > > > > > > Thanks for the clarification!
> > > > > > > > > >
> > > > > > > > > > Citation/co-citation/co-authorship datasets such as DBLP, Cora, Citeseer, Pubmed are well-known to be homophilic (most edges/hyperedges contain nodes of the same label). As a consequence of this, methods exploiting graph/hypergraph structure in addition to vertex features are well-known to outperform/match feature-only baselines such as MLP. It is highly unlikely that a popular and well-understood model such as GCN (on clique expansion) is significantly less effective than MLP (in which all vertices are treated as isolated vertices) as reported in Table 1 under Pubmed, Citeseer.
> > > > > > > > > >
> > > > > > > > > > My score will remain the same since it is unclear how significant GHCN is over GCN on the clique expansion. The authors are suggested to develop and carefully tune GCN on clique expansion if there are (1) issues with publicly available codes, and (2) differences with numbers reported in other papers.

---

> > > > > > > > > > > ### Author Response · Authors · 2021-11-30
> > > > > > > > > > > **Answer**
> > > > > > > > > > >
> > > > > > > > > > > Note that HGNN is not GCN.
> > > > > > > > > > >
> > > > > > > > > > > We have discussed on the performance of HGNN in the main text in Sec. 4.2 below, please do read them:
> > > > > > > > > > >
> > > > > > > > > > > > It’s worth noting that our methods gain superior
> > > > > > > > > > > performance on disconnected datasets compared to HGNN. HGNN shows poor performance on
> > > > > > > > > > > disconnected datasets, mainly due to the row in the adjacency matrix of equivalent undigraph
> > > > > > > > > > > corresponding to an isolated point is 0, resulting direct loss of its vertex information (an example
> > > > > > > > > > > can see Fig. 4 in appendix). And our methods utilize the renormalization trick, which can
> > > > > > > > > > > maintain the features of isolated vertices during aggregation.
> > > > > > > > > > >
> > > > > > > > > > >
> > > > > > > > > > > Furthermore, we have tried our best to tune hyperparameters and will do more. However the main observations  in this paper    remain the same.

---

> > > > > > > > > > > > ### Comment · Reviewer_RmER · 2021-11-30
> > > > > > > > > > > > **Thanks for the reply**
> > > > > > > > > > > >
> > > > > > > > > > > > Thanks for the extensive effort during the response period and the clarification on HGNN.
> > > > > > > > > > > >
> > > > > > > > > > > > If HGNN is not exactly GCN on clique expansion of hypergraph, then it is important and fundamental to compare GHCN with GCN on clique expanded graph since it is a straightforward baseline for hypergraphs with edge-independent vertex weights. And this baseline should be equipped with the renormalisation trick since it was originally proposed in the GCN paper [1].
> > > > > > > > > > > >
> > > > > > > > > > > > [1] Semi-Supervised Classification with Graph Convolutional Networks, ICLR'17

---

> > > > > > > > > > > > > ### Author Response · Authors · 2021-11-30
> > > > > > > > > > > > > **Thanks for your advice about the baseline !**
> > > > > > > > > > > > >
> > > > > > > > > > > > > Thanks for your comments and advice.
> > > > > > > > > > > > >
> > > > > > > > > > > > > Recall that the GHCN follows the derivation of ChebNet(with renormalization trick of GCN) after getting the generalized hypergraph Laplacian. We have replied under the reviewer zaDF (Part 2, as quoted below) about the two models we proposed.
> > > > > > > > > > > > >
> > > > > > > > > > > > > >These two spectral convolution operations are actually deduced from the corresponding undirected graph. **Our core contribution is to build up the equivalency between the defined generalized hypergraph and the undirected graph based on a perspective of random walk.** The equivalency is a plug-and-play theoretical framework that any undirected graph convolutional networks can be selected as a backbone for learning hypergraphs. In principle, we can use existing undirected graph techniques to deal with equivalent undirected graphs.
> > > > > > > > > > > > >
> > > > > > > > > > > > > Note that as stated above, the main contributions and observations would remain the same.
> > > > > > > > > > > > >
> > > > > > > > > > > > > Further, we compare GHCN with GCN(with renormalization) on the clique expanded graph. The results show that GHCN performs better than the baselines.
> > > > > > > > > > > > >
> > > > > > > > > > > > > |Methods|cora coauthorship((11.8\%))|dblp coauthorship(0%)|Cora cocitation(47.0%)|Pubmed cocitation(80.5%)|citeseer cocitation(55.9%)|
> > > > > > > > > > > > > |-|-|-|-|-|-|
> > > > > > > > > > > > > HGNN | 69.23$\pm$1.6 | 88.55$\pm$ 0.18 |55.60$\pm$ 1.8 |46.41$\pm$0.7 | 38.98$\pm$1.1
> > > > > > > > > > > > > |GCN(renorm) on clique expanded graph|74.87$\pm$0.82 |88.99$\pm$0.19 |69.19$\pm$1.8| 75.17$\pm$1.3 | 62.52$\pm$1.2
> > > > > > > > > > > > > GHCN (ours)| 74.79$\pm$0.91 | 89.04$\pm$ 0.19 |69.45$\pm$ 2.0 |**75.37$\pm$1.2** | 62.67$\pm$1.2
> > > > > > > > > > > > > SHSC (ours) | **76.05$\pm$ 0.75** | **89.17$\pm$0.21**  | **70.64$\pm$1.8** | 75.08$\pm$ 1.1 |**65.14$\pm$0.97**

---

> > > > > > > > > > > > > > ### Comment · Reviewer_RmER · 2021-11-30
> > > > > > > > > > > > > > **Thanks for the additional results**
> > > > > > > > > > > > > >
> > > > > > > > > > > > > > Thanks again for the effort. In section 4 of the paper, the paper says
> > > > > > > > > > > > > >
> > > > > > > > > > > > > > > The weight matrix of edges, we set, to be an identity matrix by default in our GHCN and SHSC models. Notably, although GHCN and SHSC are designed for EDVW-hypergraph learning, they work for EIVW hypergraph as well thanks to the unified Laplacian
> > > > > > > > > > > > > > in Corollary 1. In our experiments, **citation network classification belongs to EIVW-hypergraph learning tasks**
> > > > > > > > > > > > > >
> > > > > > > > > > > > > > It is still unclear how significant GHCN is from GCN on clique expanded graph when the task belongs to EIVW-hypergraph learning tasks. The differences between the two models and the optimal hyperparameters need to be discussed to make the contributions more compelling.

---

> > > > > > > > > > > > > > > ### Author Response · Authors · 2021-11-30
> > > > > > > > > > > > > > > **Thanks for the further comments!**
> > > > > > > > > > > > > > >
> > > > > > > > > > > > > > > Thanks for your attention again!
> > > > > > > > > > > > > > >
> > > > > > > > > > > > > > > However, the statement "GCN on clique expanded graph when the task belongs to EIVW-hypergraph learning tasks" is confusing. We would like to  discuss under the literal understanding:
> > > > > > > > > > > > > > >
> > > > > > > > > > > > > > > 1. The "expanded clique graph" means expanding each hyperedge to a clique **without considering hyperedge-vertex weights($\mathbf Q_1,\mathbf Q_2$)**. Because we can not directly and trivially get the weighted edges in a clique through the hyperedge-vertex weights($\mathbf Q_1,\mathbf Q_2$). So we directly adopted the $\mathbf{H}\mathbf{WD}_e^{-1} \mathbf{H}^{\top}$ as the weights of the clique graph(used in HGNN implicitly).
> > > > > > > > > > > > > > > In the last response, we have compared the GHCN with GCN on the expanded clique graph in the citation network (EIVW-hypergraph tasks). Our model performs better than GCN(on clique expanded graph) on EIVW-hypergraph mainly because of the $\rho$ introduced, and different results can be obtained by choosing different $\rho$(for example one can see appendix I.10 ). When $\rho$ is a power function $(\cdot)^{\sigma}$, we have previously provided sensitivity analysis in appendix I.7.
> > > > > > > > > > > > > > >
> > > > > > > > > > > > > > > 2. In another view, **the proposed random walk also provides a way to expand a hypergraph to a clique undigraph with weights matrix $\mathbf{Q_2}\mathbf{W} \rho(\mathbf{D}_e) \mathbf{Q_2}^{\top}$).** If you mean "GCN on clique expanded graph" in this way, "GCN on clique expanded graph" is our model GHCN.

---

### Official Review · Reviewer_NwNk · 2021-11-01

**Correctness:** 4
**Technical Novelty And Significance:** 3
**Empirical Novelty And Significance:** 3
**Recommendation:** 6
**Confidence:** 3

**Main Review:**

Strengths:
1. The theory for the equivalency between generalized hypergraph and undigraph is interesting, which allows using any GNNs designed for undigraphs to solve hypergraph problems in principle.
2. The strong empirical performance.
3. The introduction of two new tasks: protein quality assessment and protein fold classification for hypergraph GNNs.

Weaknesses:
1. The Notations part in Sec. 2 is not clear at first glance. For example, what is $q_e(u)$ and what is $q_(u)$? They are used without definitions.
2. The Introduction contains many unfamiliar terms for people not working on hypergraphs, such as edge-dependent (independent)-vertex weights. I am confused on their meanings until reading into Sec 2.1
3. The theory only shows that performing two-step random walk on weighted hypergraphs can be equivalent to performing random walk on a weighted graph under some conditions. However, it does not clearly tell whether the mapping from hypergraph to graph is injective, i.e., from the weighted graph you can perfectly recover the original hypergraph. Can the authors comment on that?

**Summary Of The Paper:**

This paper proves under what conditions the equivalency between a generalized hypergraph (a hypergraph with node weights $Q_1$ and $Q_2$ for two-step random walk) and an undirected graph holds. The theory applies to ordinary hypergraphs as a special case with $Q_1=Q_2=H$. Then, the authors derive the stationary random walk distribution, the hypergraph Laplacian, as well as the generalized spectral hypergraph convolution form leveraging the Laplacian. To alleviate oversmoothing, they further propose to use a diffusion kernel to build a simple hypergraph spectral convolution. Experiments on four tasks show better performance than state-of-the-art methods.

**Summary Of The Review:**

Though I am not familiar with the hypergraph field, this paper seems to provide some useful theory and techniques for hypergraph learning. I resort to other reviewers to evaluate the novelty and significance of the theory.

---

> ### Author Response · Authors · 2021-11-17
> **Response to Reviewer NwNk**
>
> Thank you for your careful and valuable comments!
>
> ### Comment:
> >1. The Notations part in Sec. 2 is not clear at first glance. For example, what is  $q(u)$  and what is  $q_e(u)$ ? They are used without definitions.
> 2. The Introduction contains many unfamiliar terms for people not working on hypergraphs, such as edge-dependent (independent) vertex weights. I am confused about their meanings until reading into Sec 2.1.
>
> **ANSWER:**
>
> Thank you for the constructive suggestions. We feel very sorry that our representation makes this work hard to understand for the researchers who are not working on hypergraphs.  We have added more explanation about the hypergraphs with edge-dependent vertex weights in the first paragraph of the introduction. And the $q_e(u)\in \mathbb{R}$ denotes the vertex weights of $v$ associated with the hyperedge $e$. $Q(u,e)=q_e(u)$ means the vertex weights matrix $\mathbf Q$ is edge-dependent. $q(u)\in \mathbb R$ is also a vertex weight, but the weight is not related to any hyperedge and is only related to vertex $u$. So when $Q(u,e)=q(u)$, the matrix $\mathbf Q$ is edge-independent.
>
>
> ### Comment:
> >3. The theory only shows that performing a two-step random walk on weighted hypergraphs can be equivalent to performing a random walk on a weighted graph under some conditions. However, it does not clearly tell whether the mapping from hypergraph to graph is injective, i.e., from the weighted graph you can perfectly recover the original hypergraph. Can the authors comment on that?
>
> **ANSWER:**
>
> The mapping from the hypergraphs to graphs is not injective. For example, given two generalized hypergraphs $\mathcal{H_1} = \mathcal{H}(\mathcal{V,E},\mathbf{W},\mathbf{Q}_1 ,\mathbf{Q}_2)$ and  $\mathcal{H_2} = \mathcal{H}(\mathcal{V,E},\mathbf{W},\mathbf{Q}_2 ,\mathbf{Q}_2)$ where $\mathbf Q_1,\mathbf Q_2$ are both edge-independent, then $\mathcal{H_1}\neq \mathcal{H_2}$. According to Theorem 1 and Corollary 1, $\mathcal{H_1}$ and $\mathcal{H_2}$ are both equivalent to a clique undigraph $\mathcal G^C$ with edge-weights matrix $\mathbf Q_2\mathbf W\rho(\mathbf D_e)\mathbf Q^{\top}_2$. Thank you again for giving worthwhile advice to improve this work. We would pay more attention to the detailed explanation in the final version.
>
>
> ### Comment:
> > Though I am not familiar with the hypergraph field, this paper seems to provide some useful theory and techniques for hypergraph learning. I resort to other reviewers to evaluate the novelty and significance of the theory.
>
> **ANSWER:**
>
> Our core theoretical contribution is building up the equivalency between the defined generalized hypergraphs and undirected graphs from a perspective of random walk, i.e. Theorem 1. The unified random walk and Theorem 1 are novel, and the whole framework of utilizing the equivalency theory for EDVW-hypergraph learning is unprecedented. Here, we further explain the significance of this theory:
> - i)The equivalency led by the random walk is meaningful and valuable. As we stated in the analysis of Theorem 1, the equivalency conditions give new insights from hypergraph foundation, hypergraph application, and hypergraph learning.
>
> - ii) This equivalency provides a plug-and-play theoretical framework that not only can be used to deduce the two spectral convolutions but also could seamlessly incorporate existing undirected graph models for (EDVW-)hypergraph learning.
>
>
> Finally, thank you once again for your recognition of our work.

---

> > ### Comment · Reviewer_NwNk · 2021-11-30
> > **Response to authors**
> >
> > Thanks the authors for the response. The theory establishes an equivalency between random walks on hypergraphs and graphs, however the mapping from hypergraph to graph is not injective, which indicates that there might be some ambiguity issues (different hypergraphs mapped to the same graph, thus having the same representations). I feel this part should be discussed in more depth in the revised paper. My score remains the same.

---

> > > ### Author Response · Authors · 2021-11-30
> > > **Answers to further comments**
> > >
> > > Thanks for your constructive suggestions and recognition! Due to very limited time, we can just give a rough answer here. We will elaborate more in the final version.
> > >
> > > We would like to discuss the non-injective mapping with the followed statements:
> > >
> > > 1. Our equivalence is essentially based on the transition matrix of the proposed random walk on hypergraphs.
> > >
> > > 2.  Therefore, different hypergraphs under the proposed random walk, satisfying conditions in Theorem 1, would lead to one same Markov process (same transition matrix). This is the essential reason why the map is not injective.
> > >
> > > 3. As the example proposed above: $\mathcal{H_1} = \mathcal{H}(\mathcal{V,E},\mathbf{W},\mathbf{Q}_1 ,\mathbf{Q}_2)$​ and $\mathcal{H_2} = \mathcal{H}(\mathcal{V,E},\mathbf{W},\mathbf{Q}_2 ,\mathbf{Q}_2)$​ where $\mathbf Q_1,\mathbf Q_2$​ are both edge-independent. Those two hypergraphs lead to a same underlying Markov Process which means that the **edge-vertex weights in the first-step**($\mathbf Q_1$) of our random walk lose its influence on the whole process. As a result, this underlying Markov Process depends only on the **second-step edge-vertex weights($\mathbf Q_2$​)**. So the two hypergraphs map to a same undirected graph.
> > >
> > > **For a deeper understanding of the non-injective map**：
> > >
> > >   As long as we design the hypergraph with the specificity information contained in $\mathbf Q_2$, it will not cause any ambiguity issue of hypergraph representations. The equivalent undirected graphs only retain the information of $\mathbf Q_2$. Therefore，this mapping can be viewed as a compression of redundant information ($\mathbf Q_1$).  It is worth noting that this issue does not have any impact on the derivation of our algorithms and experimental results.
> > >
> > >
> > > We will add detailed explanations regarding the property of the mapping from hypergraphs to graphs in the final version.

---

### Official Review · Reviewer_NQ7e · 2021-11-07

**Correctness:** 4
**Technical Novelty And Significance:** 3
**Empirical Novelty And Significance:** 4
**Recommendation:** 8
**Confidence:** 3

**Main Review:**

The main strengths of this paper are as follows:

+ The paper is well-written and clearly explains both pros and cons of the methods. The technical contributions are sufficiently novel and deep. The experimental results justify the main claims made by the paper.

+ There are a number of new ideas in the paper that may be useful independently.

+ The considered problem of developing Hypergraph convolutional NN is practically relevant and useful in many applications (the paper already shows 4 such applications).

The main weakness of the paper are as follows:

To a large extent I did not find any significant weakness of this paper. The paper was very well-written and they already go in significant depth on the weakness of their methods.

**Summary Of The Paper:**

This paper considers the problem of showing equivalency of Generalized Hypergraphs to undirected graphs (in the sense that a natural random walk on the hypergraph is equivalent to the natural random walk on a weighted undirected clique graph). They do so via establishing a Hypergraph Laplacian and identifying its properties that help prove this equivalence. They leverage this equivalency to build a Hypergraph  convolution neural network by viewing the weighted undirected graph as a lower-order encoder of hypergraphs. They use empirical studies to show that the constructed Hypergraph Convolution NN on four different network based classification task, where the underlying network structure forms a hypergraph.

**Summary Of The Review:**

Overall I am very positive about this paper.  AS mentioned above, I find the techniques and the evaluation to be solid with very little in terms of weakness.

---

> ### Author Response · Authors · 2021-11-17
> **Response to Reviewer NQ7e**
>
> Thank you very much for your comprehensive understanding of the content of our work! Your strong endorsement of the core contributions is a crucial support to our work.
>
> Our core theoretical contribution is building up the equivalency between the defined generalized hypergraph and the undirected graph based on a perspective of random walk, which is novel and valuable. We work on exploring the principles of hypergraph convolution design. Specifically, not only do we design the two hypergraph convolution networks but also provide the theoretical analysis to explore the essence of the hypergraph convolutions. Furthermore, we offer a new route to design hypergraph convolutional networks (via equivalency), as you mentioned, the route is very practical and effective in real-world applications.
>
> Finally, thank you once again for your recognition of our work, which will inspire us to continue to improve this area in further depth.

---

### Official Review · Reviewer_zaDF · 2021-11-08

**Correctness:** 3
**Technical Novelty And Significance:** 2
**Empirical Novelty And Significance:** 2
**Recommendation:** 5
**Confidence:** 3

**Main Review:**

Strengths
1. This paper provides some theoretical understanding of the equivalency of hypergraphs and undirected graphs.

Weakness
1. The paper is not very well written. The motivation is not clearly described.
2. Some concepts and definitions are not clearly presented.
3. Traditional GNN methods for undirected graphs should be included as baselines.

More comments

1. It would be better if the authors could provide more motivation for transferring hypergraphs into undirected graphs. What are the benefits? Are the equivalency through random walks good enough for downstream tasks? Will there be information loss when transferring hypergraphs into undirected graphs?
2. Definition 1 is not clearly presented. Specifically, why there are suddenly two Q matrices, what are their relation to the common definition of hypergraph, which only has a single matrix Q. Also, what is $\rho$ for?
3. What is the concept of ``oversmoothing issue'' for hypergraphs? Is it defined upon the corresponding undirected graphs?
4. In Section 3, are these two spectral operations developed based on the Laplacian matrix for the corresponding undirected graphs? If so, how are they different from the existing methods for undirected graphs? Why not directly adopt existing operations for undirected graphs? Do they have any specialty corresponding to the hypergraphs?
5. In Section 4.1, what are the benefits of modeling the citation datasets as hypergraphs? Also, other baselines for undirected graphs such as GCN and APPNP (the SHSC is a bit similar to APPNP) should be included.
6. In Section 4.3, why is constructing such hypergraphs for the protein reasonable? It would be better if the authors could provide more explanations.



**Summary Of The Paper:**

In this paper, the authors aim at building the equivalency between hypergraphs and undirected graphs. They introduce the concept of generalized hypergraphs and demonstrate their equivalency with undirgraphs in terms of random walk. They further proposed two spectral-based convolution operations for the undigraphs.


**Summary Of The Review:**

Overall, the motivation for building such equivalency is not very clearly presented. The proposed convolution operations seem to be ``general'' for undirected graphs but not specifically related to the hypergraphs (except for the involved Laplacian is derived from the hypergraph). Also, it would be better if the authors could include existing GNN methods for undirected graphs for comparison to demonstrate the superiority of their proposed framework.

---

> ### Author Response · Authors · 2021-11-17
> **Response to Reviewer zaDF (Part 1)**
>
> We thank the reviewer for the extensive comments that help us to improve our work. Please see our detailed answers below.
> ### Comment:
> >1. It would be better if the authors could provide more motivation for transferring hypergraphs into undirected graphs. What are the benefits? Are the equivalency through random walks good enough for downstream tasks? Will there be information loss when transferring hypergraphs into undirected graphs?
>
> **ANSWER:**
>
> Here we would like to elaborate the motivation and benefits of building the equivalency condition between hypergraphs and graphs.
>
> - 1) (About the motivation and benefits.) In a nut shell, our motivation of building the equivalency condition lays in the problem:  **under what conditions one can use the undigraph technical routes to construct a hypergraph Laplacian**. This problem can help us to understand the insight when and why we can leverage the techniques in GNNs to design models for hypergraph learning.
>
> - 2) (About our contribution.) Chitra & Raphael (2019)[1] prove that only edge-independent hypergraphs can be equivalent to the undirected graphs, while under our unified random walk on hypergraphs, an EDVW-hypergraph under the condition that $\mathbf Q_1 = k\mathbf Q_2$ can also utilize the undirected graph technique route to deduce its hypergraph Laplacian and convolutional models.
>
> - 3) (Fitting for the downstream tasks.) Compared with other methods for establishing equivalency, our approach is a good fit for downstream tasks. Intuitively, the method we use will be more reasonable and the loss of information is smaller since our GHCN and SHSC are directly based on the random walk-based Laplacian and diffusion kernel led by the random walk, respectively. Then the message aggregation from the equivalent graph will be equal to the corresponding hypergraph via the connection of random walks.
>
> For information loss:
> - Information loss exists.
> According to corollary 1, the equivalent undigraph Laplacian $\mathbf L$ is only related to $\mathbf Q_2$, independent of $\mathbf  Q_1$, implying possible information loss of $\mathbf Q_1$. Notably，it can be viewed as information compression when condition (2) holds (i.e. $\mathbf Q_1=k\mathbf Q_2$).  Actually, in practical applications we can directly design $\mathbf Q_1=\mathbf Q_2$, then there is no information loss.
>
> We hope that the above elaboration would eliminate your confusion and sincerely appreciate that the reviewer raises any further questions if you are still confused by our answers.
>
> ### Comment:
>
> >2. Definition 1 is not clearly presented. Specifically, why there are suddenly two Q matrices, what are their
> relation to the common definition of hypergraphs, which only has a single matrix Q. Also, what is $\rho$ for?
>
> **ANSWER:**
>
> - The two vertex weights matrix $\mathbf Q_1, \mathbf Q_1$ in random walk:
>
> We only provided a limited description in Definition 1 due to the space limitation in the main text.
>
> Formally, the defined hypergraph with two $\mathbf Q$ is a generalization of the common hypergraph ($\mathbf {Q}_1=\mathbf {Q}_2=\mathbf {H}$). Actually, the introduced $\mathbf Q_1$ has realistic and theoretical significance.
>
> 1. Modeling the fine-grained high-order information.
> The two matrices can be constructed heuristically to model the more fine-grained high-order information associated with the first step and the second step of random walk.  $\mathbf Q_1\neq \mathbf{Q_2}$ means that the contribution of $v$ to edge $e$ in the in-edge process is different from the out-edge process. Furthermore, the introduced two $\mathbf Q$ make the random walk be able to capture the more comprehensive hypergraph information by establishing equivalency to digraphs. (We have built the equivalency between hypergraphs and digraphs in Appendix E. The equivalency between undigraph can be viewed as a special case of digraph).
>
> 2. Providing the theoretical basis to investigate the equivalency between EDVW-hypergraphs and undigraphs.
>
>
> - The explanation of $\rho$:
>
> The $\rho$ acts on the degree of the hyperedge and is used to control the random process. We set $\rho(\cdot)=(\cdot)^\sigma$~(power function) for example. According to Definition 1, for positive values of $\sigma$, it is clear that hyperedges with larger sizes will dominate the random process. Conversely, when $\sigma$ is negative, hyperedges with small size are likely to drive the random walk process.
>
> ### Comment:
> > 3. What is the concept of ''over-smoothing issue'' for hypergraphs? Is it defined upon the corresponding undirected graphs?
>
> **ANSWER:**
>
> In our paper, the over-smoothing issue for hypergraphs means the nodes embeddings of hypergraphs are indistinguishable.
> The equivalent clique graphs have the same vertices as hypergraphs, so the over-smoothing issue for hypergraphs is defined upon the corresponding undirected graphs. We will add this description in the final version.

---

> > ### Author Response · Authors · 2021-11-17
> > **Response to Reviewer zaDF (Part 2)**
> >
> > ### Comment:
> > > 4. In Section 3, are these two spectral operations developed based on the Laplacian matrix for the corresponding undirected graphs? If so, how are they different from the existing methods for undirected graphs? Why not directly adopt existing operations for undirected graphs? Do they have any specialty corresponding to the hypergraphs?
> >
> > **ANSWER:**
> >
> > These two spectral convolution operations are actually deduced from the corresponding undirected graph. **Our core contribution is to build up the equivalency between the defined generalized hypergraph and the undirected graph based on a perspective of random walk.** The equivalency is a plug-and-play theoretical framework that any undirected graph convolutional networks can be selected as a backbone for learning hypergraphs. In principle, we can use existing undirected graph techniques to deal with equivalent undirected graphs.
> >
> > In addition, we would like to further explain the two proposed models.
> > Actually, GHCN is directly derived from GCN, replacing the graph Laplacian with the hypergraph Laplacian. SHSC is designed by ourself and the introduced discounted factor $\alpha$ can alleviate long-range information aggregation, which can prevent over-smoothing issues as the number of layers grows (see Appendix I.6 table 14 empirically and F.4 theoretically). Moreover, SHSC is able to capture both low-frequency and high-frequency information of the graph signal, depending on the selection of the appropriate hyper-parameters ($\alpha$,$\beta$) (more details can be seen from table 15 in Appendix I.8).
> >
> > Here we added APPNP, ChebNet as complementary baselines, and the experimental results are shown in the following table (We update table 19 in Appendix I.13 of our manuscript). The experimental results reveal that our GHCN and SHSC outperform the baselines. We would add other experiments in the manuscript when possible.
> >
> > |Methods|cora coauthorship(11.8\%)|dblp coauthorship(0%)|Cora cocitation(47.0%)|Pubmed cocitation(80.5%)|citeseer cocitation(55.9%)|
> > |-|-|-|-|-|-|
> > |APPNP|68.25$\pm$1.9 |86.99$\pm$0.33 |64.50$\pm$1.7 | 72.02$\pm$1.4|55.05$\pm$2.0
> > |ChebNet|42.27$\pm$2.17 |85.81$\pm$5.22| 45.55$\pm$2.37 | 65.73$\pm$1.3| 48.44$\pm$0.91 |
> > GHCN (ours)| 74.79$\pm$0.91 | 89.04$\pm$0.19 |69.45$\pm$2.0 |**75.37$\pm$1.2** | 62.67$\pm$1.2
> > SHSC (ours) | **76.05$\pm$ 0.75** | **89.17$\pm$0.21**  | **70.64$\pm$1.8** | 75.08$\pm$1.1 |**65.14$\pm$0.97**
> >
> > (The ratio of isolated vertices is shown in the parentheses)
> >
> > ### Comment:
> > >5. In Section 4.1, what are the benefits of modeling the citation datasets as hypergraphs? Also, other baselines for undirected graphs such as GCN and APPNP (the SHSC is a bit similar to APPNP) should be included.
> >
> >
> > **ANSWER:**
> >
> > Indeed, the citation hypergraph dataset is a benchmark in the hypergraph community for evaluating the performance of hypergraph NNs. The dataset is constructed by Yadati et al. [2]. Specifically, those hypergraphs are created from Cora, Citeseer, Pubmed, and DBLP by assigning each document as a node and each hyperedge represents (a) all documents co-authored by an author in co-authorship dataset and (b) all documents cited together by a document in co-citation dataset. The relations of co-authorship or co-citation are complex and go beyond pairwise associations, so researchers assume a hypergraph will be powerful than a graph to model those relationships.
> >
> >  In addition, we need to clarify that GHCN acting on hypergraph is GCN acting on the equivalent clique undirected graphs. So here we just add the experiments of APPNP acting on the equivalent clique undirected graphs on the citation classification task and the results can be seen in the table above.
> >
> >
> >
> > ### Comment:
> > >6. In Section 4.3, why is constructing such hypergraphs for the protein reasonable? It would be better if the authors could provide more explanations.
> >
> > **ANSWER:**
> >
> > A protein is folded by chains of amino acids to form a 3D structure. The function of a protein is primarily determined by its 3D structure, which contains the high-order interaction of amino-acids. Inspired by
> > Maruyama et al. [3] (Constructing the protein hypergraph to solve protein conformation problem), we represent the protein as an EDVW-hypergraph to explore the 3D structure and the high-order relationship between amino-acids to obtain a comprehensive protein representation via hypergraph NNs. Additionally, the experiments of protein learning also suggest that the hypergraph protein is better than the sequence protein and the graph protein.

---

> > > ### Author Response · Authors · 2021-11-17
> > > **Reference**
> > >
> > > ### Reference
> > >
> > > [1] Uthsav Chitra and Benjamin Raphael.  Random walks on hypergraphs with edge-dependent vertex weights. In International Conference on Machine Learning, pp. 1172–1181. PMLR
> > >
> > > [2] Naganand Yadati,  Madhav Nimishakavi,  Prateek Yadav,  Vikram Nitin,  Anand Louis,  and ParthaTalukdar.  Hypergcn:  A new method for training graph convolutional networks on hypergraphs.In Advances in Neural Information Processing Systems, pp. 1511–1522, 2019.
> > >
> > > [3] Osamu Maruyama, Takayoshi Shoudai, Emiko Furuichi, Satoru Kuhara, and Satoru Miyano. Learning confirmation rules. In International Conference on Discovery Science, pp. 243–257. Springer, 2001

---

> > > > ### Comment · Reviewer_zaDF · 2021-11-26
> > > > **Thanks for the response**
> > > >
> > > > Thank the authors for the response.
> > > >
> > > > I have the following further concerns.
> > > > 1. As per the authors, they transform the hypergraphs into undirected simple graphs, and then the proposed methods (GHCN and SHSC) are indeed designed for undirected graphs. The current writing of the paper makes it feel like their methods are proposed specifically for hypergraphs. However, these methods are not directly related to hypergraphs.  The authors need to make this clear. Furthermore, it is a bit inappropriate that the authors claim they proposed GHCN, as this is a direct application of GCN to the undirected graphs (transferred from the hypergraphs) and the contribution of transforming hypergraphs to undirected graphs has been separately claimed and discussed.
> > > >
> > > > 2. The evaluation is problematic especially for those citation datasets, which are commonly adopted benchmarks in GNN research. These datasets have been transformed to hypergraphs and then again transformed back to undirected simple graphs (by the methods proposed in this paper). So, to demonstrate such complicated transformations are helpful and meaningful, the authors need to include GNN baselines and directly apply them to the original simple graphs (without transforming them to hypergraphs). Furthermore, the authors need to include more GNN methods and apply them to the undirected graphs transformed from the hypergraphs as well. By the way, according to the current literature, APPNP achieves stronger performance on Cora, Citeseer, Pubmed than GCN if they are directly applied to the original simple graphs. However, according to the author's additional results in the rebuttal, APPNP performs much worse than GCN (or GHCN) when applied to the undirected graphs transformed from the hypergraphs. The authors may need to conduct more careful hyperparameter-tuning for APPNP (and other methods if included) for a fair comparison.
> > > >
> > > > 3. As per the authors' response, the "over-smoothing issue" for hypergraphs is defined based on their corresponding undirected graphs. There are quite a lot of works addressing this issue for undirected simple graphs. Hence, if the authors claim "addressing over-smoothing issue'' as a major contribution, it would be better if they can include those methods (dealing with over-smoothing issue) developed for simple graphs as baselines.

---

> > > > > ### Author Response · Authors · 2021-11-28
> > > > > **Reply to the further comments raised by the Reviewer zaDF  (part 1) !**
> > > > >
> > > > > Thank you very much for your detailed comments and valuable suggestions！ Note that there may exist some misunderstandings regarding this work, which we will clarify in detail below.
> > > > >
> > > > >
> > > > > ### Comment:
> > > > > > However, these methods are not directly related to hypergraphs. The authors need to make this clear. Furthermore, it is a bit inappropriate that the authors claim they proposed GHCN, as this is a direct application of GCN to the undirected graphs (transferred from the hypergraphs) and the contribution of transforming hypergraphs to undirected graphs has been separately claimed and discussed.
> > > > >
> > > > > **ANSWER:**
> > > > >
> > > > > Thanks for your constructive suggestions!
> > > > >
> > > > > We will add a more refined description of our methods. For GHCN,  it is derived from the generalized hypergraphs proposed by us.
> > > > > The original statement that
> > > > > GHCN is proposed is to highlight the difference from GCN (or HGNN). We will revise the statement in the final version.
> > > > >
> > > > > ### Comment:
> > > > >
> > > > > > The evaluation is problematic especially for those citation datasets, which are commonly adopted benchmarks in GNN research. These datasets have been transformed to hypergraphs and then again transformed back to undirected simple graphs (by the methods proposed in this paper). So, to demonstrate such complicated transformations are helpful and meaningful, the authors need to include GNN baselines and directly apply them to the original simple graphs (without transforming them to hypergraphs).
> > > > >
> > > > >
> > > > > **ANSWER:**
> > > > >
> > > > >
> > > > > This may be a misunderstanding of these citation network datasets.
> > > > >
> > > > > 1. We need to clarify that these datasets used here are benchmarks for hypergraph-NNs and not for GNNs. Actually, many higher-order relationships in the real world can be directly modeled as hypergraphs, with supporting materials from [1-7]. The purpose of our experiments in citation networks is to evaluate how well our methods can obtain better performance than other existing hypergraph NNs for a given hypergraph.
> > > > >
> > > > > 2. The construction of the citation datasets used here is not directly related to the citation datasets used in GNN research. In other words, citation hypergraphs are not transformed from citation simple graphs.  Specifically, citation datasets in GNNs are modeled as graphs: papers are vertices and citation relationships between two papers are represented as edges. On the other hand, citation datasets used here are constructed directly from citation relationships or coauthorships. For example, in Cora(coauthorship), the 1072 authors are represented as 1072 hyperedges, each of which incidents several vertices(papers), while in Cora(GNNs), each edge in the 5429 edges just represents the citation relationship of pair-vertex(paper). So the adjacent relationship between Cora(GNNs) and Cora(coauthorship) is essentially different.
> > > > >
> > > > > 3. Applying the GNN baselines to the original citation simple graphs may not be used to demonstrate the validity of our transformation due to the difference between citation hypergraph and simple graph(As discussed in point 2). Actually, this paper is focusing on the performance of this transformation on downstream tasks of hypergraph learning. Therefore, we can consider this transformation to be helpful and meaningful as long as its derived hypergraph CNNs can perform better than the existing hypergraph networks.
> > > > >
> > > > > References:
> > > > > [1] Naganand Yadati, Madhav Nimishakavi, Prateek Yadav, Vikram Nitin, Anand Louis, and ParthaTalukdar. Hypergcn: A new method for training graph convolutional networks on hypergraphs.In Advances in Neural Information Processing Systems, pp. 1511–1522, 2019.
> > > > >
> > > > > [2] Y. Dong, W. Sawin, and Y. Bengio, “Hnhn: Hypergraph networks with hyperedge neurons,” Graph Representation Learning and Beyond Workshop at ICML 2020
> > > > >
> > > > > [3] Naganand Yadati. Neural message passing for multi-relational ordered and recursive hypergraphs.Advances in Neural Information Processing Systems, 33, 2020
> > > > >
> > > > > [4] Kaize Ding, Jianling Wang, Jundong Li, Dingcheng Li, and Huan Liu. Be more with less: Hypergraph attention networks for inductive text classification. In Proceedings of the 2020 Conference on Empirical Methods in Natural Language Processing (EMNLP), pp. 4927–4936, 2020.
> > > > >
> > > > > [5] Zhang, Z., Lin, H., and Gao, Y. Dynamic hypergraph structure learning. In Proceedings of the Twenty-Seventh International Joint Conference on Artifificial Intelligence, IJCAI-18, pp. 3162–3169. International Joint Conferences on Artifificial Intelligence Organization, 7 2018a.
> > > > >
> > > > > [6] Wang, Jianling, et al. "Session-based Recommendation with Hypergraph Attention Networks." Proceedings of the 2021 SIAM International Conference on Data Mining (SDM). Society for Industrial and Applied Mathematics, 2021.
> > > > >
> > > > > [7] Kim, Eun-Sol, et al. "Hypergraph attention networks for multimodal learning." Proceedings of the IEEE/CVF Conference on Computer Vision and Pattern Recognition. 2020.

---

> > > > > > ### Author Response · Authors · 2021-11-28
> > > > > > **Reply to the further comments raised by the Reviewer zaDF (part 2) !**
> > > > > >
> > > > > > ### Comment:
> > > > > > > Furthermore, the authors need to include more GNN methods and apply them to the undirected graphs transformed from the hypergraphs as well.
> > > > > >
> > > > > > **ANSWER:**
> > > > > >
> > > > > > Thanks for your valuable suggestion. We will add more undirected graph NNs that process the equivalent undirected graphs to verify the effectiveness of the existing undirected graph techniques in learning hypergraphs via equivalency.
> > > > > >
> > > > > > Meanwhile, our focus is on the generic pipeline: utilizing undirected graph techniques to solve the hypergraph problems. This is theoretically feasible both for existing and for any future design of undirected graph convolutions.
> > > > > > Note that we are focusing on proposing and theoretically proving that this pipeline can be used to solve hypergraph learning problems as a subroute. As a result, we have not made it a priority in our current work to establish a benchmark for all existing GNN models.
> > > > > >
> > > > > > ### Comment:
> > > > > > > By the way, according to the current literature, APPNP achieves stronger performance on Cora, Citeseer, Pubmed than GCN if they are directly applied to the original simple graphs. However, according to the author's additional results in the rebuttal, APPNP performs much worse than GCN (or GHCN) when applied to the undirected graphs transformed from the hypergraphs. The authors may need to conduct more careful hyperparameter-tuning for APPNP (and other methods if included) for a fair comparison.
> > > > > >
> > > > > > **ANSWER:**
> > > > > >
> > > > > > Thanks for your careful analysis and constructive suggestion.
> > > > > > We will conduct more careful hyperparameter-tuning for APPNP and update the results as soon as possible.
> > > > > >
> > > > > > However, we can't promise the APPNP will outperform GCN due to the bias from the datasets. In practice, since simple graph and hypergraph datasets are different forms of data, the performance of undirected graph neural networks on hypergraphs cannot be guaranteed to exactly follow the original performance on graphs.
> > > > > >
> > > > > >
> > > > > > ### Comment:
> > > > > > > Hence, if the authors claim "addressing over-smoothing issue'' as a major contribution, it would be better if they can include those methods (dealing with over-smoothing issue) developed for simple graphs as baselines.
> > > > > >
> > > > > > **ANSWER:**
> > > > > >
> > > > > > This may be a misunderstanding of the contribution about "addressing over-smoothing issue". We just provide an analysis for SHSC and explain how it can mitigate over-smoothing. Indeed, we
> > > > > > did not claim it is our major contribution, and this work is not targeted for
> > > > > > designing a trick specifically for handling the over-smoothing issue of hypergraph learning.
> > > > > > However, we will leave it as an important future work.

---

> > > > > > > ### Comment · Reviewer_zaDF · 2021-11-29
> > > > > > > **Thanks for the response**
> > > > > > >
> > > > > > > Thanks for the response. The response addresses some of my concerns.
> > > > > > >
> > > > > > > However, there remain a few questions not properly answered. For example, it is clearly stated in the paper that SHSC is proposed to address the over-smoothing issue (in the abstract, introduction, and other places). More importantly, SHSC is not related to hypergraphs by design (it is designed for simple undirected graphs). Without proper experiments and comparison with other baselines, the effectiveness of the proposed method cannot be verified. Note that the contribution of transferring hypergraphs to simple undirected graphs (Section 2) has been separately claimed. Hence, the contribution of Section 3 should be carefully considered.  More specifically, when considering the novelty/contribution of the methods proposed in Section 3, we need to compare them with existing methods for simple undirected graphs. Also, it is important for the authors to empirically compare with existing baselines (for simple undirected graphs).

---

> > > > > > > > ### Author Response · Authors · 2021-11-29
> > > > > > > > **Reply to the further comments raised by the Reviewer zaDF!**
> > > > > > > >
> > > > > > > > ### Comment:
> > > > > > > > > It is clearly stated in the paper that SHSC is proposed to address the over-smoothing issue (in the abstract, introduction, and other places).
> > > > > > > >
> > > > > > > > **ANSWER:**
> > > > > > > >
> > > > > > > > Thank you for your careful review and valuable suggestions!
> > > > > > > > This is indeed a misunderstanding caused by our inappropriate expressions in the paper. We will describe the over-smoothing issue as an analysis of SHSC rather than a contribution in the final version.
> > > > > > > >
> > > > > > > >
> > > > > > > > ### Comment:
> > > > > > > > > More importantly, SHSC is not related to hypergraphs by design (it is designed for simple undirected graphs). Without proper experiments and comparison with other baselines, the effectiveness of the proposed method cannot be verified. Note that the contribution of transferring hypergraphs to simple undirected graphs (Section 2) has been separately claimed. Hence, the contribution of Section 3 should be carefully considered. More specifically, when considering the novelty/contribution of the methods proposed in Section 3, we need to compare them with existing methods for simple undirected graphs. Also, it is important for the authors to empirically compare with existing baselines (for simple undirected graphs).
> > > > > > > >
> > > > > > > > **ANSWER:**
> > > > > > > >
> > > > > > > > Thanks for your thoughtful consideration!
> > > > > > > > Note we have always claimed that our core contribution is the equivalency condition between the defined hypergraphs and undirected graphs. This view provides readers a simple and convenient perspective to understand the intuition of designing spectral convolutions for hypergraphs, i.e. those models can be considered as deducing from the Laplacian of corresponding undirected graphs. However, from the insight, this equivalency is built upon the proposed random walk of the underlying Markov process in the hypergraph. This means that SHSC can be directly derived from the route: **random walk on the hypergraph --> transition matrix of underlying Markov process --> discounted Markov diffusion kernel --> SHSC** without considering its corresponding undirected graphs. As a result, we think SHSC can be seen as designed both for hypergraphs and simple undirected graphs. Therefore, we believe that your point is well taken and we have also verified the validity of SHSC directly in the undirected graph. We have added the experiments of comparing with existing methods for simple undirected graphs here:
> > > > > > > >
> > > > > > > > |Methods|cora(GNNs)  |citeseer(GNNs)  | pubmed(GNNs)
> > > > > > > > |-|-|-|-|
> > > > > > > > |GCN|81.4 $\pm$ 0.4 | 70.9$\pm$0.5 | 79.0$\pm$0.4
> > > > > > > > |APPNP| 83.3$\pm$0.5 | 71.7$\pm$0.6 | 80.1$\pm$0.2
> > > > > > > > |SSGC[1]| 82.5$\pm$0.10 | 73.0$\pm$0.01| 79.9$\pm$0.10
> > > > > > > > |ChebNet|78.0$\pm$0.4| 70.1$\pm$0.5| 78.0$\pm$0.4 |
> > > > > > > > |DGC[2]| 83.3$\pm$0.0 | 73.3$\pm$0.1 | 80.3$\pm$0.1
> > > > > > > > |SHSC(ours)|  83.3$\pm$0.05 |  73.5$\pm$0.01 | 80.4$\pm$0.01
> > > > > > > >
> > > > > > > > The results suggest that our SHSC also achieves competitive performance in simple graph learning.  We will add the results to the final version.
> > > > > > > >
> > > > > > > > ### Reference:
> > > > > > > >
> > > > > > > >
> > > > > > > > [1] Zhu, Hao, and Piotr Koniusz. "Simple spectral graph convolution." International Conference on Learning Representations. 2020.
> > > > > > > >
> > > > > > > > [2] Wang Y, Wang Y, Yang J, et al. Dissecting the Diffusion Process in Linear Graph Convolutional Networks[J]. In Advances in Neural Information Processing Systems, 2021.

---

### Author Response · Authors · 2021-11-21
**Kind reminder to the Reviewers**

Dear Reviewers,

Thanks a lot for your valuable comments that help us to improve our work! We are wondering whether your concerns have been addressed properly. We would be glad to answer any further questions you may have after reviewing the answers.

Best regards,

The authors.

---

### Decision · Program_Chairs · 2022-01-20

**Decision:**

Reject

**Comment:**

Standard algorithms for deep hypergraph learning have not been designed for hypergraphs with edge-dependent vertex weights (EDVWs), where the weight of a vertex can depend on the edge of which it isa member. This paper develops a connection between EDVW-hypergraphs and undirected simple graphs, thus enabling the use of existing undirected-graph neural networks as subroutines. This is done via a unified random-walk framework.

(Two typos: ``equivalency" should be ``equivalence", and ``undigraphs" should be ``undirected graphs".)

The theory of equivalence between EDVW-hypergraphs and undirected graphs via random walks is a good contribution. The experimentation across different domains is laudable.

However, there are concerns over the lack of key baselines in the experiments. The author rebuttal has presented additional results with some baselines: sensitive hyperparameters (e.g., learning rate) are not tuned for the baselines. The clarity of the paper is mixed. The map from hypergraphs to graphs is not injective, so there could be ambiguity issues (different hypergraphs mapped to the same graph, thus having the same representations).

Also, the contributions of Section 3 (designed for simple undirected graphs alone) do not appear significantly novel.